# Nested Expectations with Kernel Quadrature

**Zonghao Chen** [1]  **Masha Naslidnyk** [1]  **François-Xavier Briol** [2]

## Abstract

This paper considers the challenging computational task of estimating nested expectations. Existing algorithms, such as nested Monte Carlo or multilevel Monte Carlo, are known to be consistent but require a large number of samples at both inner and outer levels to converge. Instead, we propose a novel estimator consisting of nested kernel quadrature estimators and we prove that it has a faster convergence rate than all baseline methods when the integrands have sufficient smoothness. We then demonstrate empirically that our proposed method does indeed require fewer samples to estimate nested expectations on real-world applications including Bayesian optimisation, option pricing, and health economics.

## 1. Introduction

We consider the computational task of estimating a nested expectation, which is the expectation of a function that itself depends on another unknown conditional expectation. More precisely, let $\mathbb{Q}$ be a Borel probability measure with density $q$ on $\Theta$ and $\mathbb{P}_\theta$ a Borel probability measure with density $p_\theta$ on $\mathcal{X} \subseteq \mathbb{R}^{d_\mathcal{X}}$ which is parameterized by $\theta \in \Theta \subseteq \mathbb{R}^{d_\Theta}$. Given integrable functions $f : \mathbb{R} \to \mathbb{R}$ and $g : \mathcal{X} \times \Theta \to \mathbb{R}$, we are interested in estimating:

$$I := \mathbb{E}_{\theta \sim \mathbb{Q}} \left[ f \left( \mathbb{E}_{X \sim \mathbb{P}_\theta} \left[ g(X, \theta) \right] \right) \right] \tag{1}$$

$$= \int_\Theta f \left( \underbrace{\int_\mathcal{X} g(x, \theta) p_\theta(x) dx}_{\text{inner conditional expectation}} \right) q(\theta) d\theta.$$

$$\underbrace{\phantom{= \int_\Theta f \left( \int_\mathcal{X} g(x, \theta) p_\theta(x) dx \right) q(\theta) d\theta}}_{\text{outer expectation}}$$

Nested expectations arise within a wide range of tasks, such as the computation of objectives in Bayesian experimental design (Beck et al., 2020; Goda et al., 2020; Rainforth

---

[1]Department of Computer Science, University College London [2]Department of Statistical Science, University College London. Correspondence to: Zonghao Chen <zonghao.chen.22@ucl.ac.uk>.

*Proceedings of the 42nd International Conference on Machine Learning*, Vancouver, Canada. PMLR 267, 2025. Copyright 2025 by the author(s).

---

et al., 2024), of acquisition functions in active learning and Bayesian optimisation (Ginsbourger and Le Riche, 2010; Yang et al., 2024), of objectives in distributionally-robust optimisation (Shapiro et al., 2023; Bariletto and Ho, 2024; Dellaporta et al., 2024), and of statistical divergences (Song et al., 2020; Kanagawa et al., 2023). Computing nested expectations is also a key task beyond machine learning, including in fields ranging from value of information for decision making (Giles and Goda, 2019; Mala, 2024) to finance and insurance (Gordy and Juneja, 2010; Giles and Haji-Ali, 2019), manufacturing (Andradóttir and Glynn, 2016) and geology (Goda et al., 2018).

The estimation of nested expectations is particularly challenging since there are two levels of intractability: the inner conditional expectation, and the outer expectation, both of which must be approximated accurately in order to approximate the nested expectation $I$ accurately. The most widely used algorithm for this problem is *nested Monte Carlo (NMC)* (Lee and Glynn, 2003; Hong and Juneja, 2009; Rainforth et al., 2018). It approximates the inner and outer expectations using Monte Carlo estimators with $N$ and $T$ samples respectively. NMC is consistent under mild conditions, but has a relatively slow rate of convergence. Depending on the regularity of the problem, existing results indicate that we require either $\mathcal{O}(\Delta^{-3})$ or $\mathcal{O}(\Delta^{-4})$ evaluations of $g$ to obtain a root mean squared error smaller or equal to $\Delta$. This tends to be prohibitively expensive; for example, we would expect in the order of either 1 or 100 million observations to obtain an error of $\Delta = 0.01$. This is infeasible for many applications where obtaining samples or evaluating $g$ is expensive.

This issue has led to the development of a number of methods aiming to reduce the cost. Bartuska et al. (2023) proposed replacing the Monte Carlo estimators with quasi-Monte Carlo (QMC) (Dick et al., 2013). This algorithm, called *nested QMC (NQMC)*, requires only $\mathcal{O}(\Delta^{-2.5})$ function evaluations to obtain an error of size $\Delta$ (so that we only need in the order of $100,000$ observations for an error of $\Delta = 0.01$). However, NQMC requires strong regularity assumptions which may not hold in practice (a monotone second and third derivative for $f$). Separately, Bujok et al. (2015); Giles and Haji-Ali (2019); Giles and Goda (2019) proposed to use *multi-level Monte Carlo (MLMC)* and showed that this can further reduce the number of func-

tion evaluations to $\mathcal{O}(\Delta^{-2})$ (so that we only need in the order of $10,000$ observations for an error of $\Delta = 0.01$). The algorithm has relatively mild assumptions on $f$ and $g$, which makes it broadly applicable but sub-optimal for applications where $f$ and $g$ are smooth and where we might therefore expect further reductions in cost.

To fill this gap in the literature, we propose a novel algorithm called *nested kernel quadrature (NKQ)*, which is presented in Section 3. NKQ replaces the inner and outer MC estimators of NMC with kernel quadrature (KQ) estimators (Sommariva and Vianello, 2006). We show in Section 4 that NKQ requires only $\tilde{\mathcal{O}}(\Delta^{-\frac{d_{\mathcal{X}}}{s_{\mathcal{X}}} - \frac{d_{\Theta}}{s_{\Theta}}})$ function evaluations to guarantee an error smaller or equal to $\Delta$. Here $\tilde{\mathcal{O}}$ denotes $\mathcal{O}$ up to logarithmic terms, $s_{\mathcal{X}}, s_{\Theta}$ are constants relating to the smoothness of $f$ and $g$ in $\mathcal{X}$ and $\Theta$, and we have $s_{\mathcal{X}} > d_{\mathcal{X}}/2$ and $s_{\Theta} > d_{\Theta}/2$. In the least favorable case, we therefore recover the $\mathcal{O}(\Delta^{-4})$ of NMC, but when the integrand is smooth and the dimension is not too large, we are able to have a cost which scales better than $\mathcal{O}(\Delta^{-2})$ and the method significantly outperforms all competitors. In those cases, we may only need in the order of a few hundred or thousands observations for an error of $\Delta = 0.01$. This fast rate is demonstrated numerically in Section 5, where we show that NKQ can provide significant accuracy gains in problems from Bayesian optimisation to option pricing and health economics. Moreover, we show that NKQ can be combined with QMC and MLMC, providing an avenue to further accelerate convergence.

## 2. Background

**Notation** Let $\mathbb{N}_+$ denote the positive integers and $\mathbb{N} = \mathbb{N}_+ \cup \{0\}$. For $h : \mathcal{X} \subseteq \mathbb{R}^d \to \mathbb{R}$, $x_{1:N}$ and $h(x_{1:N})$ are vectorized notation for $[x_1, \ldots, x_N]^\top \in \mathbb{R}^{N \times d}$ and $[h(x_1), \ldots, h(x_N)]^\top \in \mathbb{R}^{N \times 1}$ respectively. For a vector $a = [a_1, \ldots, a_d]^\top \in \mathbb{R}^d$, define $\|a\|_b = (\sum_{i=1}^d a_i^b)^{1/b}$. For a distribution $\pi$ supported on $\mathcal{X}$ and $0 < p \le \infty$, $L_p(\pi)$ is the space of functions $h : \mathcal{X} \to \mathbb{R}$ such that $\|h\|_{L_p(\pi)} := \mathbb{E}_{X \sim \pi}[|h(X)|^p] < \infty$ and $L_\infty(\pi)$ is the space of functions that are bounded $\pi$-almost everywhere. When $\pi$ is the Lebesgue measure $\mathcal{L}_{\mathcal{X}}$ over $\mathcal{X}$, we write $L_p(\mathcal{X}) := L_p(\mathcal{L}_{\mathcal{X}})$. For $\beta \in \mathbb{N}$, $C^\beta(\mathcal{X})$ denotes the space of functions whose partial derivatives of up to and including order $\beta$ are continuous. For two positive sequences $\{a_n\}_{n \in \mathbb{N}_+}$ and $\{b_n\}_{n \in \mathbb{N}_+}$, $a_n \asymp b_n$ means that $\lim_{n \to \infty} \frac{a_n}{b_n}$ is a positive constant, $a_n = \mathcal{O}(b_n)$ means that $\lim_{n \to \infty} \frac{a_n}{b_n} < \infty$ and $a_n = \tilde{\mathcal{O}}(b_n)$ means that $a_n = \mathcal{O}(b_n (\log b_n)^r)$ for some positive constant $r$.

**Existing Methods for Nested Expectations** Standard Monte Carlo (MC) is an estimator which can be used to approximate expectations/integrals through samples (Robert and Casella, 2000). Given an arbitrary function $h : \mathcal{X} \to \mathbb{R}$

with $h \in L_1(\pi)$, and $N$ independent and identically distributed (i.i.d.) realisations $x_{1:N}$ from $\pi$, standard MC approximates the expectation of $h$ under $\pi$ as follows:

$$\mathbb{E}_{X \sim \pi}[h(X)] \approx \frac{1}{N} \sum_{n=1}^N h(x_n).$$

For the nested expectation $I$ in (1), the use of a MC estimator for both the inner and outer expectation leads to the *nested Monte Carlo (NMC)* estimator (Hong and Juneja, 2009; Rainforth et al., 2018) given by

$$\hat{I}_{\text{NMC}} := \frac{1}{T} \sum_{t=1}^T f\left( \frac{1}{N} \sum_{n=1}^N g(x_n^{(t)}, \theta_t) \right), \qquad (2)$$

where $\theta_{1:T}$ are $T$ i.i.d. realisations from $\mathbb{Q}$ and $x_{1:N}^{(t)}$ are $N$ i.i.d. realisations from $\mathbb{P}_{\theta_t}$ for each $t \in \{1, \ldots, T\}$. The root mean-squared error of this estimator goes to zero at rate $\mathcal{O}(N^{-\frac{1}{2}} + T^{-\frac{1}{2}})$ when $f$ is Lipschitz continuous (Rainforth et al., 2018). Hence, taking $N = T = \mathcal{O}(\Delta^{-2})$ leads to an algorithm which requires $N \times T = \mathcal{O}(\Delta^{-4})$ function evaluations to obtain error smaller or equal to $\Delta$. When $f$ has bounded second order derivatives, the root mean-squared error converges at the improved rate of $\mathcal{O}(N^{-1} + T^{-\frac{1}{2}})$ (Rainforth et al., 2018). Taking $N = \sqrt{T} = \mathcal{O}(\Delta^{-1})$ therefore leads to an algorithm requiring $N \times T = \mathcal{O}(\Delta^{-3})$ function evaluations to get an error of $\Delta$ (Gordy and Juneja, 2010; Rainforth et al., 2018). Despite its simplicity, NMC therefore requires a large number of evaluations to reach a given $\Delta$.

As a result, two extensions have been proposed. Firstly, Bartuska et al. (2023) proposed to use (2), but to replace the i.i.d. samples with QMC points. QMC points are points which aim to fill $\mathcal{X}$ in a somewhat uniform fashion (Dick et al., 2013), with well-known examples including Sobol or Halton sequences. Bartuska et al. (2023) used randomized QMC points, which removes the bias of standard QMC by using a randomized low discrepancy sequence (Owen, 2003). For nested expectations, they showed that nesting randomized QMC estimators can lead to a faster convergence rate and hence a smaller cost of $\mathcal{O}(\Delta^{-2.5})$. However, the approach is only applicable when $\mathbb{P}_\theta$ and $\mathbb{Q}$ are Lebesgue measures on unit cubes (or smooth transformations thereof), and the rate only holds when $f$ has monotone second and third order derivatives.

Alternatively, Bujok et al. (2015); Giles (2015); Giles and Haji-Ali (2019); Giles and Goda (2019) proposed to use *multi-level Monte Carlo* (MLMC), which decomposes the nested expectation using a telescoping sum on the outer integral, then approximates each term with MC. The integrand with the $\ell$'th fidelity level is constructed as the composition of $f$ with an inner MC estimator based on $N_\ell$ samples. More precisely, the MLMC treatment of nested expectations consist of using:

$$\hat{I}_{\text{MLMC}} := \sum_{l=1}^{L} \frac{1}{T_\ell} \sum_{t=1}^{T_\ell} (f(J_{\ell,t}) - f(J_{\ell-1,t})) + \frac{1}{T_0} \sum_{t=1}^{T_0} f(J_{0,t})$$

where $J_{\ell,t} := \frac{1}{N_\ell} \sum_{n=1}^{N_\ell} g(x_n^{(t)}, \theta_t)$ for $\ell \in \{0, \dots, L\}$, (3)

Under some regularity conditions, Theorem 1 from Giles (2015) shows that taking $N_\ell = \mathcal{O}(2^\ell)$ and $T_\ell = \mathcal{O}(2^{-2\ell}\Delta^{-2})$ leads to an estimator requiring $\mathcal{O}(\Delta^{-2})$ function evaluations to obtain root mean squared error smaller or equal to $\Delta$. Although MLMC has the best known efficiency for nested expectations, $N_l$ and $T_l$ need to grow exponentially with $l$, and we therefore need a very large sample size for its theoretical convergence rate to become evident in practice (Giles and Haji-Ali, 2019; Giles and Goda, 2019). MLMC also requires making several challenging design choices, including the coarsest level to use, and the number of samples per level. Most importantly, MLMC as well as all existing methods fail to account for the smoothness of the functions $f$ and $g$.

**Kernel Quadrature** *Kernel quadrature (KQ)* (Sommariva and Vianello, 2006; Rasmussen and Ghahramani, 2002; Briol et al., 2019) provides an alternative to standard MC for (non-nested) expectations. Consider an arbitrary function $h : \mathcal{X} \to \mathbb{R}$ and distribution $\pi$ on $\mathcal{X}$, and suppose we would like to approximate $\mathbb{E}_{X \sim \pi}[h(X)]$. KQ is an estimator which can be used when $h$ is sufficiently regular, in the sense that it belongs to a reproducing kernel Hilbert space (RKHS) (Berlinet and Thomas-Agnan, 2004) $\mathcal{H}_k$ with kernel $k$. We recall that for a positive semi-definite kernel $k : \mathcal{X} \times \mathcal{X} \to \mathbb{R}$, the RKHS $\mathcal{H}_k$ is a Hilbert space with inner product $\langle \cdot, \cdot \rangle_{\mathcal{H}_k}$ and norm $\| \cdot \|_{\mathcal{H}_k}$ (Aronszajn, 1950) such that: (i) $k(x, \cdot) \in \mathcal{H}_k$ for all $x \in \mathcal{X}$, and (ii) the reproducing property holds, i.e. for all $h \in \mathcal{H}_k$, $x \in \mathcal{X}$, $h(x) = \langle h, k(x, \cdot) \rangle_{\mathcal{H}_k}$. An important example of RKHS is the Sobolev space $W_2^s(\mathcal{X})$ ($s > \frac{d}{2}$), which consists of functions of certain smoothness encoded through the square integrability of their weak partial derivatives up to order $s$,

$$W_2^s(\mathcal{X}) := \{ h \in L_2(\mathcal{X}) : D^\beta h \in L_2(\mathcal{X})$$
$$\text{for all } \beta \in \mathbb{N}^d \text{ with } |\beta| \le s \}, \quad s \in \mathbb{N}_+ \quad (4)$$

where $D^\beta f$ denotes the $\beta$-th (weak) partial derivative of $f$. Assuming $h \in \mathcal{H}_k$ and $\mathbb{E}_{X \sim \pi}[\sqrt{k(X, X)}] < \infty$, the KQ estimator $\hat{I}_{\text{KQ}} = \sum_{n=1}^{N} w_n h(x_n)$ uses weights obtained by minimizing an upper bound on the absolute error:

$$\left| I - \hat{I}_{\text{KQ}} \right| = \left| \mathbb{E}_{X \sim \pi}[h(X)] - \sum_{n=1}^{N} w_n h(x_n) \right|$$

$$\le \| h \|_{\mathcal{H}_k} \left\| \mu_\pi(X) - \sum_{n=1}^{N} w_n k(x_n, \cdot) \right\|_{\mathcal{H}_k},$$

where $\mu_\pi(\cdot) = \mathbb{E}_{X \sim \pi}[k(X, \cdot)]$ is the *kernel mean embedding (KME)* of $\pi$ in the RKHS $\mathcal{H}_k$ (Smola et al., 2007). Minimizing the right hand side with an additive regulariser term $\lambda \|f\|_{\mathcal{H}_k}$ over the choice of weights leads to the following KQ estimator:

$$\hat{I}_{\text{KQ}} := \mu_\pi(x_{1:N}) (\boldsymbol{K} + N\lambda \boldsymbol{I}_N)^{-1} h(x_{1:N}), \quad (5)$$

where $\boldsymbol{I}_N$ is the $N \times N$ identity matrix, $\boldsymbol{K} = k(x_{1:N}, x_{1:N}) \in \mathbb{R}^{N \times N}$ is the Gram matrix and $\lambda \ge 0$ is a regularisation parameter ensuring the matrix is numerically invertible. The KQ weights are given by $w_{1:N} = \mu_\pi(x_{1:N})(\boldsymbol{K} + N\lambda \boldsymbol{I}_N)^{-1}$ and are optimal when $\lambda = 0$.

KQ takes into account the structural information that $h \in \mathcal{H}_k$ so the absolute error $|I - \hat{I}_{\text{KQ}}|$ goes to $0$ at a fast rate as $N \to \infty$. Specifically, when the RKHS $\mathcal{H}_k$ is norm-equivalent to the Sobolev space $W_2^s(\mathcal{X})$ ($s > \frac{d}{2}$), KQ achieves the rate $\mathcal{O}(N^{-\frac{s}{d}})$ (Kanagawa and Hennig, 2019; Kanagawa et al., 2020). This is known to be minimax optimal (Novak, 2006; 2016), and significantly faster than the $\mathcal{O}(N^{-\frac{1}{2}})$ rate of standard MC. Interestingly, existing proof techniques that obtain this rate take $\lambda = 0$ in (5) and require the Gram matrix $\boldsymbol{K}$ to be invertible, whilst the new proof technique based on kernel ridge regression in this paper obtains the same optimal rate while allowing a positive regularization $\lambda \asymp N^{-\frac{2s}{d}} (\log N)^{\frac{2s+2}{d}}$, which improves numerical stability when inverting $\boldsymbol{K}$. (See Remark B.2)

Despite the optimality of the KQ convergence rate, the rate constant can be reduced by selecting points $x_{1:N}$ other than through i.i.d. sampling. Strategies include importance sampling (Bach, 2017; Briol et al., 2017), QMC point sets (Briol et al., 2019; R. Jagadeeswaran, 2019; Bharti et al., 2023; Kaarnioja et al., 2025), realisations from determinental point processes (Belhadji et al., 2019), point sets with symmetry properties (Karvonen and Särkkä, 2018; Karvonen et al., 2019) and adaptive designs (Osborne et al., 2012; Gunter et al., 2014; Briol et al., 2015; Gessner et al., 2020). Most relevant to our work is the combination of KQ with MLMC to improve accuracy in multifidelity settings (Li et al., 2023).

Two main drawbacks of KQ compared to MC are the worst-case computational cost of $\mathcal{O}(N^3)$ (due to computation of the inverse of the Gram matrix), and the need for a closed-form expression of the KME $\mu_\pi$. Fortunately, numerous approaches can mitigate these drawbacks. To reduce the cost, one can use geometric properties of the point set (Karvonen and Särkkä, 2018; Karvonen et al., 2019; Kuo et al., 2024), Nyström approximations (Hayakawa et al., 2022; 2023), randomly pivoted Cholesky (Epperly and Moreno, 2023), or the fast Fourier transform (Zeng et al., 2009). To obtain a closed-form KME, KQ users typically refer to existing derivations (see Table 1 in Briol et al. (2019) or Wenger et al. (2021)), or use Stein reproducing kernels

(Oates et al., 2017; 2019; Si et al., 2021; Sun et al., 2023).

In this paper, we tackle both drawbacks through a change of variable trick. Suppose we can find a continuous transformation map $\Phi$ such that $x_{1:N} = \Phi(u_{1:N})$ where $u_{1:N}$ are samples from a simpler distribution $\mathbb{U}$ of our choice. A direct application of change of variables theorem (Section 8.2 of Stirzaker (2003)) proves that $\mathbb{E}_{X \sim \pi} h(X) = \int_{\mathcal{U}} h(\Phi(u))d\mathbb{U}(u)$, so the integrand changes from $h : \mathcal{X} \to \mathbb{R}$ to $h \circ \Phi : \mathcal{U} \to \mathbb{R}$ and the kernel quadrature estimator becomes $\hat{I}_{\mathrm{KQ}} = \mu_{\mathbb{U}}(u_{1:N}) \left( \boldsymbol{K}_{\mathcal{U}} + N\lambda \boldsymbol{I}_N \right)^{-1} (h \circ \Phi)(u_{1:N})$, where $\boldsymbol{K}_{\mathcal{U}} = k_{\mathcal{U}}(u_{1:N}, u_{1:N})$. The measure $\mathbb{U}$ is typically chosen such that the KME is known in closed-form, and the KQ weights $\mu_{\mathbb{U}}(u_{1:N}) \left( \boldsymbol{K}_{\mathcal{U}} + N\lambda \boldsymbol{I}_N \right)^{-1}$ can be precomputed and stored so that KQ becomes a weighted average of function evaluations with $\mathcal{O}(N)$ computational complexity. The main technical challenge of using the change of variable trick is to find such transform map $\Phi$. See Appendix F.1 for further details.

Before concluding, we note that the KQ estimator is often called *Bayesian quadrature (BQ)* (Diaconis, 1988; O'Hagan, 1991; Rasmussen and Ghahramani, 2002; Briol et al., 2019; Hennig et al., 2022) since it can be derived as the mean of the pushforward of a Gaussian measure on $h$ conditioned on $h(x_{1:N})$ (Kanagawa et al., 2018). The advantage of the Bayesian interpretation is that it provides finite-sample uncertainty quantification, and it also allows for efficient hyperparameter selection via empirical Bayes.

## 3. Nested Kernel Quadrature

We can now present our novel algorithm: *nested kernel quadrature (NKQ)*. To simplify the formulas, we write

$$J(\theta) := \mathbb{E}_{X \sim \mathbb{P}_\theta} [g(X, \theta)], \quad F(\theta) := f(J(\theta)), \quad (6)$$

so that the nested expectation in (1) can be written as $I = \mathbb{E}_{\theta \sim \mathbb{Q}}[F(\theta)]$. We will assume that we have access to

$$\theta_{1:T} := [\theta_1, \ldots, \theta_T]^\top \in \Theta^T,$$
$$x_{1:N}^{(t)} := \left[ x_1^{(t)}, \ldots, x_N^{(t)} \right] \in \mathcal{X}^N,$$
$$g\left(x_{1:N}^{(t)}, \theta_t\right) := \left[ g(x_1^{(t)}, \theta_t), \ldots, g(x_N^{(t)}, \theta_t) \right] \in \mathbb{R}^N,$$

for all $t \in \{1, \ldots, T\}$, and $f$ is a function that can be evaluated. We do not specify how the point sets are generated, although further (mild) assumptions will be imposed for our theory in Section 4. Using the same number of function evaluations $N$ per $\theta_t$ is not essential, but we assume this as it significantly simplifies our notation. Given the above, we are now ready to define NKQ as the following two-stage algorithm, which is illustrated in Figure 1.

**Stage I** For each $t \in \{1, \ldots, T\}$, we estimate the inner conditional expectation $J$ evaluated at $\theta_t$ with $N$ observa-

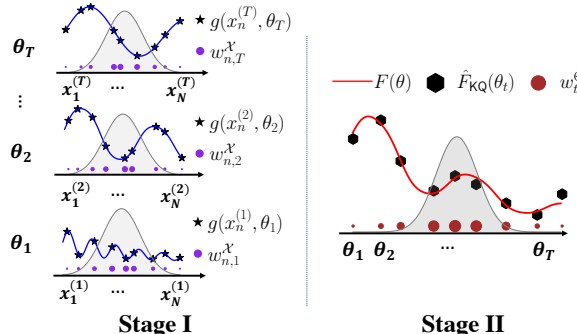

**Figure 1.** *Illustration of NKQ.* In stage I, we estimate $J(\theta_t)$ using $\hat{J}_{\mathrm{KQ}}(\theta_t) = \sum_{n=1}^N w_{n,t}^{\mathcal{X}} g(x_n^{(t)}, \theta_t)$ for all $t \in \{1, \ldots, T\}$. In stage II, we estimate $I$ with $\hat{I}_{\mathrm{NKQ}} = \sum_{t=1}^T w_t^\Theta \hat{F}_{\mathrm{KQ}}(\theta_t)$ where $\hat{F}_{\mathrm{KQ}}(\theta_t) := f(\hat{J}_{\mathrm{KQ}}(\theta_t))$. The shaded areas depict $\mathbb{P}_\theta$ (for stage I) and $\mathbb{Q}$ (for stage II).

tions $x_{1:N}^{(t)}$ and $g(x_{1:N}^{(t)}, \theta_t)$ using a KQ estimator:

$$\hat{J}_{\mathrm{KQ}}(\theta_t) := \mu_{\mathbb{P}_{\theta_t}} \left( x_{1:N}^{(t)} \right) \left( \boldsymbol{K}_{\mathcal{X}}^{(t)} + N\lambda_{\mathcal{X}} \boldsymbol{I}_N \right)^{-1} g\left( x_{1:N}^{(t)}, \theta_t \right). \quad (7)$$

Here $k_{\mathcal{X}}$ is a reproducing kernel on $\mathcal{X}$, $\mu_{\mathbb{P}_{\theta_t}}(\cdot) = \mathbb{E}_{X \sim \mathbb{P}_{\theta_t}}[k_{\mathcal{X}}(X, \cdot)]$ is the KME of $\mathbb{P}_{\theta_t}$ and $\boldsymbol{K}_{\mathcal{X}}^{(t)} = k_{\mathcal{X}}(x_{1:N}^{(t)}, x_{1:N}^{(t)})$ is an $N \times N$ Gram matrix. Using the same kernel $k_{\mathcal{X}}$ for each $t \in \{1, \ldots, T\}$ is not essential, but we assume this to be the case for simplicity. Given these KQ estimates, we then we apply the function $f$ to get $\hat{F}_{\mathrm{KQ}}(\theta_t) = f(\hat{J}_{\mathrm{KQ}}(\theta_t))$.

**Stage II** We use a KQ estimator to approximate the outer expectation using the output of Stage I:

$$\hat{I}_{\mathrm{NKQ}} := \mu_{\mathbb{Q}}(\theta_{1:T})(\boldsymbol{K}_\Theta + T\lambda_\Theta \boldsymbol{I}_T)^{-1} \hat{F}_{\mathrm{KQ}}(\theta_{1:T}). \quad (8)$$

Here $k_\Theta$ is a reproducing kernel on $\Theta$, $\mu_{\mathbb{Q}} = \mathbb{E}_{\theta \sim \mathbb{Q}}[k_\Theta(\theta, \cdot)]$ is the embedding of $\mathbb{Q}$ and $\boldsymbol{K}_\Theta = k_\Theta(\theta_{1:T}, \theta_{1:T})$ is a $T \times T$ Gram matrix.

Combining stage I and II, NKQ can be expressed in a single equation as a nesting of two quadrature rules:

$$\hat{I}_{\mathrm{NKQ}} = \sum_{t=1}^T w_t^\Theta f \left( \sum_{n=1}^N w_{n,t}^{\mathcal{X}} g(x_n^{(t)}, \theta_t) \right), \quad (9)$$

where $w_{1,t}^{\mathcal{X}}, \ldots, w_{N,t}^{\mathcal{X}}$ are the KQ weights used in stage I for $\hat{J}_{\mathrm{KQ}}(\theta_t)$ and $w_1^\Theta, \ldots, w_T^\Theta$ are the KQ weights used in stage II. Although these weights are stage-wise optimal when $\lambda_{\mathcal{X}} = \lambda_\Theta = 0$ thanks to the optimality of KQ weights, it is unclear whether they are globally optimal due to the non-linearity of $f$. Note that NMC can be recovered by taking all stage I weights to be $1/N$ and all stage II weights to be $1/T$, which is sub-optimal. In addition to the algorithmic simplicity of our proposed estimator NKQ, we demonstrate its superior performance in terms of both rate of convergence (Section 4) and empirical performances (Section 5).

NKQ inherits the two main drawbacks of KQ. Firstly, solving the linear systems to obtain the stage I and II weights has a worst-case computational complexity of $\mathcal{O}(TN^3 + T^3)$. Secondly, NKQ requires closed-form KMEs at both stages: $\mu_{\mathbb{P}_{\theta_t}}$ for all $t \in \{1, \ldots, T\}$ in stage I, and $\mu_{\mathbb{Q}}$ in stage II. Fortunately, we can often use the approaches discussed in the previous section to reduce the complexity to $\mathcal{O}(TN + T)$ and obtain closed-form kernel embeddings.

NKQ requires the selection of hyperparameters, including for the kernels in both stage I and II. We typically take $k_{\mathcal{X}}$ and $k_{\Theta}$ to be Matérn kernels whose orders are determined by the smoothness of $f$ and $g$ (as justified by Theorem 1; see Section 4 for details). This leaves us with a choice of kernel hyperparameters which include lengthscales $\gamma_{\mathcal{X}}, \gamma_{\Theta} > 0$ and amplitudes $A_{\mathcal{X}}, A_{\Theta} > 0$. The lengthscales are selected via the median heuristic. The regularizers are set to $\lambda_{\mathcal{X}} = \lambda_{0,\mathcal{X}} N^{-\frac{2s_{\mathcal{X}}}{d_{\mathcal{X}}}} (\log N)^{\frac{2s_{\mathcal{X}}+2}{d_{\mathcal{X}}}}$ and $\lambda_{\Theta} = \lambda_{0,\Theta} T^{-\frac{2s_{\Theta}}{d_{\Theta}}} (\log T)^{\frac{2s_{\Theta}+2}{d_{\Theta}}}$ following Theorem 1, where $\lambda_{0,\mathcal{X}}, \lambda_{0,\Theta}$ are selected with grid search over $\{0.01, 0.1, 1.0\}$. Finally, we standardise our function values (by subtracting the empirical mean then dividing by the empirical standard deviation), and then set the amplitudes to $A_{\mathcal{X}} = A_{\Theta} = 1$. This last choice could further be improved using a grid search, but we do not do this as we do not notice significant improvements when doing so in experiments and this tends to increase the cost.

Before presenting our theoretical results, we briefly comment on the connection with existing KQ methods. If we could evaluate the exact expression for the inner conditional expectation $J(\theta)$ pointwise, then (following (5)) the KQ estimator for $I$ would be $\bar{I}_{\mathrm{KQ}} = \mu_{\mathbb{Q}}(\theta_{1:T})(\boldsymbol{K}_{\Theta} + T\lambda_{\Theta}\boldsymbol{I}_T)^{-1}F(\theta_{1:T})$. Comparing with (8), NKQ can thus be seen as KQ with noisy function values $\hat{F}_{\mathrm{KQ}}(\theta_{1:T})$ (replacing the exact values $F(\theta_{1:T})$ in (8)). Although it is proved in Cai et al. (2023) that noisy observations make KQ converge at a slower rate, we prove that the stage II observation noise is of the same order as the stage I error, and consequently, we can still treat stage II KQ as noiseless kernel ridge regression and the additional error caused by the stage II observation noise would be *subsumed* by the stage I error (See Remark 4.1). NKQ is also closely related to a family of regression-based methods for estimating conditional expectations (Longstaff and Schwartz, 2001; Chen et al., 2024b). Indeed, with a slight modification of Stage II in (8), we can obtain an estimator of $J(\theta)$ that we call *conditional kernel quadrature (CKQ)*

$$\hat{J}_{\mathrm{CKQ}}(\theta) := k_{\Theta}(\theta, \theta_{1:T})(\boldsymbol{K}_{\Theta} + T\lambda_{\Theta}\boldsymbol{I}_T)^{-1}\hat{J}_{\mathrm{KQ}}(\theta_{1:T}). \tag{10}$$

CKQ highly resembles *conditional BQ (CBQ)* (Chen et al., 2024b); the difference is in stage II, where CBQ uses heteroskedastic Gaussian process regression whilst CKQ uses kernel ridge regression. Interestingly, the proof in this paper leads to a much better rate for CKQ than the best known rate for CBQ (see Remark 4.2).

## 4. Theoretical Results

In this section, we derive a convergence rate for the absolute error $|\hat{I}_{\mathrm{NKQ}} - I|$ as the number of samples $N, T \to \infty$. Before doing so, we recall the connection between RKHSs and Sobolev spaces. A kernel $k$ on $\mathbb{R}^d$ is said to be translation invariant if $k(x, x') = \Psi(x - x')$ for some positive definite function $\Psi$ whose Fourier transform $\hat{\Psi}(\omega)$ is a finite non-negative measure on $\mathbb{R}^d$ (Wendland, 2004, Theorem 6.6). Suppose $\mathcal{X}$ has a Lipschitz boundary, if $k$ is translation invariant and its Fourier transform $\hat{\Psi}(\omega)$ decays as $\mathcal{O}(1 + \|\omega\|_2^2)^{-s}$ when $\omega \to \infty$ for $s > d/2$, then its RKHS $\mathcal{H}_k$ is norm equivalent to the Sobolev space $W_2^s(\mathcal{X})$ (Wendland, 2004, Corollary 10.48). More specifically, it means that their set of functions coincide and there are constants $c_1, c_2 > 0$ such that $c_1\|h\|_{\mathcal{H}_k} \leq \|h\|_{W_2^s(\mathcal{X})} \leq c_2\|h\|_{\mathcal{H}_k}$ holds for all $h \in \mathcal{H}_k$. In this paper, we call such kernel a *Sobolev reproducing kernel of smoothness $s$*. An important example of Sobolev kernel is the Matérn kernel— the RKHS of a Matérn-$\nu$ kernel is norm-equivalent to $W_2^s(\mathcal{X})$ with $s = \nu + d/2$. All Sobolev kernels are bounded, i.e. $\sup_{x \in \mathcal{X}} k(x, x) \leq \kappa$ for some positive constant $\kappa$. When the context is clear, we use $\|f\|_{s,2} := \|f\|_{W_2^s(\mathcal{X})}$ to denote the Sobolev space norm.

**Theorem 1.** *Let $\mathcal{X} = [0,1]^{d_{\mathcal{X}}}$ and $\Theta = [0,1]^{d_{\Theta}}$. Suppose $\theta_{1:T}$ are i.i.d. samples from $\mathbb{Q}$ and $x_{1:N}^{(t)}$ are i.i.d samples from $\mathbb{P}_{\theta_t}$ for all $t \in \{1, \cdots, T\}$. Suppose further that $k_{\mathcal{X}}$ and $k_{\Theta}$ are Sobolev kernels of smoothness $s_{\mathcal{X}} > d_{\mathcal{X}}/2$ and $s_{\Theta} > d_{\Theta}/2$, and that the following conditions hold*

*(1) There exist $G_{0,\Theta}, G_{1,\Theta}, G_{0,\mathcal{X}}, G_{1,\mathcal{X}} > 0$ such that $G_{0,\Theta} \leq q(\theta) \leq G_{1,\Theta}$ and $G_{0,\mathcal{X}} \leq p_{\theta}(x) \leq G_{1,\mathcal{X}}$ for any $\theta \in \Theta$ and $x \in \mathcal{X}$.*

*(2) There exists $S_1 > 0$ such that for any $\theta \in \Theta$ and any $\beta \in \mathbb{N}^{d_{\Theta}}$ with $|\beta| \leq s_{\Theta}$, $\|D_{\theta}^{\beta}g(\cdot, \theta)\|_{s_{\mathcal{X}},2} \leq S_1$.*

*(3) There exist $S_2, S_3 > 0$ such that for any $x \in \mathcal{X}$, $\|g(x, \cdot)\|_{s_{\Theta},2} \leq S_2$ and $\|\theta \mapsto p_{\theta}(x)\|_{s_{\Theta},2} \leq S_3 \leq 1$.*

*(4) There exists $S_4 > 0$ such that derivatives of $f$ up to and including order $s_{\Theta} + 1$ are bounded by $S_4$.*

*Then, there exists $N_0, T_0 \in \mathbb{N}^+$ such that for $N > N_0, T > T_0$, we can take $\lambda_{\mathcal{X}} \asymp N^{-2\frac{s_{\mathcal{X}}}{d_{\mathcal{X}}}} (\log N)^{\frac{2s_{\mathcal{X}}+2}{d_{\mathcal{X}}}}$ and $\lambda_{\Theta} \asymp T^{-2\frac{s_{\Theta}}{d_{\Theta}}} (\log T)^{\frac{2s_{\Theta}+2}{d_{\Theta}}}$ to obtain the following bound*

$$\left| I - \hat{I}_{NKQ} \right|$$
$$\leq \tau \left( C_1 N^{-\frac{s_{\mathcal{X}}}{d_{\mathcal{X}}}} (\log N)^{\frac{s_{\mathcal{X}}+1}{d_{\mathcal{X}}}} + C_2 T^{-\frac{s_{\Theta}}{d_{\Theta}}} (\log T)^{\frac{s_{\Theta}+1}{d_{\Theta}}} \right),$$

*which holds with probability at least $1 - 8e^{-\tau}$. $C_1, C_2$ are*

*two constants independent of $N, T, \tau$.*

**Corollary 1.** *Suppose all assumptions in Theorem 1 hold. If we set $N = \tilde{\mathcal{O}}(\Delta^{-\frac{d_\mathcal{X}}{s_\mathcal{X}}})$ and $T = \tilde{\mathcal{O}}(\Delta^{-\frac{d_\Theta}{s_\Theta}})$, then $N \times T = \tilde{\mathcal{O}}(\Delta^{-\frac{d_\mathcal{X}}{s_\mathcal{X}} - \frac{d_\Theta}{s_\Theta}})$ samples are sufficient to guarantee that $|I - \hat{I}_{NKQ}| \leq \Delta$ holds with high probability.*

To prove these results, we can decompose $|I - \hat{I}_{\mathrm{NKQ}}|$ into the sum of stage I and stage II errors, which can be bounded by terms of order $N^{-\frac{s_\mathcal{X}}{d_\mathcal{X}}} (\log N)^{\frac{s_\mathcal{X}+1}{d_\mathcal{X}}}$ and $T^{-\frac{s_\Theta}{d_\Theta}} (\log T)^{\frac{s_\Theta+1}{d_\Theta}}$ respectively; see Appendix C. Interestingly, note that the stage II error does not suffer from the fact that we are using noisy observations $\hat{F}_{\mathrm{KQ}}(\theta_{1:T})$ and we maintain the standard KQ rate up to logarithm terms (see Remark 4.1). We emphasize that our bound indicates that the tail of $|I - \hat{I}_{\mathrm{NKQ}}|$ is sub-exponential. This contrasts with existing work on Monte Carlo methods, which only provides upper bounds on the expectation of error with no constraints on its tails (Giles, 2015; Bartuska et al., 2023).

We now briefly discuss our assumptions. Assumption (1) is mild and allows $L_2(\mathbb{P}_\theta)$ (resp. $L_2(\mathbb{Q})$) to be norm equivalent to $L_2(\mathcal{X})$ (resp. $L_2(\Theta)$), which is widely used in statistical learning theory that involves Sobolev spaces (Fischer and Steinwart, 2020; Suzuki and Nitanda, 2021). Since our proof essentially translates quadrature error into generalization error of kernel ridge regression, Assumptions (2), (3), (4) ensure that functions $f, g$ and the density $p$ have enough regularity so that the regression targets in both stage I and stage II belong to the correct Sobolev spaces. These are more restrictive, but are essential to obtain our fast rate and are common assumptions in the KQ literature. Assumptions (2), (3), (4) can be relaxed if mis-specification is allowed; see e.g. Fischer and Steinwart (2020); Kanagawa et al. (2023); Zhang et al. (2023). Theorem 1 shows that for NKQ to have a fast convergence rate, one ought to use Sobolev kernels which are as smooth as possible in both stages. Furthermore, when $s_\mathcal{X} = s_\Theta = \infty$ (e.g. when the integrand and kernels belong to Gaussian RKHSs), our proof could be modified to show an exponential rate of convergence in a similar fashion as Briol (2018, Theorem 10) or Karvonen et al. (2020).

**Remark 4.1** (Noisy observations in Stage II of NKQ). *Note that NKQ employs noisy observations $\{\theta_t, \hat{F}_{KQ}(\theta_t)\}_{t=1}^T$ in stage II KQ rather than the ground truth observations $\{\theta_t, F(\theta_t)\}_{t=1}^T$. Although Cai et al. (2023) establishes that KQ with noisy observations converges at a slower rate than KQ with noiseless observations, a key distinction in our setting is that, as shown in (C.30), the observation noise in stage II KQ is of order $\tilde{\mathcal{O}}(N^{-\frac{s_\mathcal{X}}{d_\mathcal{X}}})$, whereas the noise in Cai et al. (2023) remains at a constant level. As a result, we can still use KQ in stage II as if the observations $\{\hat{F}_{KQ}(\theta_t)\}_{t=1}^T$ are noiseless, and the additional error it introduces happens to be of the*

| Method | Cost |
|---|---|
| NMC | $\mathcal{O}(\Delta^{-3})$ or $\mathcal{O}(\Delta^{-4})$ |
| NQMC | $\mathcal{O}(\Delta^{-2.5})$ |
| MLMC | $\mathcal{O}(\Delta^{-2})$ |
| **NKQ (Corollary 1)** | $\tilde{\mathcal{O}}\left(\Delta^{-\frac{d_\mathcal{X}}{s_\mathcal{X}} - \frac{d_\Theta}{s_\Theta}}\right)$ |

*Table 1.* Cost of methods for nested expectations, measured through the number of function evaluations required to ensure $|I - \hat{I}| \leq \Delta$. NMC rate is taken from Theorem 3 of Rainforth et al. (2018), NQMC rate is taken from Proposition 4 of Bartuska et al. (2023), MLMC rate is taken from Section 3.1 of Giles (2018). Smaller exponents $r$ in $\Delta^{-r}$ indicate a cheaper method.

*same order as the stage I error $\tilde{\mathcal{O}}(N^{-\frac{s_\mathcal{X}}{d_\mathcal{X}}})$ and is therefore subsumed by it.*

**Remark 4.2** (Convergence rate for CKQ). *For the CKQ estimator $\hat{J}_{CKQ}$ (defined in (10)) that approximates the parametric / conditional expectation $J(\theta)$ uniformly over all $\theta \in \Theta$, the error can be upper bounded in the same way as NKQ. The stage I error and can be shown to be $\tilde{\mathcal{O}}(N^{-\frac{s_\mathcal{X}}{d_\mathcal{X}}})$ using the same analysis from (C.13) to (C.16); and the stage II error and can be shown to be $\tilde{\mathcal{O}}(T^{-\frac{s_\Theta}{d_\Theta}})$ using the same analysis from (C.18) to (C.34). Combining the two error terms, we have $\|J - \hat{J}_{CKQ}\|_{L_2(\mathbb{Q})} = \tilde{\mathcal{O}}(N^{-\frac{s_\mathcal{X}}{d_\mathcal{X}}} + T^{-\frac{s_\Theta}{d_\Theta}})$ holds with probability at least $1 - 8e^{-\tau}$. The rate is better than the best known rate $\mathcal{O}(N^{-\frac{s_\mathcal{X}}{d_\mathcal{X}}} + T^{-\frac{1}{4}})$ of CBQ proved in Theorem 1 of (Chen et al., 2024b) since $\frac{s_\Theta}{d_\Theta} > \frac{1}{2} > \frac{1}{4}$. The intuition behind the faster rate is that CKQ benefits from the extra flexibility of choose regularization parameters $\lambda_\mathcal{X}, \lambda_\Theta$; while CBQ, as a two stage Gaussian Process based approach, is limited to choose $\lambda_\Theta$ equal to the heteroskedastic noise from the first stage. It may be possible to modify the proof of (Chen et al., 2024b) to improve the rate further, but this has not been explored to date.*

In Table 1, we compare the *cost* of all methods evaluated by the number of evaluations required to ensure $|\hat{I} - I| \leq \Delta$. We can see that NKQ is the only method that explicitly exploits the smoothness of $g, p, f$ in the problem so that it outperforms all other methods when $\frac{d_\mathcal{X}}{s_\mathcal{X}} + \frac{d_\Theta}{s_\Theta} < 2$.

**Remark 4.3** (Combine NKQ with MLMC and QMC). *We have previously mentioned that KQ could potentially be combined with other algorithms to further improve efficiency, and studied this for both MLMC and QMC. For the former (i.e. NKQ+MLMC), we derived a new method called* multi-level NKQ (MLKQ), *which closely related to multilevel BQ algorithm of Li et al. (2023) and for which we were able to prove a rate of $\tilde{\mathcal{O}}(\Delta^{-1 - \frac{d_\mathcal{X}}{2s_\mathcal{X}} - \frac{d_\Theta}{2s_\Theta}})$. Similarly to NKQ, when $\frac{d_\mathcal{X}}{s_\mathcal{X}} + \frac{d_\Theta}{s_\Theta} < 2$, the rate for MLKQ*

is faster than that of NMC, NQMC and MLMC. However, the rate we managed to prove is slower than that for NKQ, and a slower convergence was also observed empirically (see *Figure 6*). We speculate that the worse performance is caused by the accumulation of bias from the KQ estimators at each level. See *Appendix D.2* for details.

We also consider combining NKQ and QMC. In this case, we expect the same rate as in *Theorem 1* can be recovered by resorting to the fill distance technique in scatter data approximation (*Wendland, 2004*). This is confirmed empirically in *Section 5*, where we observe that using QMC points can achieve similar or even better performance than NKQ with i.i.d. samples.

## 5. Experiments

We now illustrate NKQ over a range of applications, including some where the theory does not hold but where we still observe significant gains in accuracy. The code to reproduce all experiments is available at https://github.com/hudsonchen/nest_kq.

**Synthetic Experiment**  We start by verifying the bound in Theorem 1 using the following synthetic example: $\mathbb{Q} = U[0,1]$, $\mathbb{P}_\theta = U[0,1]$, $g(x,\theta) = x^{\frac{5}{2}} + \theta^{\frac{5}{2}}$, and $f(z) = z^2$, in which case $I = 0.4115$ can be computed analytically. We estimate $I$ with i.i.d. samples $\theta_{1:T} \sim U[0,1]$ and i.i.d. samples $x_{1:N}^{(t)} \sim U[0,1]$ for $t \in \{1, \ldots, T\}$. The assumptions from Theorem 1 are satisfied with $s_{\mathcal{X}} = s_\Theta = 2$ and $d_{\mathcal{X}} = d_\Theta = 1$ (see Appendix F.2). Therefore, to reach the absolute error threshold $\Delta$, we choose $N = T = \Delta^{-0.5}$ for NKQ following Corollary 1. On the other hand, based on Theorem 3 of Rainforth et al. (2018), the optimal way of assigning samples for NMC is to choose $N = \sqrt{T} = \Delta^{-1}$.

In Figure 2 **Top**, we see that the optimal choice of $N$ and $T$ suggested by the theory indeed results in a faster rate of convergence for both NMC and NKQ. For this synthetic problem, we confirm that both the theoretical rates of NKQ (Cost $= \Delta^{-1}$) and NMC (Cost $= \Delta^{-3}$) from Theorem 1 and Rainforth et al. (2018, Theorem 3) are indeed realized. We also adapt the synthetic problem to higher dimensions ($d_{\mathcal{X}} = d_\Theta = d$) in (F.42) and observe in Figure 2 **Bottom** that the performance gap between NKQ and NMC closes down as dimension grows. Such behaviour is expected because the cost of NKQ is $\tilde{\mathcal{O}}(\Delta^{-\frac{d_{\mathcal{X}}}{s_{\mathcal{X}}} - \frac{d_\Theta}{s_\Theta}})$ and therefore degrades as the dimensions $d_{\mathcal{X}}$ and $d_\Theta$ increase; whilst the cost of NMC remains the same.

We also conduct ablation studies, which are reserved for Figure 5 in the appendix. In the left-most plot, we see that the result are not too sensitive to $\lambda_0$, although very large values decrease accuracy whilst very small values cause numerical issues. In the middle plot, we see that selecting

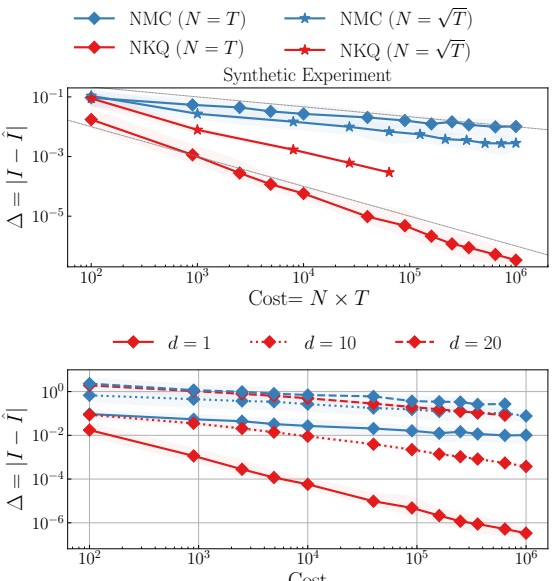

*Figure 2. Synthetic experiment.* **Top:** Verification of theoretical results. The thin grey lines are theoretical rates of $\Delta = \text{Cost}^{-1}$ and $\Delta = \text{Cost}^{-1/3}$. **Bottom:** Comparison of NKQ and NMC as dimension $d$ increases. Results are averaged over 1000 independent runs, while shaded regions give the 25%-75% quantiles.

the kernel lengthscale using the median heuristic provides very good performance. In the right-most plot, we see that NKQ with Matérn-$\frac{3}{2}$ kernels outperforms Matérn-$\frac{1}{2}$ kernel, indicating practitioners should use Sobolev kernels with the highest order of smoothness permissible by Theorem 1.

**Risk Management in Finance**  We now move beyond synthetic examples, starting in finance. Financial institutions often face the challenge of estimating the expected loss of their portfolios in the presence of potential economic shocks, which amounts to numerically solving stochastic differential equations (SDEs) over long time horizons (Achdou and Pironneau, 2005). Given the high cost of such simulations, data-efficient methods like NKQ are particularly desirable.

Suppose a shock occurs at time $\eta$ and alters the price of an asset by a factor of $1 + s$ for some $s \geq 0$. Conditioned on the asset price $S(\eta) = \theta$ at the time of shock, the loss of an option associated with that asset at maturity $\zeta$ with price $S(\zeta) = x$ can be expressed as $J(\theta) = \mathbb{E}_{X \sim \mathbb{P}_\theta}[g(X)]$, where $g(x) = \psi(x) - \psi((1 + s)x)$ measures the shortfall in option payoff and the distribution $\mathbb{P}_\theta$ is induced by the price of the asset which is described by the Black-Scholes formula. The payoff function we consider is that of a butterfly call option: $\psi(x) = \max(x - K_1, 0) + \max(x - K_2, 0) - 2\max(x - (K_1 + K_2)/2, 0)$ for $K_1, K_2 \geq 0$. Since we incur a loss only if the final shortfall is positive, the expected loss of the option at maturity can be expressed as $I = \mathbb{E}_{\theta \sim \mathbb{Q}}[\max(\mathbb{E}_{X \sim \mathbb{P}_\theta}[g(X)], 0)]$. Under this setting,

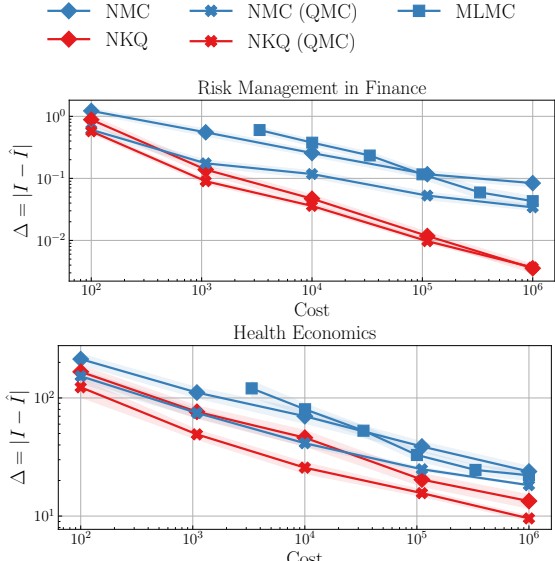

Figure 3. **Top:** *Financial risk management.* **Bottom:** *Health economics.* Results are averaged over 100 independent runs, while shaded regions give the 25%-75% quantiles.

$d_\Theta = d_\mathcal{X} = 1$ and $I = 3.077$ can be computed analytically.

In this experiment, Assumptions (2)(3) are satisfied with $s_\Theta = s_\mathcal{X} = 1$, but the *max* function is not in $C^2(\mathbb{R})$ which violates Assumption (4) (see Appendix F.3). Nevertheless, we still run NKQ with $k_\mathcal{X}$ and $k_\Theta$ being Matérn-$\frac{1}{2}$ kernels and choose $N = T = \Delta^{-1}$ for NKQ following Corollary 1. For NMC, we follow Gordy and Juneja (2010) and choose $N = \sqrt{T} = \Delta^{-1}$. For MLMC, we use $L = 5$ levels and allocate samples at each level following Giles and Goda (2019).

In Figure 3 **Top**, we present the mean absolute error of NKQ, NMC and MLMC with increasing cost. We see that NKQ outperforms both NMC and MLMC as expected. For each method, we obtain the empirical rate $r$ by linear regression in log-log space, and compare this against the theoretical rate in Table 1. For NMC, our estimate of $\hat{r} = 2.97$ matches theory ($r = 3$), but when using QMC samples instead, our estimate of $\hat{r} = 2.74$ shows we under-perform compared to the theoretical rate ($r = 2.5$). This is likely because the domains are unbounded and the measures are not uniform, breaking key assumptions. Finally, for NKQ, we obtain $\hat{r} = 1.90$ for i.i.d samples and $\hat{r} = 1.91$ for QMC samples which match (and even slightly outperform) the theoretical rate ($r = 2$).

**Value of Information for Healthcare Decision Making** In medical decision-making, a key metric to evaluate the cost-benefit trade-off of conducting additional tests on patients is the *expected value of partial perfect information (EVPPI)* (Brennan et al., 2007; Heath et al., 2017). Formally, let $g_c$ denote the patient outcome (such as quality-

adjusted life-years) under treatment $c$ in a set of possible treatments $\mathcal{C}$, and $\theta$ represent the additional variables that may be measured. Then, $J_c(\theta) = \mathbb{E}_{X \sim \mathbb{P}_\theta}[g_c(X, \theta)]$ represents the expected patient outcome given the measurement of $\theta$. The EVPPI is defined as $I = I_1 - \max_{c \in \mathcal{C}}(I_{2,c})$, where $I_1 = \mathbb{E}_{\theta \sim \mathbb{Q}}[\max_{c \in \mathcal{C}} J_c(\theta)]$ and $I_{2,c} = \mathbb{E}_{\theta \sim \mathbb{Q}}[J_c(\theta)]$ and therefore $I$ consists of $|\mathcal{C}| + 1$ nested expectations.

We follow Section 4.2 of Giles and Goda (2019), where both $\mathbb{P}_\theta$ and $\mathbb{Q}$ are Gaussians, and $g_1(x, \theta) = 10^4(\theta_1 x_5 x_6 + x_7 x_8 x_9) - (x_1 + x_2 x_3 x_4)$ and $g_2(x, \theta) = 10^4(\theta_2 x_{13} x_{14} + x_{15} x_{16} x_{17}) - (x_{10} + x_{11} x_{12} x_4)$. The exact practical meanings of each dimension of $x$ and $\theta$ can be found in Appendix F.4, but includes quantities such as 'cost of treatment' and 'duration of side effects'. Here we have $d_\mathcal{X} = 17$ and $d_\Theta = 2$, the former being relatively high dimensional. The ground truth EVPPI under this setting is $I = 538$ provided in Giles and Goda (2019).

For estimating both $I_1$ and $I_{2,c}$, Assumptions (2)(3) are satisfied with infinite smoothness $s_\mathcal{X} = s_\Theta = \infty$, but the *max* function in $I_1$ is only in $C^0(\mathbb{R})$ which violates Assumption (4). As a result, for estimating $I_1$ we take $k_\mathcal{X}$ to be a Gaussian kernel and $k_\Theta$ to be Matérn-$\frac{1}{2}$ kernel (so as to be conservative about the smoothness in $\theta$). For estimating $I_{2,c}$, we select both $k_\mathcal{X}$ and $k_\Theta$ to be Gaussian kernels. For NKQ, we choose $N = T = \Delta^{-1}$ whereas for NMC, we choose $N = \sqrt{T} = \Delta^{-1}$. For MLMC, we use $L = 5$ levels and allocate the samples at each level following Giles and Goda (2019). We run NKQ and NMC with both i.i.d. samples and QMC samples. In Figure 3 **Bottom**, we present the mean absolute error of NKQ, NMC and MLMC with increasing cost. We can see that NKQ consistently outperforms other baselines.

**Bayesian Optimization** We conclude with an application in Bayesian optimization. Typical acquisition functions are greedy approaches that maximize the immediate reward, while look-ahead acquisition functions optimize accumulated reward over a planning horizon, which results in reduced number of required function evaluations (Ginsbourger and Le Riche, 2010; González et al., 2016; Wu and Frazier, 2019; Yang et al., 2024). The utility of a two-step look ahead acquisition functions can be written as the following nested expectation.

$$\alpha(z; \mathcal{D}) := \mathbb{E}_{f_{|\mathcal{D}}}\left[g(f_{|\mathcal{D}}, z) + \max_{z'} \mathbb{E}_{f_{|\mathcal{D}'}}\left[g\left(f_{|\mathcal{D}'}, z'\right)\right]\right],$$

where $f_{|\mathcal{D}}, f_{|\mathcal{D}'}$ are the posterior distributions given data $\mathcal{D}$ and $\mathcal{D}' := \mathcal{D} \cup (z, f_{|\mathcal{D}}(z))$. In this experiment, the prior is a Gaussian process with zero mean and Matérn-0.5 covariance so the posterior $f_{|\mathcal{D}}$ remain a Gaussian process. The initial starting data $\mathcal{D}_0$ consists of 2 points sampled uniformly from a prespecified interval. Here, $g$ is the reward function and we use q-expected improvement (Wang et al., 2020) with $q = 2$ so $z = (z_1, z_2)$ and $g(f_{|\mathcal{D}}, z) =$

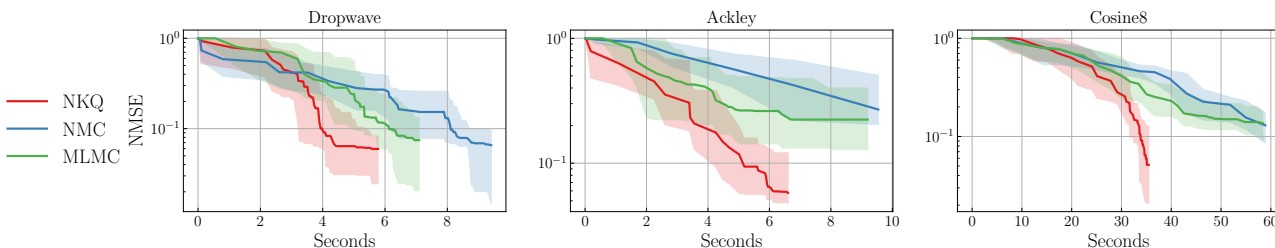

Figure 4. *Bayesian optimization with look ahead acquisition function.* The plots are NMSE against accumulative wall clock time. Results are averaged over 100 independent runs, while shaded regions give the 25%-75% quantiles.

$\max_{j=1,2}(f_{|\mathcal{D}}(z_j) - r_{\max}), 0)$. The constant $r_{\max}$ is the maximum reward obtained from previous queries. Although $f_{|\mathcal{D}}$ (resp. $f_{|\mathcal{D}'}$) is a Gaussian process, we only ever consider its evaluation on $z$ (resp. $z'$), and we therefore only have to integrate against two-dimensional Gaussians. Notationally speaking, $f_{|\mathcal{D}'}(z_1, z_2)$ correspond to $x$ and $f_{|\mathcal{D}}(z_1, z_2)$ correspond to $\theta$ in (1) (i.e. $d_{\mathcal{X}} = d_{\Theta} = 2$), but we use the notation of $f_{|\mathcal{D}'}, f_{|\mathcal{D}}$ to stay consistent with the GP literature. As a result of the *max* operation, $s_{\mathcal{X}} = 1$ but we do not have sufficient smoothness in $\Theta$.

We benchmark NKQ, NMC and MLMC on three synthetic tasks from BoTorch (Balandat et al., 2020). For NKQ, both $k_{\mathcal{X}}$ and $k_{\Theta}$ are Matérn-$\frac{1}{2}$ kernels since we want to be conservative about the smoothness. Although both $\mathbb{Q}$ and $\mathbb{P}_{\theta}$ are Gaussian so closed-form KMEs are available, we use the "change of variable trick" which maps Gaussian distributions to two uniform distributions over $[0, 1]^d$ (see Appendix F.5) to reduce the computational complexity of NKQ to $\mathcal{O}(T \times N)$. To reach a specific error threshold $\Delta = 0.01$, following Table 1, we choose $N = T = \Delta^{-2}$ for NMC and $N = T = \Delta^{-1}$ for NKQ. For MLMC, we use the same code as Yang et al. (2024). The normalized mean squared error (NMSE) $\frac{\| \max_{z \in \mathcal{D}_{\mathcal{S}}} f_{\mathrm{BB}}(z) - f_{\mathrm{BB}}(z^*) \|^2}{\| \max_{z \in \mathcal{D}_0} f_{\mathrm{BB}}(z) - f_{\mathrm{BB}}(z^*) \|^2}$ is used as performance metric, where $\mathcal{D}_0$ (resp. $\mathcal{D}_{\mathcal{S}}$) is queried data at initialization (resp. after $\mathcal{S}$ iterations), $f_{\mathrm{BB}}$ is the black box function to be optimized and $f_{\mathrm{BB}}(z^*)$ is the maximum reward.

In Figure 4, we compare the efficiency of each method by plotting their NMSE against cumulative computational time in wall clock. We can see that NKQ achieves the lowest NMSE among all methods under a fixed amount of computational time in all three datasets, even though the assumptions of Theorem 1 are not all satisfied. Since the Dropwave, Ackley, and Cosine8 functions are synthetic and computationally cheap (see Appendix F.5), we expect the advantages of NKQ to be more pronounced for Bayesian optimization on real-world expensive problems. Furthermore, many other utility functions in Bayesian optimization—such as predictive entropy search—involve nested expectations (Balandat et al., 2020). We leave the empirical evaluation of NKQ on these utility functions to

future work.

## 6. Conclusion

This paper introduces a novel estimator for nested expectations based on kernel quadrature. We prove in Theorem 1 that our method has a faster rate of convergence than existing methods provided that the problem has sufficient smoothness. This theoretical result is consistent with the empirical evidence in several numerical experiments. Additionally, even when the problem is not as smooth as the theory requires, NKQ can still outperform baseline methods potentially due to the use of non-equal weights.

Following our work, there remain a number of interesting future problems and we now highlight two main ones. Firstly, we propose a combination of KQ and MLMC that we call MLKQ in Appendix D.2. However, we believe our current theoretical rate for MLKQ is sub-optimal due to the sub-optimal allocation of samples at each level. Further work will therefore be needed to determine whether this is a viable approach in some cases. Secondly, for applications where function evaluations are extremely expensive, NKQ could be extended to its Bayesian counterpart. This would allow us to use the finite sample uncertainty quantification for adaptive selection of samples, which could further improve performance. Finally, establishing minimax lower bounds for nested expectation remains an open and compelling problem. The main difficulty lies in its two-stage structure. To the best of our knowledge, existing minimax lower bounds for two-stage problem typically reduce the problem to a one-stage problem before deriving the bound; see, for example, Chen and Reiss (2011, Chapter 3), Meunier et al. (2024, Appendix F.1), and Zhang et al. (2025). However, it remains unclear how to directly establish meaningful minimax lower bounds of genuinely two-stage problems, such as our nested expectation.

## Impact Statement

This paper presents work whose goal is to advance the field of Machine Learning. There are many potential societal consequences of our work, none which we feel must be specifically highlighted here.

## Acknowledgments

The authors acknowledge useful discussions with Philipp Hennig and support from the Engineering and Physical Sciences Research Council (ESPRC) through grants [EP/S021566/1] (for ZC and MN) and [EP/Y022300/1] (for FXB).

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

# Supplementary Material

**Table of Contents**

**Additional notations** For two normed vector spaces $A, B$, $A \cong B$ means that $A$ and $B$ are norm equivalent, i.e. their sets coincide and the corresponding norms are equivalent. In other words, there are constants $c_1, c_2 > 0$ such that $c_1 \|h\|_A \leq \|h\|_B \leq c_2 \|h\|_A$ holds for all $h \in A$, written as $\| \cdot \|_A \cong \| \cdot \|_B$. For $A \subseteq B$, $A$ is said to be continuously embedded in $B$ if the inclusion map between them is continuous, written as $A \hookrightarrow B$. $\|T\|$ denotes the norm of an operator $T : A \to B$. For a function $f : \mathcal{X} \subseteq \mathbb{R}^d \to \mathbb{R}$ and $\alpha \in \mathbb{N}^d$, we use $\partial_x^\alpha f$ to denote the standard derivative $\partial x_1^{\alpha_1} \cdots \partial x_n^{\alpha_n} f$ and $D_x^\alpha f$ to denote the weak derivative. For $f \in W_2^s(\mathcal{X})$, we use $\|f\|_{s,2} := \|f\|_{W_2^s(\mathcal{X})}$ to denote its Sobolev space norm. $\lesssim$ means $\leq$ up to some positive multiplicative constants.

## A. Existing Results on Kernel Ridge Regression

In this section, we present Proposition 1 to 3 which are adaptation of theorems from Fischer and Steinwart (2020) applied to Sobolev spaces. These propositions are foundations of the proof of Theorem 1 in Appendix C.

In the standard regression setting, we are given $N$ observations $\{x_i, y_i\}_{i=1}^N$ which are i.i.d sampled from an unknown joint distribution $\mathbf{P}$ on $\mathcal{X} \times \mathbb{R}$. Here, $\mathcal{X} \subset \mathbb{R}^d$ is a compact domain. The marginal distribution of $\mathbf{P}$ on $\mathcal{X}$ is $\pi$, and the conditional distribution $\mathbf{P}(\cdot \mid x)$ satisfies the Bernstein moment condition (Fischer and Steinwart, 2020). In other words, there exists constants $\sigma, L > 0$ independent of $x$ such that

$$\int_{\mathbb{R}} |y - h^*(x)|^m \, \mathbf{P}(dy \mid x) \leq \frac{1}{2} m! \sigma^2 L^{m-2} \tag{A.1}$$

is satisfied for $\pi$-almost all $x \in \mathcal{X}$ and all $m \geq 2$. For example, (A.1) is satisfied with $\sigma = L = \sigma_0$ when $\mathbf{P}(\cdot \mid x)$ is a Gaussian distribution with bounded variance $\sigma_0$. Additionally, (A.1) is also satisfied when there is no noise in the observation so $\sigma = L = 0$, which will be discussed in Appendix B.

In a regression problem, the target of interest is the Bayes predictor $h^* : \mathcal{X} \to \mathbb{R}$, $x \mapsto \mathbb{E}[Y \mid X = x]$. One way of estimating $h^*$ is through kernel ridge regression (Fischer and Steinwart, 2020): given a reproducing kernel $k : \mathcal{X} \times \mathcal{X} \to \mathbb{R}$, the kernel ridge regression estimator $\hat{h}_\lambda : \mathcal{X} \to \mathbb{R}$ is defined as the solution to the following optimization problem ($\lambda > 0$):

$$\hat{h}_\lambda = \arg\min_{h \in \mathcal{H}_k} \left\{ \lambda \|h\|_{\mathcal{H}_k}^2 + \frac{1}{N} \sum_{i=1}^N (y_i - h(x_i))^2 \right\}. \tag{A.2}$$

$\mathcal{H}_k$ is the reproducing kernel Hilbert space (RKHS) associated with a kernel $k$. Fortunately, it has the following closed-form expression (Gretton, 2013, Section 7)

$$\hat{h}_\lambda = k(\cdot, x_{1:N}) \left( k(x_{1:N}, x_{1:N}) + N\lambda \boldsymbol{I}_N \right)^{-1} y_{1:N}.$$

We also introduce an auxiliary function $h_\lambda : \mathcal{X} \to \mathcal{Y}$ which is the solution to another optimization problem:

$$h_\lambda = \arg\min_{f \in \mathcal{H}_k} \left\{ \lambda \|f\|_{\mathcal{H}}^2 + \int_{\mathcal{X} \times \mathbb{R}} (y - f(x))^2 \mathbf{P}(dx, dy) \right\}. \tag{A.3}$$

In regression setting, it is of interest to study the generalization error between the estimator $\hat{h}_\lambda$ and the Bayes optimal predictor $h^*$, $\|\hat{h}_\lambda - h^*\|_{L_2(\pi)}$, and particularly its asymptotic rate of convergence towards 0 as the number of samples $N$ tend to infinity. The generalization error can be decomposed into two terms, through a triangular inequality,

$$\|\hat{h}_\lambda - h^*\|_{L_2(\pi)} \leq \|\hat{h}_\lambda - h_\lambda\|_{L_2(\pi)} + \|h_\lambda - h^*\|_{L_2(\pi)}, \tag{A.4}$$

where the first term $\|\hat{h}_\lambda - h_\lambda\|_{L_2(\pi)}$ is known as the estimation error and the second term $\|h_\lambda - h^*\|_{L_2(\pi)}$ is known as the approximation error. Next, we are going to present propositions that study these two terms separately under the following list of conditions.

(S1) $k$ is a Sobolev reproducing kernel of smoothness $s > \frac{d}{2}$.

(S2) $\pi$ is a probability measure on $\mathcal{X}$ with density $p : \mathcal{X} \to \mathbb{R}$. There exist positive constants $G_0, G_1$ such that $G_0 \leq p(x) \leq G_1$ for any $x \in \mathcal{X}$.

(S3) The Bayes predictor $h^* \in W_2^s(\mathcal{X})$.

(S4) There exists universal constants $\sigma, L > 0$ such that (A.1) holds.

**Proposition 1** (Approximation error). *Under Assumptions (S1)-(S4),*

$$\|h_\lambda - h^*\|_{L_2(\pi)} \leq \|h^*\|_{s,2} \lambda^{\frac{1}{2}}.$$

*Proof.* This is direct application of Lemma 14 of (Fischer and Steinwart, 2020) with $\beta = 1$ and $\gamma = 0$. $\square$

**Proposition 2** (Estimation error). *Suppose Assumptions (S1)-(S4) hold. Let $\mathcal{N}(\lambda)$ be the effective dimension defined in Lemma 1, and $k_\alpha$ be defined in Lemma 2. If $N > A_{\lambda,\tau}$, then with probability at least $1 - 4e^{-\tau}$,*

$$\|h_\lambda - \hat{h}_\lambda\|^2_{L_2(\pi)} \leq \frac{576\tau^2}{N} \left( L^2 D \lambda^{-\frac{d}{2s}} + M^2 \lambda^{1-\frac{d}{2s}} \|h^*\|^2_{s,2} + M^2 \frac{L_\lambda^2}{N} \lambda^{-\frac{d}{2s}} \right), \tag{A.5}$$

*where $D$ and $M$ are constants independent of $N$, and $g_\lambda, A_{\lambda,\tau}, L_\lambda$ are defined as follows*

$$g_\lambda := \log\left( 2e\mathcal{N}(\lambda) \frac{\|\Sigma_\pi\| + \lambda}{\|\Sigma_\pi\|} \right), \quad A_{\lambda,\tau} := 8k_\alpha^2 \tau g_\lambda \lambda^{-\frac{d}{2s}}, \quad L_\lambda := \max\left\{ L, \lambda^{\frac{1}{2}-\frac{d}{4s}} \left( \|h^*\|_{L_\infty(\pi)} + k_\alpha \|h^*\|_{s,2} \right) \right\}.$$

*Proof.* This proposition is a special case of Theorem 16 in Fischer and Steinwart (2020) under the following adaptations towards our Sobolev space setting: 1) Lemma 1 proves that $\mathcal{N}(\lambda) \leq D\lambda^{-\frac{d}{2s}}$ and Lemma 2 proves that $k_\alpha \leq M$ for $\alpha = \frac{d}{2s}$. 2) $\|h^* - h_\lambda\|_{L_\infty(\pi)}$ is upper bounded by $\lambda^{\frac{1}{2}-\frac{d}{4s}} \left( \|h^*\|_{L_\infty(\pi)} + k_\alpha \|h^*\|_{s,2} \right)$ proved in Corollary 15 of Fischer and Steinwart (2020). $\|\Sigma_\pi\|$ is the norm of the covariance operator defined in (E.40). $\square$

**Proposition 3.** *Suppose Assumptions (S1)-(S4) hold. For $A_{\lambda,\tau}$ and $L_\lambda$ defined above in Proposition 2, if $N > A_{\lambda,\tau}$, then with probability at least $1 - 4e^{-\tau}$,*

$$\left\| \hat{h}_\lambda - h^* \right\|^2_{L_2(\pi)} \leq \frac{576\tau^2}{N} \left( L^2 D \lambda^{-\frac{d}{2s}} + M^2 \lambda^{1-\frac{d}{2s}} \|h^*\|^2_{s,2} + 2M^2 \frac{L_\lambda^2}{N} \lambda^{-\frac{d}{2s}} \right) + \|h^*\|^2_{s,2} \lambda.$$

*Proof.* By the triangle inequality in (A.4), combining Proposition 1 and Proposition 2 finishes the proof. $\square$

# B. Noiseless Kernel Ridge Regression (Kernel Quadrature)

In this section, we present the upper bound on the generalization error $\|h^* - \hat{h}_\lambda\|_{L_2(\pi)}$ in Proposition 3 adapted to the noiseless regression setting, which will be employed in the proof of Theorem 1 in the next section. Our proof follows the outline of the proof for Theorem 1 in (Fischer and Steinwart, 2020), modified for our choice of regularization parameter $\lambda$. Note that this section is of independent interest to some readers as it presents the first standalone proof on the convergence rate of kernel quadrature that 1): it allows positive regularization parameter $\lambda > 0$ and 2): it provides convergence in high probability rather than in expectation. The closely-related work is Bach (2017) which requires i.i.d samples from an intractable distribution; and Long et al. (2024) which provides a more general analysis on noiseless kernel ridge regression in both well-specified and mis-specified setting.

Suppose we have $N$ observations $x_{1:N}$ which are i.i.d sampled from an unknown distribution $\pi$ on $\mathcal{X}$ along with $N$ *noiseless* function evaluations $h^*(x_{1:N})$ where $h^* : \mathcal{X} \subset \mathbb{R}^d \to \mathbb{R}$. The setting appears for instance when the measurement of the output values is very accurate, or when the output values are obtained as a result of computer experiments.

**Proposition 4.** *Let $\mathcal{X} \subset \mathbb{R}^d$ be compact, and $x_{1:N}$ be $N$ i.i.d. samples from $\pi$. Define $\hat{h}_{\lambda_N}(\cdot) := k(\cdot, x_{1:N}) \left( k(x_{1:N}, x_{1:N}) + N\lambda_N \boldsymbol{I}_N \right)^{-1} h^*(x_{1:N})$, and suppose conditions (S1)-(S4) are satisfied. Then, if $\lambda_N \asymp N^{-\frac{2s}{d}} (\log N)^{\frac{2s+2}{d}}$, there exists an $N_0 > 0$ such that for all $N > N_0$,*

$$\|h^* - \hat{h}_{\lambda_N}\|_{L_2(\pi)} \leq \mathfrak{C}\tau N^{-\frac{s}{d}} (\log N)^{\frac{s+1}{d}} \|h^*\|_{s,2} \tag{B.6}$$

*holds with probability at least $1 - 4e^{-\tau}$, for a constant $\mathfrak{C} = \mathfrak{C}(\mathcal{X}, G_0, G_1)$ that only depends on $\mathcal{X}, G_0, G_1$.*

*Proof.* Notice that $\hat{h}_{\lambda_N}$ is precisely the solution to the optimization problem defined in (A.2) only with $y_i$ replaced by $h^*(x_i)$. Similarly, we define $h_{\lambda_N}$ as the solution to the optimization problem defined in (A.3) only with $y$ replaced by $h^*(x)$. Note that Assumption (S4) is instantly satisfied with $L = 0$.

Similar to the proof of Proposition 3, we decompose the generalization error into an estimation error term $\|h_{\lambda_N} - \hat{h}_{\lambda_N}\|_{L_2(\pi)}$ and an approximation error term $\|h_{\lambda_N} - h^*\|_{L_2(\pi)}$.

**Approximation error** Take $\lambda_N \asymp N^{-\frac{2s}{d}}(\log N)^{\frac{2s+2}{d}}$, then from Proposition 1, we have

$$\|h_{\lambda_N} - h^*\|_{L_2(\pi)} \le \lambda_N^{\frac{1}{2}}\|h^*\|_{s,2} \asymp N^{-\frac{s}{d}}(\log N)^{\frac{s+1}{d}}\|h^*\|_{s,2}.$$

**Estimation error** Recall all the constants $g_{\lambda_N}$, $A_{\lambda_N,\tau}$ and $L_{\lambda_N}$ defined in Proposition 2. Since $L = 0$, we know the constant $L_{\lambda_N} = \lambda_N^{\frac{1}{2}-\frac{d}{4s}}\left(\|h^*\|_{L_\infty(\pi)} + k_\alpha\|h^*\|_{s,2}\right)$. In order to apply Proposition 2, we need to check there indeed exists $N_0$ such that $N \ge A_{\lambda_N,\tau}$ is satisfied for all $N \ge N_0$. To this end, we are going to verify that $\lim_{N\to\infty} A_{\lambda_N,\tau}/N \to 0$. Notice that

$$\lim_{N\to\infty}\frac{A_{\lambda_N,\tau}}{N} = \frac{8k_\alpha^2 \tau g_{\lambda_N}\lambda_N^{-\frac{d}{2s}}}{N} = 8(\log N)^{-\frac{s+1}{s}}k_\alpha^2\tau\log\left(2e\mathcal{N}(\lambda_N)\frac{\|\Sigma_\pi\| + \lambda_N}{\|\Sigma_\pi\|}\right)$$

where $\mathcal{N}(\lambda_N)$ and $k_\alpha^2$ are defined in Lemma 1 and Lemma 2. Since $\lim_{N\to\infty}\lambda_N = \lim_{N\to\infty}N^{-\frac{2s}{d}}(\log N)^{\frac{2s+2}{d}} = 0$, there exists $N'$ such that $\lambda_N \le \|\Sigma_\pi\|$ for all $N \ge N'$. Therefore, as $N$ tends to infinity,

$$
\begin{aligned}
\lim_{N\to\infty}\frac{A_{\lambda_N,\tau}}{N} &\le \lim_{N\to\infty} 8(\log N)^{-\frac{s+1}{s}}k_\alpha^2\tau\log\left(4e\mathcal{N}(\lambda_N)\right)\\
&\le \lim_{N\to\infty} 8(\log N)^{-\frac{s+1}{s}}k_\alpha^2\tau\log\left(4eD\lambda_N^{-\frac{d}{2s}}\right)\\
&= \lim_{N\to\infty} 8(\log N)^{-\frac{s+1}{s}}k_\alpha^2\tau\log\left(4eD\right) + \lim_{N\to\infty} 8(\log N)^{-\frac{s+1}{s}}k_\alpha^2\tau\log\left(N(\log N)^{-\frac{s+1}{s}}\right) \quad\text{(B.7)}\\
&\le \lim_{N\to\infty} 16(\log N)^{-\frac{s+1}{s}}k_\alpha^2\tau\log\left(N\right)\\
&= 0,
\end{aligned}
$$

where $M$ and $D$ are constants defined in Lemma 1 and Lemma 2. So there exists $N''$ such that $N \ge A_{\lambda_N,\tau}$ for all $N \ge N''$. Taking $N_0 = \max\{N', N''\}$, then we have $N \ge A_{\lambda_N,\tau}$ for all $N \ge N_0$. From Proposition 2, we know that with probability at least $1 - 4e^{-\tau}$,

$$
\begin{aligned}
\|h_{\lambda_N} - \hat{h}_{\lambda_N}\|_{L_2(\pi)}^2 &\le \frac{576\tau^2}{N}\left(M^2\lambda_N^{1-\frac{d}{2s}}\|h^*\|_{s,2}^2 + M^2\lambda_N^{1-\frac{d}{2s}}\left(\|h^*\|_{L_\infty(\pi)} + M\|h^*\|_{s,2}\right)^2\frac{1}{N}\lambda_N^{-\frac{d}{2s}}\right)\\
&\asymp \frac{576\tau^2}{N}\left(M^2 N^{1-\frac{2s}{d}}(\log N)^{\frac{s+1}{s}\frac{2s-d}{d}}\|h^*\|_{s,2}^2 + M^2\left(\|h^*\|_{L_\infty(\pi)} + M\|h^*\|_{s,2}\right)^2 N^{1-\frac{2s}{d}}(\log N)^{\frac{s+1}{2s}\frac{4s-d}{d}}\right)\\
&\le 576\tau^2 N^{-\frac{2s}{d}}(\log N)^{\frac{2s+2}{d}}\left(M^2\|h^*\|_{s,2}^2 + M^2\left(\|h^*\|_{L_\infty(\pi)} + M\|h^*\|_{s,2}\right)^2\right).
\end{aligned}
$$

So we have, with probability at least $1 - 4e^{-\tau}$,

$$\|h_{\lambda_N} - \hat{h}_{\lambda_N}\|_{L_2(\pi)} \le 24\tau N^{-\frac{s}{d}}(\log N)^{\frac{s+1}{d}}\left((M + M^2)\|h^*\|_{s,2} + M\|h^*\|_{L_\infty(\pi)}\right).$$

**Combine approximation and estimation error** Combining the above two inequalities on approximation error $\|h_{\lambda_N} - h^*\|_{L_2(\pi)}$ and estimation error $\|h_{\lambda_N} - \hat{h}_{\lambda_N}\|_{L_2(\pi)}$, we have with probability at least $1 - 4e^{-\tau}$,

$$\|h^* - \hat{h}_{\lambda_N}\|_{L_2(\pi)} \le 24\tau N^{-\frac{s}{d}}(\log N)^{\frac{s+1}{d}}\left((1 + M + M^2)\|h^*\|_{s,2} + M\|h^*\|_{L_\infty(\pi)}\right).$$

Finally, following the arguments of Lemma 2 that the operator norm of $W_2^s(\mathcal{X}) \hookrightarrow L_\infty(\mathcal{X})$ is bounded, we have $\|h^*\|_{L_\infty(\pi)} \le R\|h^*\|_{s,2}$ where $R$ is a constant that depends on $\mathcal{X}, G_0, G_1$. With probability at least $1 - 4e^{-\tau}$,

$$\|h^* - \hat{h}_{\lambda_N}\|_{L_2(\pi)} \le 24\tau N^{-\frac{s}{d}}(\log N)^{\frac{s+1}{d}}(1 + (1+R)M + M^2)\|h^*\|_{s,2} = \mathfrak{C}\tau N^{-\frac{s}{d}}(\log N)^{\frac{s+1}{d}}\|h^*\|_{s,2},$$

for $\mathfrak{C} := 24(1 + (1+R)M + M^2)$, which concludes the proof. $\qquad\square$

**Corollary 2.** *Let $\mathcal{X}$ be a compact domain in $\mathbb{R}^d$ and $x_{1:N}$ are $N$ i.i.d samples from $\pi$. $\hat{I}_{KQ} := \mathbb{E}_{X\sim\pi}[k(X, x_{1:N})]\left(k(x_{1:N}, x_{1:N}) + N\lambda_N \boldsymbol{I}_N\right)^{-1}h^*(x_{1:N})$ is the KQ estimator defined in (5). Suppose conditions (A1)-(A3) are satisfied. Take $\lambda_N \asymp N^{-\frac{2s}{d}}(\log N)^{\frac{2s+2}{d}}$, then there exists $N_0 > 0$ such that for $N > N_0$,*

$$\left|\hat{I}_{KQ} - \int_{\mathcal{X}}h^*(x)d\pi(x)\right| \le \mathfrak{C}\tau N^{-\frac{s}{d}}(\log N)^{\frac{s+1}{d}} \quad\text{(B.8)}$$

*holds with probability at least $1 - 4e^{-\tau}$. Here $\mathfrak{C}$ is a constant that is independent of $N$.*

The proof of Corollary 2 is a direct application of Proposition 4 after observing the following.

$$\left| \hat{I}_{\text{KQ}} - \int h^*(x) d\pi(x) \right| \leq \int_{\mathcal{X}} \left| \hat{h}_{\lambda_N}(x) - h^*(x) \right| d\pi(x) = \| h^* - \hat{h}_{\lambda_N} \|_{L_1(\pi)} \leq \| h^* - \hat{h}_{\lambda_N} \|_{L_2(\pi)}.$$

**Remark B.1.** *We prove in Proposition 4 that the generalization error of $\hat{h}_{\lambda_N}$ in noiseless regression setting is $\tilde{\mathcal{O}}(N^{-\frac{s}{d}})$, which is faster than the minimax optimal rate $\mathcal{O}(N^{-\frac{s}{2s+d}})$ in standard regression setting. The fast rate is expected because we are in the noiseless regime so "overfitting" is not a problem — hence our choice of regularization parameter $\lambda_N \asymp N^{-\frac{2s}{d}} (\log N)^{\frac{2s+2}{d}}$ decays to 0 at a faster rate than $\lambda_N \asymp N^{-\frac{2s}{2s+d}}$ in standard kernel ridge regression (Fischer and Steinwart, 2020, Corollary 5). The $\tilde{\mathcal{O}}(N^{-\frac{s}{d}})$ rate is also optimal (up to logarithm terms) and cannot be further improved because it matches the lower bound of interpolation (Sections 1.3.11 and 1.3.1 of Novak (2006), Section 1.2, Chapter V of Ritter (2000)).*

**Remark B.2** (Comparison to existing upper bound of kernel (Bayesian) quadrature)**.** *The upper bound in Corollary 2 matches existing analysis based on scattered data approximation in the literature of both kernel quadrature and Bayesian quadrature (Sommariva and Vianello, 2006; Briol et al., 2019; Wynne et al., 2021) and is known to be minimax optimal (Novak, 2016; 2006). Existing analysis takes $\lambda = 0$ and requires the Gram matrix $k(x_{1:N}, x_{1:N})$ to be invertible, in contrast, our result allows a positive regularization parameter $\lambda_N \asymp N^{-\frac{2s}{d}} (\log N)^{\frac{2s+2}{d}}$ which improves numerical stability of matrix inversion in practice. One closely-related work is Bach (2017), but it requires i.i.d samples from an intractable distribution.*

## C. Proof of Theorem 1

**Remark C.1.** *In this section, we use $p(x; \theta)$ to denote the density $p_\theta(x)$ so that we can use $p(x; \cdot)$ to denote the mapping $\theta \mapsto p_\theta(x)$. Although we introduce a shorthand notation of kernel mean embedding in the main text, $\mu_\pi = \mathbb{E}_{X \sim \pi}[k(X, \cdot)]$, in this section we are going to write it out with its explicit formulation.*

For any $\theta \in \Theta$, $\hat{F}_{\text{KQ}} : \Theta \to \mathbb{R}$ and $\hat{J}_{\text{KQ}} : \Theta \to \mathbb{R}$ are two functions that generalize the definition of $\hat{F}_{\text{KQ}}(\theta_t)$ and $\hat{J}_{\text{KQ}}(\theta_t)$ in (7) to all $\theta \in \Theta$. To be more specific, for any $\theta \in \Theta$, given samples $x_{1:N}^{(\theta)} := \left[ x_1^{(\theta)}, \ldots, x_N^{(\theta)} \right]^\top$ consisting of $N$ i.i.d. samples from $\mathbb{P}_\theta$,

$$\hat{J}_{\text{KQ}}(\theta; x_{1:N}^{(\theta)}) := \left( \int_{\mathcal{X}} k_{\mathcal{X}}(x, x_{1:N}^{(\theta)}) d\mathbb{P}_\theta(x) \right) \left( k_{\mathcal{X}}(x_{1:N}^{(\theta)}, x_{1:N}^{(\theta)}) + N\lambda_{\mathcal{X}} \boldsymbol{I}_N \right)^{-1} g(x_{1:N}^{(\theta)}, \theta), \tag{C.9}$$

$$\hat{F}_{\text{KQ}}(\theta; x_{1:N}^{(\theta)}) := f(\hat{J}_{\text{KQ}}(\theta; x_{1:N}^{(\theta)})), \tag{C.10}$$

where we explicitly specify the dependence of samples $x_{1:N}^{(\theta)}$ on $\theta$ in the above two equations. Next, we define

$$\bar{J}_{\text{KQ}}(\theta) := \mathbb{E}_{x_{1:N}^{(\theta)} \sim \mathbb{P}_\theta} \left[ \hat{J}_{\text{KQ}}(\theta; x_{1:N}^{(\theta)}) \right] = \int \hat{J}_{\text{KQ}}(\theta; x_{1:N}) \prod_{i=1}^{N} p(x_i; \theta) dx_{1:N},$$

$$\bar{F}_{\text{KQ}}(\theta) := \mathbb{E}_{x_{1:N}^{(\theta)} \sim \mathbb{P}_\theta} \left[ \hat{F}_{\text{KQ}}(\theta; x_{1:N}^{(\theta)}) \right] = \int \hat{F}_{\text{KQ}}(\theta; x_{1:N}) \prod_{i=1}^{N} p(x_i; \theta) dx_{1:N}, \tag{C.11}$$

which marginalize out the dependence on samples $x_{1:N}^{(\theta)}$. We can see that $\bar{J}_{\text{KQ}} \in L_2(\mathbb{Q})$ since $g(x, \cdot) \in W_2^{s_\Theta}(\Theta) \subset L_2(\Theta) \cong L_2(\mathbb{Q})$ from Assumption (1) and (3); and $p(x_i; \cdot) \in L_2(\mathbb{Q})$. Also $\bar{F}_{\text{KQ}} \in L_2(\mathbb{Q})$ because $f$ is Lipschitz continuous from Assumption (4). Therefore, the absolute error $|I - \hat{I}_{\text{NKQ}}|$ can be decomposed as follows:

$$\left| I - \hat{I}_{\text{NKQ}} \right|$$

$$= \left| \int_\Theta F(\theta) q(\theta) d\theta - \left( \int_\Theta k_\Theta(\theta, \theta_{1:T}) q(\theta) d\theta \right) \left( k_\Theta(\theta_{1:T}, \theta_{1:T}) + T\lambda_\Theta \boldsymbol{I}_T \right)^{-1} \hat{F}_{\text{KQ}}(\theta_{1:T}) \right|$$

$$\leq \left| \int_\Theta F(\theta) q(\theta) d\theta - \int_\Theta \bar{F}_{\text{KQ}}(\theta) q(\theta) d\theta \right|$$

$$+ \left| \int_\Theta \bar{F}_{\text{KQ}}(\theta) q(\theta) d\theta - \left( \int_\Theta k_\Theta(\theta, \theta_{1:T}) q(\theta) d\theta \right) \left( k_\Theta(\theta_{1:T}, \theta_{1:T}) + T\lambda_\Theta \boldsymbol{I}_T \right)^{-1} \hat{F}_{\text{KQ}}(\theta_{1:T}) \right|$$

$$\leq \underbrace{\mathbb{E}_{\theta \sim \mathbb{Q}} \left[ \left| F(\theta) - \bar{F}_{\mathrm{KQ}}(\theta) \right| \right]}_{\text{Stage I error}} + \underbrace{\left\| \bar{F}_{\mathrm{KQ}}(\cdot) - k(\cdot, \theta_{1:T})(k_{\Theta}(\theta_{1:T}, \theta_{1:T}) + T\lambda_{\Theta} \boldsymbol{I}_T)^{-1} \hat{F}_{\mathrm{KQ}}(\theta_{1:T}) \right\|_{L_2(\mathbb{Q})}}_{\text{Stage II error}}. \tag{C.12}$$

The last inequality holds because $\| \cdot \|_{L_1(\mathbb{Q})} \leq \| \cdot \|_{L_2(\mathbb{Q})}$. Next, we analyze Stage I error and Stage II error separately.

**Stage I Error** From Assumption (4), $f$ is Lipschitz continuous and the Lipschitz constant is bounded by $S_4$,

$$\begin{aligned}
\left| \bar{F}_{\mathrm{KQ}}(\theta) - F(\theta) \right| &= \left| \mathbb{E}_{x_{1:N}^{(\theta)} \sim \mathbb{P}_\theta} \hat{F}_{\mathrm{KQ}}(\theta; x_{1:N}^{(\theta)}) - F(\theta) \right| \\
&\leq \mathbb{E}_{x_{1:N}^{(\theta)} \sim \mathbb{P}_\theta} \left| \hat{F}_{\mathrm{KQ}}(\theta; x_{1:N}^{(\theta)}) - F(\theta) \right| \\
&\leq S_4 \, \mathbb{E}_{x_{1:N}^{(\theta)} \sim \mathbb{P}_\theta} \left| \hat{J}_{\mathrm{KQ}}(\theta; x_{1:N}^{(\theta)}) - J(\theta) \right|,
\end{aligned} \tag{C.13}$$

where the first inequality holds by Jensen inequality and the last inequality holds by Lipschitz continuity of $f$. Define

$$\hat{g}(x, \theta; x_{1:N}^{(\theta)}) = k_{\mathcal{X}}(x, x_{1:N}^{(\theta)})(k_{\mathcal{X}}(x_{1:N}^{(\theta)}, x_{1:N}^{(\theta)}) + N\lambda_{\mathcal{X}} \boldsymbol{I}_N)^{-1} g(x_{1:N}^{(\theta)}, \theta). \tag{C.14}$$

Here $\hat{g}(\cdot, \theta; x_{1:N}^{(\theta)}) \in L_2(\mathbb{P}_\theta)$ because the Sobolev reproducing kernel $k_{\mathcal{X}}$ is bounded and measurable; and $g(\cdot, \theta) \in L_2(\mathbb{P}_\theta)$ by Assumption (3). Thus,

$$\left| \hat{J}_{\mathrm{KQ}}(\theta; x_{1:N}^{(\theta)}) - J(\theta) \right| = \left| \int (\hat{g}(x, \theta; x_{1:N}^{(\theta)}) - g(x, \theta)) p(x; \theta) dx \right| \leq \left\| \hat{g}(\cdot, \theta; x_{1:N}^{(\theta)}) - g(\cdot, \theta) \right\|_{L_2(\mathbb{P}_\theta)}. \tag{C.15}$$

Based on Assumption (2), $g(\cdot, \theta) \in W_2^{s_{\mathcal{X}}}(\mathcal{X})$ for any $\theta \in \Theta$. Therefore, based on Proposition 4, if one takes $\lambda_{\mathcal{X}, N} \asymp N^{-2\frac{s_{\mathcal{X}}}{d_{\mathcal{X}}}}(\log N)^{\frac{2s_{\mathcal{X}}+2}{d_{\mathcal{X}}}}$, then there exists $N_0$ such that for $N > N_0$,

$$\left\| \hat{g}(\cdot, \theta; x_{1:N}^{(\theta)}) - g(\cdot, \theta) \right\|_{L_2(\mathbb{P}_\theta)} \leq \mathfrak{C}\tau N^{-\frac{s_{\mathcal{X}}}{d_{\mathcal{X}}}}(\log N)^{\frac{s_{\mathcal{X}}+1}{d_{\mathcal{X}}}} \| g(\cdot, \theta) \|_{s_{\mathcal{X}}, 2}, \tag{C.16}$$

holds with probability at least $1 - 4e^{-\tau}$. The probability is taken over the distribution of $x_{1:N}^{(\theta)}$, i.e $\mathbb{P}_\theta$. Here $\mathfrak{C}$ is a constant independent of $N$. Hence, with Lemma 4, we have

$$\mathbb{E}_{x_{1:N}^{(\theta)} \sim \mathbb{P}_\theta} \left\| \hat{g}(\cdot, \theta; x_{1:N}^{(\theta)}) - g(\cdot, \theta) \right\|_{L_2(\mathbb{P}_\theta)} \leq \mathfrak{C} N^{-\frac{s_{\mathcal{X}}}{d_{\mathcal{X}}}}(\log N)^{\frac{s_{\mathcal{X}}+1}{d_{\mathcal{X}}}} \| g(\cdot, \theta) \|_{s_{\mathcal{X}}, 2}. \tag{C.17}$$

By plugging the above inequality back into (C.15), we obtain

$$\mathbb{E}_{x_{1:N}^{(\theta)} \sim \mathbb{P}_\theta} \left| \hat{J}_{\mathrm{KQ}}(\theta; x_{1:N}^{(\theta)}) - J(\theta) \right| \leq \mathfrak{C} N^{-\frac{s_{\mathcal{X}}}{d_{\mathcal{X}}}}(\log N)^{\frac{s_{\mathcal{X}}+1}{d_{\mathcal{X}}}} \| g(\cdot, \theta) \|_{s_{\mathcal{X}}, 2}.$$

Therefore, the Stage I error can be upper bounded by

$$\begin{aligned}
\mathbb{E}_{\theta \sim \mathbb{Q}} \left| F(\theta) - \bar{F}_{\mathrm{KQ}}(\theta) \right| &\leq S_4 \, \mathbb{E}_{x_{1:N}^{(\theta)} \sim \mathbb{P}_\theta} \left| \hat{J}_{\mathrm{KQ}}(\theta; x_{1:N}^{(\theta)}) - J(\theta) \right| \| g(\cdot, \theta) \|_{s_{\mathcal{X}}, 2} \\
&\leq S_4 S_1 \mathfrak{C} N^{-\frac{s_{\mathcal{X}}}{d_{\mathcal{X}}}}(\log N)^{\frac{s_{\mathcal{X}}+1}{d_{\mathcal{X}}}} \\
&= C_3 N^{-\frac{s_{\mathcal{X}}}{d_{\mathcal{X}}}}(\log N)^{\frac{s_{\mathcal{X}}+1}{d_{\mathcal{X}}}},
\end{aligned} \tag{C.18}$$

where $C_3 := S_4 S_1 \mathfrak{C}$ is a constant independent of $N$.

**Stage II Error** The upper bound on the stage II error is done in five steps. In step one, we prove that $\hat{J}_{\mathrm{KQ}}(\cdot; x_{1:N}^{(\theta)}) \in W_2^{s_{\Theta}}(\Theta)$ given fixed samples $x_{1:N}^{(\theta)}$. In step two, we show that $J \in W_2^{s_{\Theta}}(\Theta)$. In step three, we upper bound $\| \hat{J}_{\mathrm{KQ}}(\cdot; x_{1:N}^{(\theta)}) \|_{s_{\Theta}, 2}$ through the triangular inequality that $\| \hat{J}_{\mathrm{KQ}}(\cdot; x_{1:N}^{(\theta)}) \|_{s_{\Theta}, 2} \leq \| J \|_{s_{\Theta}, 2} + \| J - \hat{J}_{\mathrm{KQ}}(\cdot; x_{1:N}^{(\theta)}) \|_{s_{\Theta}, 2}$. In step four, we upper bound $\bar{F}_{\mathrm{KQ}}(\theta) = \mathbb{E}_{x_{1:N}^{(\theta)}} \left[ f \left( \hat{J}_{\mathrm{KQ}}(\cdot; x_{1:N}^{(\theta)}) \right) \right]$ through marginalizing out the samples $x_{1:N}^{(\theta)}$. In the last step, we use kernel ridge regression bound proved in Proposition 3 to upper bound the stage II error.

*Step One.* In this step, we are going to show that $\hat{J}_{KQ}$ lies in the Sobolev space $W_2^{s_\Theta}(\Theta)$ given fixed samples $x_{1:N}^{(\theta)}$. Notice that the dependence of $\hat{J}_{KQ}(\theta)$ on $\theta$ is through two mappings: $\theta \mapsto \int_{\mathcal{X}} k_{\mathcal{X}}(x, x_{1:N}^{(\theta)}) p(x; \theta) dx$ and $\theta \mapsto g(x_{1:N}^{(\theta)}, \theta)$. We are going to show that $\theta \mapsto \int_{\mathcal{X}} k_{\mathcal{X}}(x, x_i^{(\theta)}) p(x; \theta) dx$ lies in the Sobolev space $W_2^{s_\Theta}(\Theta)$ for any $i \in \{1, \ldots, N\}$. To this end, we are going to demonstrate it possesses weak derivatives up to and including order $s_\Theta$ that lie in $\mathcal{L}^2(\Theta)$. Take $\varphi : \Theta \to \mathbb{R}$ to be any infinitely differentiable function with compact support in $\Theta$ (commonly denoted as $\varphi \in C_c^\infty(\Theta)$), with its standard, non-weak derivative of order $\beta$ denoted by $\partial^\beta \varphi$. Since $\theta \mapsto p(x; \theta) \in W_2^{s_\Theta}(\Theta)$, for any $|\beta| \leq s_\Theta$ it has a weak derivative $\theta \mapsto D_\theta^\beta p(x; \theta) \in \mathcal{L}^2(\Theta)$. Then,

$$
\int_\Theta \varphi(\theta) \int_{\mathcal{X}} k_{\mathcal{X}}(x, x_i^{(\theta)}) D_\theta^\beta p(x; \theta) dx \stackrel{(i)}{=} \int_{\mathcal{X}} k_{\mathcal{X}}(x, x_i^{(\theta)}) \int_\Theta \varphi(\theta) D_\theta^\beta p(x; \theta) d\theta dx
$$
$$
\stackrel{(ii)}{=} (-1)^{|\beta|} \int_{\mathcal{X}} k_{\mathcal{X}}(x, x_i^{(\theta)}) \int_\Theta \partial^\beta \varphi(\theta) p(x; \theta) d\theta dx \stackrel{(iii)}{=} (-1)^{|\beta|} \int_\Theta \partial^\beta \varphi(\theta) \int_{\mathcal{X}} k_{\mathcal{X}}(x, x_i^{(\theta)}) p(x; \theta) dx d\theta. \tag{C.19}
$$

In the above chain of derivations, we are allowed to swap the integration order in $(i)$ by the Fubini theorem (Rudin, 1964) because $k_{\mathcal{X}}$ is bounded and the fact that $\theta \mapsto \varphi(\theta) \cdot D_\theta^\beta p(x; \theta) \in L_1(\Theta)$ since $D_\theta^\beta p(x; \cdot) \in L_2(\Theta)$ (Assumption (3)) and $\varphi \in L_2(\Theta)$; $(ii)$ holds by definition of weak derivatives for $D_\theta^\beta p(x; \theta)$; and $(iii)$ holds again by the Fubini theorem. By definition of weak derivatives, (C.19) shows that $\int_{\mathcal{X}} k_{\mathcal{X}}(x, x_i^{(\theta)}) p(x; \theta) dx$ has a weak derivative of order $\beta$ of the form

$$
D_\theta^\beta \left[ \int_{\mathcal{X}} k_{\mathcal{X}}(x, x_i^{(\theta)}) p(x; \theta) dx \right] = \int_{\mathcal{X}} k_{\mathcal{X}}(x, x_i^{(\theta)}) D_\theta^\beta p(x; \theta) dx
$$

Also, since $k_{\mathcal{X}}$ is bounded and $\theta \mapsto D_\theta^\beta p(x; \theta) \in \mathcal{L}^2(\Theta)$, the weak derivative above is in $\mathcal{L}^2(\Theta)$. Consequently, we have

$$
\sum_{|\beta| \leq s_\Theta} \int_\Theta \left| D_\theta^\beta \int_{\mathcal{X}} k_{\mathcal{X}}(x, x_i^{(\theta)}) p(x; \theta) dx \right|^2 d\theta = \sum_{|\beta| \leq s_\Theta} \int_\Theta \left| \int_{\mathcal{X}} k_{\mathcal{X}}(x, x_i^{(\theta)}) D_\theta^\beta p(x; \theta) dx \right|^2 d\theta
$$
$$
\stackrel{(i)}{\leq} \text{Vol}(\mathcal{X}) \sum_{|\beta| \leq s_\Theta} \int_\Theta \int_{\mathcal{X}} \left| k_{\mathcal{X}}(x, x_i^{(\theta)}) D_\theta^\beta p(x; \theta) dx \right|^2 d\theta
$$
$$
\stackrel{(ii)}{\leq} \text{Vol}(\mathcal{X}) \sum_{|\beta| \leq s_\Theta} \kappa^2 \int_\Theta \int_{\mathcal{X}} \left| D_\theta^\beta p(x; \theta) \right|^2 dx d\theta
$$
$$
\stackrel{(iii)}{=} \text{Vol}(\mathcal{X}) \kappa^2 \int_{\mathcal{X}} \| p(x; \cdot) \|_{s_\Theta, 2}^2 dx.
$$

In the above chain of derivations, $(i)$ holds because $|\int_{\mathcal{X}} f(x) dx|^2 \leq \text{Vol}(\mathcal{X}) \int_{\mathcal{X}} |f(x)|^2 dx$ for compact $\mathcal{X}$, $(ii)$ holds because $k_{\mathcal{X}}$ is upper bounded by $\kappa$ and $\int_\Theta |D_\theta^\beta p(x; \theta)|^2 d\theta < \infty$ from Assumption (3), $(iii)$ holds because $p(x; \cdot) \in W_2^{s_\Theta}(\Theta)$ for any $x \in \mathcal{X}$ based on Assumption (3). Also, one can interchange the order of integration in $(iii)$ by the Fubini's theorem (Rudin, 1964).

As a result, for any $i, j \in \{1, \ldots, N\}$, we have $f_{1,i} : \theta \mapsto \int_{\mathcal{X}} k_{\mathcal{X}}(x, x_i^{(\theta)}) p(x; \theta) dx \in W_2^{s_\Theta}(\Theta)$ and $f_{2,j} : \theta \mapsto g(x_j^{(\theta)}, \theta) \in W_2^{s_\Theta}(\Theta)$ from Assumption (3). Therefore, we know from Lemma 3 that their product $f_{1,i} \cdot f_{2,j} \in W_2^{s_\Theta}(\Theta)$ hence $\hat{J}_{KQ}$ as a linear combination of $f_{1,i} \cdot f_{2,j}$ is in $W_2^{s_\Theta}(\Theta)$.

*Step Two.* In this step, we are going to show that $J : \theta \mapsto \int_{\mathcal{X}} g(x, \theta) p(x; \theta) dx$ is also in the Sobolev space $W_2^{s_\Theta}(\Theta)$. Since both $g(x, \cdot) \in W_2^{s_\Theta}(\Theta)$ and $p(x; \cdot) \in W_2^{s_\Theta}(\Theta)$, we know from Lemma 3 that $\theta \mapsto g(x, \theta) \cdot p(x, \theta) \in W_2^{s_\Theta}(\Theta)$. By following the same steps as in (C.19), we obtain that for any $|\beta| \leq s_\Theta$,

$$
D_\theta^\beta \int_{\mathcal{X}} g(x, \theta) p(x; \theta) dx = \int_{\mathcal{X}} D_\theta^\beta \Big( g(x, \theta) p(x; \theta) \Big) dx. \tag{C.20}
$$

We are now ready to study the Sobolev norm of $J$,

$$
\| J \|_{s_\Theta, 2}^2 := \sum_{|\beta| \leq s_\Theta} \int_\Theta \left| D_\theta^\beta \int_{\mathcal{X}} p(x; \theta) g(x, \theta) dx \right|^2 d\theta
$$

$$\overset{(i)}{=} \sum_{|\beta| \le s_\Theta} \int_\Theta \left| \int_\mathcal{X} D_\theta^\beta \Big( p(x; \theta) g(x, \theta) \Big) dx \right|^2 d\theta$$

$$\overset{(ii)}{\le} \mathrm{Vol}(\mathcal{X}) \int_\mathcal{X} \sum_{|\beta| \le s_\Theta} \int_\Theta \left| D_\theta^\beta \Big( p(x; \theta) g(x, \theta) \Big) \right|^2 d\theta dx$$

$$\overset{(iii)}{=} \mathrm{Vol}(\mathcal{X}) \int_\mathcal{X} \| p(x; \cdot) g(x, \cdot) \|_{s_\Theta, 2}^2 \, dx$$

$$\overset{(iv)}{\le} \mathrm{Vol}(\mathcal{X})^2 S_2^2 S_3^2 \tag{C.21}$$

Here, $(i)$ holds by (C.20), $(ii)$ holds since $|\int_\mathcal{X} f(x) dx|^2 \le \mathrm{Vol}(\mathcal{X}) \int_\mathcal{X} |f(x)|^2 dx$ for compact $\mathcal{X}$, $(iii)$ follows from the definition of Sobolev norm, and $(iv)$ holds by Lemma 3 and Assumption (3) that $\|g(x, \cdot)\|_{s_\Theta, 2} \le S_2$, $\|p(x; \cdot)\|_{s_\Theta, 2} \le S_3$.

*Step Three.* In this step, we study the Sobolev norm of $\hat{J}_{\mathrm{KQ}}$ for some fixed $x_{1:N}^{(\theta)}$, by upper bounding it with $\|J - \hat{J}_{\mathrm{KQ}}(\cdot; x_{1:N}^{(\theta)})\|_{s_\Theta, 2} + \|J\|_{s_\Theta, 2}$. Since $g(x_i^{(\theta)}, \cdot) \in W_2^{s_\Theta}(\Theta)$ by Assumption (3), it holds that $\hat{g}(x, \cdot; x_{1:N}^{(\theta)}) = k_\mathcal{X}(x, x_{1:N}^{(\theta)})(k_\mathcal{X}(x_{1:N}^{(\theta)}, x_{1:N}^{(\theta)}) + N\lambda_\mathcal{X} \boldsymbol{I}_N)^{-1} g(x_{1:N}^{(\theta)}, \cdot)$ is in $W_2^{s_\Theta}(\Theta)$ for any fixed $x_{1:N}^{(\theta)}$. Therefore,

$$\left\| p(x; \cdot) \Big( g(x, \cdot) - \hat{g}(x, \cdot; x_{1:N}^{(\theta)}) \Big) \right\|_{s_\Theta, 2} \le \| p(x; \cdot) \|_{s_\Theta, 2} \left\| g(x, \cdot) - \hat{g}(x, \cdot; x_{1:N}^{(\theta)}) \right\|_{s_\Theta, 2}$$

$$\le S_3 \left\| g(x, \cdot) - \hat{g}(x, \cdot; x_{1:N}^{(\theta)}) \right\|_{s_\Theta, 2}, \tag{C.22}$$

where the first inequality holds by Lemma 3 and the second inequality holds by Assumption (3) that $\|p(x; \cdot)\|_{s_\Theta, 2} \le S_3$. Now, we consider the Sobolev norm of $\hat{J}_{\mathrm{KQ}} - J$,

$$\left\| \hat{J}_{\mathrm{KQ}}(\cdot; x_{1:N}^{(\theta)}) - J \right\|_{s_\Theta, 2}^2 = \sum_{|\beta| \le s_\Theta} \int_\Theta \left| D_\theta^\beta \int_\mathcal{X} p(x; \theta) \Big( g(x, \theta) - \hat{g}(x, \theta; x_{1:N}^{(\theta)}) \Big) dx \right|^2 d\theta$$

$$\overset{(i)}{=} \sum_{|\beta| \le s_\Theta} \int_\Theta \left| \int_\mathcal{X} D_\theta^\beta \Big( p(x; \theta) \big( g(x, \theta) - \hat{g}(x, \theta; x_{1:N}^{(\theta)}) \big) \Big) dx \right|^2 d\theta$$

$$\overset{(ii)}{\le} \mathrm{Vol}(\mathcal{X}) \int_\mathcal{X} \sum_{|\beta| \le s_\Theta} \int_\Theta \left| D_\theta^\beta \Big( p(x; \theta) \big( g(x, \theta) - \hat{g}(x, \theta; x_{1:N}^{(\theta)}) \big) \Big) \right|^2 d\theta dx$$

$$\overset{(iii)}{=} \mathrm{Vol}(\mathcal{X}) \int_\mathcal{X} \left\| p(x; \cdot) \Big( g(x, \cdot) - \hat{g}(x, \cdot; x_{1:N}^{(\theta)}) \Big) \right\|_{s_\Theta, 2}^2 dx$$

$$\overset{(iv)}{\le} \mathrm{Vol}(\mathcal{X}) S_3^2 \int_\mathcal{X} \left\| g(x, \cdot) - \hat{g}(x, \cdot; x_{1:N}^{(\theta)}) \right\|_{s_\Theta, 2}^2 dx, \tag{C.23}$$

where the above chain of derivations (i) — (iv) follow the exact same reasoning as (C.19) and (C.21). Next, notice that

$$\int_\mathcal{X} \left\| g(x, \cdot) - \hat{g}(x, \cdot; x_{1:N}^{(\theta)}) \right\|_{s_\Theta, 2}^2 dx = \int_\mathcal{X} \sum_{|\beta| \le s_\Theta} \int_\Theta \left| D_\theta^\beta \Big( g(x, \theta) - \hat{g}(x, \theta; x_{1:N}^{(\theta)}) \Big) \right|^2 d\theta dx$$

$$= \sum_{|\beta| \le s_\Theta} \int_\Theta \left\| D_\theta^\beta g(\cdot, \theta) - D_\theta^\beta \hat{g}(\cdot, \theta; x_{1:N}^{(\theta)}) \right\|_{L_2(\mathcal{X})}^2 d\theta. \tag{C.24}$$

By Assumption (2), $D_\theta^\beta g(\cdot, \theta) \in W_2^{s_\mathcal{X}}(\mathcal{X})$ for any $|\beta| \le s_\Theta$. Therefore, by applying Proposition 4 with $h^*(\cdot) := D_\theta^\beta g(\cdot, \theta)$, and $\hat{h}_\lambda(\cdot) := D_\theta^\beta \hat{g}(\cdot, \theta; x_{1:N}^{(\theta)}) = k_\mathcal{X}(\cdot, x_{1:N}^{(\theta)})(k_\mathcal{X}(x_{1:N}^{(\theta)}, x_{1:N}^{(\theta)}) + N\lambda_\mathcal{X} \boldsymbol{I}_N)^{-1} D_\theta^\beta g(x_{1:N}^{(\theta)}, \theta; x_{1:N}^{(\theta)})$, we get that

$$\left\| D_\theta^\beta g(\cdot, \theta) - D_\theta^\beta \hat{g}(\cdot, \theta; x_{1:N}^{(\theta)}) \right\|_{L_2(\mathbb{P}_\theta)} \le \mathfrak{C} \tau N^{-\frac{s_\mathcal{X}}{d_\mathcal{X}}} (\log N)^{\frac{s_\mathcal{X}+1}{d_\mathcal{X}}} \left\| D_\theta^\beta g(\cdot, \theta) \right\|_{s_\mathcal{X}, 2} \tag{C.25}$$

holds with probability at least $1 - 4e^{-\tau}$, for a $\mathfrak{C}$ that only depends on $\mathcal{X}, G_{0,\mathcal{X}}, G_{1,\mathcal{X}}$. From Assumption (1), we know that $L_2(\mathbb{P}_\theta) \cong L_2(\mathcal{X})$ (they are norm equivalent) and $\|f\|_{L_2(\mathcal{X})} \le \mathrm{Vol}(\mathcal{X})^{-1} G_{0,\mathcal{X}}^{-1} \|f\|_{L_2(\mathbb{P}_\theta)}$ for any $f \in L_2(\mathbb{P}_\theta)$. Therefore,

for any $\theta \in \Theta$ and any $|\beta| \le s_\Theta$, with probability at least $1 - 4e^{-\tau}$,

$$\left\| D_\theta^\beta g(\cdot, \theta) - D_\theta^\beta \hat{g}(\cdot, \theta; x_{1:N}^{(\theta)}) \right\|_{L_2(\mathcal{X})} \le \mathfrak{C}\tau \text{Vol}(\mathcal{X})^{-1} G_{0,\mathcal{X}}^{-1} N^{-\frac{s_\mathcal{X}}{d_\mathcal{X}}} (\log N)^{\frac{s_\mathcal{X}+1}{d_\mathcal{X}}} \left\| D_\theta^\beta g(\cdot, \theta) \right\|_{s_\mathcal{X},2}$$

$$\le \mathfrak{C}\tau \text{Vol}(\mathcal{X})^{-1} G_{0,\mathcal{X}}^{-1} N^{-\frac{s_\mathcal{X}}{d_\mathcal{X}}} (\log N)^{\frac{s_\mathcal{X}+1}{d_\mathcal{X}}} S_1. \tag{C.26}$$

By plugging (C.26) into (C.24), and then plugging the result into (C.23), we get that with probability at least $1 - 4e^{-\tau}$,

$$\left\| \hat{J}_{\text{KQ}}(\cdot; x_{1:N}^{(\theta)}) - J \right\|_{s_\Theta,2} \le \left( \sum_{|\beta| \le s_\Theta} 1 \right) \mathfrak{C}\tau G_{0,\mathcal{X}}^{-1} \text{Vol}(\mathcal{X})^{-1} S_1 S_3 N^{-\frac{s_\mathcal{X}}{d_\mathcal{X}}} (\log N)^{\frac{s_\mathcal{X}+1}{d_\mathcal{X}}}$$

$$= \binom{s_\Theta + d_\Theta - 1}{d_\Theta - 1} \mathfrak{C}\tau G_{0,\mathcal{X}}^{-1} \text{Vol}(\mathcal{X})^{-1} S_1 S_3 N^{-\frac{s_\mathcal{X}}{d_\mathcal{X}}} (\log N)^{\frac{s_\mathcal{X}+1}{d_\mathcal{X}}}. \tag{C.27}$$

By combining this result with the bound $\|J\|_{s_\Theta,2} \le \text{Vol}(\mathcal{X}) S_2 S_3$ proven in (C.21), we get that with probability at least $1 - 4e^{-\tau}$ and any $N > N_0$ it holds that

$$\left\| \hat{J}_{\text{KQ}}(\cdot; x_{1:N}^{(\theta)}) \right\|_{s_\Theta,2} \le \left\| \hat{J}_{\text{KQ}}(\cdot; x_{1:N}^{(\theta)}) - J \right\|_{s_\Theta,2} + \|J\|_{s_\Theta,2}$$

$$\le \binom{s_\Theta + d_\Theta - 1}{d_\Theta - 1} \mathfrak{C}\tau G_{0,\mathcal{X}}^{-1} \text{Vol}(\mathcal{X})^{-1} S_1 S_3 N^{-\frac{s_\mathcal{X}}{d_\mathcal{X}}} (\log N)^{\frac{s_\mathcal{X}+1}{d_\mathcal{X}}} + \text{Vol}(\mathcal{X}) S_2 S_3$$

$$\le 2\text{Vol}(\mathcal{X}) S_2 S_3, \tag{C.28}$$

where $N_0$ is defined as the smallest integer for which the first term is subsumed by the second term.

*Step Four.* In this step, we are going to upper bound the Sobolev norm of $\bar{F}_{\text{KQ}}$. From Chapter 5, Exercise 16 of (Evans, 2022), we have $\hat{F}_{\text{KQ}} = f \circ \hat{J}_{\text{KQ}}$ is in $W_2^{s_\Theta}(\Theta)$ because $f$ has bounded derivatives up to including $s_\Theta + 1$ and $\|\hat{J}(\cdot; x_{1:N}^{(\theta)})\|_{s_\Theta,2} \le \text{Vol}(\mathcal{X}) S_2 S_3$ with probability at least $1 - 4e^{-\tau}$ proved in (C.28). Hence, $\|\hat{F}_{\text{KQ}}(\cdot; x_{1:N}^{(\theta)})\|_{s_\Theta,2} \le C_6$ holds with probability at least $1 - 4e^{-\tau}$. Next, recall the definition of $\bar{F}_{\text{KQ}}(\theta)$ in (C.11),

$$\bar{F}_{\text{KQ}}(\theta) = \mathbb{E}_{x_{1:N}^{(\theta)} \sim \mathbb{P}_\theta} \left[ \hat{F}_{\text{KQ}}\left(\theta; x_{1:N}^{(\theta)}\right) \right] = \int_\mathcal{X} \cdots \int_\mathcal{X} \hat{F}_{\text{KQ}}(\theta; x_{1:N}^{(\theta)}) p(x_1^{(\theta)}; \theta) p(x_2^{(\theta)}; \theta) \cdots p(x_N^{(\theta)}; \theta) dx_1^{(\theta)} dx_2^{(\theta)} \cdots dx_N^{(\theta)}.$$

For any $i = 1, \ldots, N$, we know that $\|p(x_i^{(\theta)}; \cdot)\|_{s_\Theta,2} \le S_3$ from Assumption (3) and $\|\hat{F}_{\text{KQ}}(\cdot; x_{1:N}^{(\theta)})\|_{s_\Theta,2} \le C_6$ proved above. Therefore, from Lemma 3 we have $\|p(x_i^{(\theta)}; \cdot)\hat{F}_{\text{KQ}}(\cdot; x_{1:N}^{(\theta)})\|_{s_\Theta,2}$ is bounded, so $x_i^{(\theta)} \mapsto p(x_i^{(\theta)}; \cdot)\hat{F}_{\text{KQ}}(\cdot; x_{1:N}^{(\theta)})$ is Bochner integrable with respect to the Lebesgue measure $\mathcal{L}_\mathcal{X}$. From Lemma 5, we have,

$$\left\| \bar{F}_{\text{KQ}} \right\|_{s_\Theta,2} \le \int_\mathcal{X} \cdots \int_\mathcal{X} \left\| \hat{F}_{\text{KQ}}(\cdot; x_{1:N}^{(\theta)}) \prod_{n=1}^N p(x_n^{(\theta)}; \cdot) \right\|_{s_\Theta,2} dx_1^{(\theta)} dx_2^{(\theta)} \cdots dx_N^{(\theta)}$$

$$\le \int_\mathcal{X} \cdots \int_\mathcal{X} \left\| \hat{F}_{\text{KQ}}(\cdot; x_{1:N}^{(\theta)}) \right\|_{s_\Theta,2} \left( \prod_{n=1}^N \|p(x_n^{(\theta)}; \cdot)\|_{s_\Theta,2} \right) dx_1^{(\theta)} dx_2^{(\theta)} \cdots dx_N^{(\theta)}$$

$$\le C_6 S_3^N \text{Vol}(\mathcal{X})^N$$

$$\le C_6. \tag{C.29}$$

The last inequality holds by $S_3 \le 1$ from Assumption (3) and $\mathcal{X} = [0,1]^{d_\mathcal{X}}$ so $\text{Vol}(\mathcal{X}) = 1$.

*Step Five.* We are now ready to upper bound the stage II error, which was defined as

$$\text{Stage II error} = \left\| \bar{F}_{\text{KQ}}(\cdot) - k(\cdot, \theta_{1:T})(k_\Theta(\theta_{1:T}, \theta_{1:T}) + T\lambda_\Theta \boldsymbol{I}_T)^{-1} \hat{F}_{\text{KQ}}(\theta_{1:T}) \right\|_{L_2(\mathbb{Q})}.$$

The idea is to treat the stage II error as the generalization error of kernel ridge regression—which can be bounded via Proposition 3. Given i.i.d. observations $(\theta_1, \hat{F}_{\text{KQ}}(\theta_1, x_{1:N}^{(\theta_1)})), \ldots, (\theta_T, \hat{F}_{\text{KQ}}(\theta_T, x_{1:N}^{(\theta_T)}))$, the target of interest in the context

of regression is the conditional mean, which in our case is precisely $\bar{F}_{\text{KQ}}(\theta) = \mathbb{E}_{x_{1:N}^{(\theta)} \sim \mathbb{P}_\theta} \hat{F}_{\text{KQ}}(\theta; x_{1:N}^{(\theta)})$ defined in (C.11). Alternatively, $\hat{F}_{\text{KQ}}(\theta; x_{1:N}^{(\theta)})$ can be treated as noisy observation of the target function $\bar{F}_{\text{KQ}}(\theta)$ where the observation noise is defined as $r : \Theta \to \mathbb{R}$ with $r(\theta; x_{1:N}^{(\theta)}) = \hat{F}_{\text{KQ}}(\theta; x_{1:N}^{(\theta)}) - \bar{F}_{\text{KQ}}(\theta)$. So we automatically have $\mathbb{E}_{x_{1:N}^{(\theta)} \sim \mathbb{P}_\theta}[r(\theta)] = 0$. For any positive integer $m \geq 2$,

$$
\begin{aligned}
\mathbb{E}_{x_{1:N}^{(\theta)} \sim \mathbb{P}_\theta}[|r(\theta)|^m] &= \mathbb{E}_{x_{1:N}^{(\theta)} \sim \mathbb{P}_\theta} \left| \hat{F}_{\text{KQ}}(\theta; x_{1:N}^{(\theta)}) - \bar{F}_{\text{KQ}}(\theta) \right|^m \\
&\stackrel{(i)}{\leq} 2^{m-1} \mathbb{E}_{x_{1:N}^{(\theta)} \sim \mathbb{P}_\theta} \left| \hat{F}_{\text{KQ}}(\theta; x_{1:N}^{(\theta)}) - F(\theta) \right|^m + 2^{m-1} \left| \bar{F}_{\text{KQ}}(\theta) - F(\theta) \right|^m \\
&= 2^{m-1} \mathbb{E}_{x_{1:N}^{(\theta)} \sim \mathbb{P}_\theta} \left| \hat{F}_{\text{KQ}}(\theta; x_{1:N}^{(\theta)}) - F(\theta) \right|^m + 2^{m-1} \left| \mathbb{E}_{x_{1:N}^{(\theta)} \sim \mathbb{P}_\theta} \hat{F}_{\text{KQ}}(\theta; x_{1:N}^{(\theta)}) - F(\theta) \right|^m \\
&\leq 2^{m-1} \mathbb{E}_{x_{1:N}^{(\theta)} \sim \mathbb{P}_\theta} \left| \hat{F}_{\text{KQ}}(\theta; x_{1:N}^{(\theta)}) - F(\theta) \right|^m + 2^{m-1} \mathbb{E}_{x_{1:N}^{(\theta)} \sim \mathbb{P}_\theta} \left| \hat{F}_{\text{KQ}}(\theta; x_{1:N}^{(\theta)}) - F(\theta) \right|^m \\
&= 2^m \mathbb{E}_{x_{1:N}^{(\theta)} \sim \mathbb{P}_\theta} \left| \hat{F}_{\text{KQ}}(\theta; x_{1:N}^{(\theta)}) - F(\theta) \right|^m \\
&\leq 2^m S_4^m \mathbb{E}_{x_{1:N}^{(\theta)} \sim \mathbb{P}_\theta} \left| \hat{J}_{\text{KQ}}(\theta; x_{1:N}^{(\theta)}) - J(\theta) \right|^m \\
&\stackrel{(ii)}{\leq} 2^m m! S_4^m S_1^m \mathfrak{C}^m N^{-m \frac{s_{\mathcal{X}}}{d_{\mathcal{X}}}} (\log N)^{m \frac{s_{\mathcal{X}}+1}{d_{\mathcal{X}}}}.
\end{aligned}
\tag{C.30}
$$

In the above chain of derivations, $(i)$ holds because $(a+b)^m \leq 2^{m-1}(a^m + b^m)$. $(ii)$ holds because we know from (C.15) and (C.16) that $|\hat{J}_{\text{KQ}}(\theta; x_{1:N}^{(\theta)}) - J(\theta)| \leq \mathfrak{C}\tau N^{-\frac{s_{\mathcal{X}}}{d_{\mathcal{X}}}} (\log N)^{\frac{s_{\mathcal{X}}+1}{d_{\mathcal{X}}}} S_1$ holds with probability at least $1 - 4e^{-\tau}$, and so $\mathbb{E}_{x_{1:N}^{(\theta)} \sim \mathbb{P}_\theta} |\hat{J}_{\text{KQ}}(\theta; x_{1:N}^{(\theta)}) - J(\theta)|^m$ can be bounded via Lemma 4. Therefore, by comparing (C.30) with (A.1), we can see that the observation noise $r$ indeed satisfy the Bernstein noise moment condition with

$$
\sigma = L = 2S_4 S_1 \mathfrak{C} N^{-\frac{s_{\mathcal{X}}}{d_{\mathcal{X}}}} (\log N)^{\frac{s_{\mathcal{X}}+1}{d_{\mathcal{X}}}} = C_7 N^{-\frac{s_{\mathcal{X}}}{d_{\mathcal{X}}}} (\log N)^{\frac{s_{\mathcal{X}}+1}{d_{\mathcal{X}}}},
$$

for $C_7 := 2S_4 S_1 \mathfrak{C}$ a constant independent of $N, T$. Before we employ Proposition 3, we need to check the Assumptions (S1)—(S4). Assumption (S1) is satisfied for our choice of kernel $k_\Theta$. Assumption (S2) is satisfied due to Assumption (1). Assumption (S3) is satisfied due to (C.29). Assumption (S4) is satisfied for the Bernstein noise moment condition verified above. Next, we compute all the constants in Proposition 3 in the current context. $\mathcal{N}(\lambda_\Theta)$ is the effective dimension defined in Lemma 1 upper bounded by $D_\Theta \lambda_\Theta^{-d_\Theta/2s_\Theta}$, $k_\alpha$ with $\alpha = \frac{2s_\Theta}{d_\Theta}$ defined in Lemma 2 is upper bounded by a constant $M_\Theta$, $\|\Sigma_\mathbb{Q}\|$ is the norm of the covariance operator defined in (E.40). Hence

$$
g_{\lambda_\Theta} := \log \left( 2e\mathcal{N}(\lambda_\Theta) \frac{\|\Sigma_\mathbb{Q}\| + \lambda_\Theta}{\|\Sigma_\mathbb{Q}\|} \right), \qquad A_{\lambda_\Theta, \tau} := 8k_\alpha^2 \tau g_{\lambda_\Theta} \lambda_\Theta^{-\frac{d_\Theta}{2s_\Theta}},
$$

$$
L_{\lambda_\Theta} := \max \left\{ L, \lambda_\Theta^{\frac{1}{2} - \frac{d_\Theta}{4s_\Theta}} \left( \|\bar{F}_{\text{KQ}}\|_{L_\infty(\mathbb{Q})} + k_\alpha \|\bar{F}_{\text{KQ}}\|_{s_\Theta, 2} \right) \right\}.
$$

Applying Proposition 3 shows that, for $T > A_{\lambda_\Theta, \tau}$,

$$
\left\| \bar{F}_{\text{KQ}} - k(\cdot, \theta_{1:T}) \left( k_\Theta(\theta_{1:T}, \theta_{1:T}) + T\lambda_\Theta \boldsymbol{I}_T \right)^{-1} \hat{F}_{\text{KQ}}(\theta_{1:T}) \right\|_{L_2(\mathbb{Q})}^2
$$
$$
\leq \frac{576\tau^2}{T} \left( L^2 D_\Theta \lambda_\Theta^{-\frac{d_\Theta}{2s_\Theta}} + M_\Theta^2 \lambda_\Theta^{1-\frac{d_\Theta}{2s_\Theta}} \|\bar{F}_{\text{KQ}}\|_{s_\Theta, 2}^2 + 2M_\Theta^2 \frac{L_{\lambda_\Theta}^2}{T} \lambda_\Theta^{-\frac{d_\Theta}{2s_\Theta}} \right) + \|\bar{F}_{\text{KQ}}\|_{s_\Theta, 2}^2 \lambda_\Theta,
\tag{C.31}
$$

holds with probability at least $1 - 4e^{-\tau}$. We take $\lambda_\Theta \asymp T^{-2\frac{s_\Theta}{d_\Theta}} (\log T)^{\frac{2s_\Theta+2}{d_\Theta}}$, then similar to the derivations from (B.7),

$$
\lim_{T \to \infty} \frac{A_{\lambda_\Theta, \tau}}{T} \leq \lim_{T \to \infty} 16(\log T)^{-\frac{s_\Theta+1}{s_\Theta}} k_\alpha^2 \tau \log(T) = 0.
\tag{C.32}
$$

It means there exists a finite $T_0 > 0$ such that $T > A_{\lambda_\Theta, \tau}$ holds for any $T > T_0$. Notice that, with probability at least $1 - 4e^{-\tau}$,

$$
\|\bar{F}_{\text{KQ}}\|_{L_\infty(\mathbb{Q})} = \|\bar{F}_{\text{KQ}}\|_{L_\infty(\Theta)} \leq R_\Theta \|\bar{F}_{\text{KQ}}\|_{s_\Theta, 2} \leq R_\Theta C_6
\tag{C.33}
$$

based on (C.29) and the fact that $W_2^{s_\Theta}(\Theta) \hookrightarrow L_\infty(\Theta)$ with $\|W_2^{s_\Theta}(\Theta) \hookrightarrow L_\infty(\Theta)\| \le R_\Theta$, we have

$$L_{\lambda_\Theta} \le \max\{L, T^{-\frac{s_\Theta}{d_\Theta}+\frac{1}{2}}(\log T)^{\frac{s_\Theta+1}{2s_\Theta}\frac{2s_\Theta-d_\Theta}{d_\Theta}}(R_\Theta+M_\Theta)C_6\}$$
$$= \max\{C_7 N^{-\frac{s_\mathcal{X}}{d_\mathcal{X}}}(\log N)^{\frac{s_\mathcal{X}+1}{d_\mathcal{X}}}, T^{-\frac{s_\Theta}{d_\Theta}+\frac{1}{2}}(\log T)^{\frac{s_\Theta+1}{2s_\Theta}\frac{2s_\Theta-d_\Theta}{d_\Theta}}(R_\Theta+M_\Theta)C_6\}.$$

So the above (C.31) can be further upper bounded by

$$\le \frac{576\tau^2}{T}\left(C_7^2 N^{-2\frac{s_\mathcal{X}}{d_\mathcal{X}}}(\log N)^{\frac{2s_\mathcal{X}+2}{d_\mathcal{X}}}D_\Theta T(\log T)^{-\frac{s_\Theta+1}{s_\Theta}} + M_\Theta^2 T^{-\frac{2s_\Theta}{d_\Theta}+1}(\log T)^{\frac{s_\Theta+1}{s_\Theta}\frac{2s_\Theta-d_\Theta}{d_\Theta}}C_6^2\right)$$

$$+ \frac{576\tau^2}{T}\cdot 2M_\Theta^2 \frac{\max\left\{C_7^2 N^{-2\frac{s_\mathcal{X}}{d_\mathcal{X}}}(\log N)^{\frac{2s_\mathcal{X}+2}{d_\mathcal{X}}}, T^{-2\frac{s_\Theta}{d_\Theta}+1}(\log T)^{\frac{s_\Theta+1}{s_\Theta}\frac{2s_\Theta-d_\Theta}{d_\Theta}}(R_\Theta+M_\Theta)^2 C_6^2\right\}}{T}T(\log T)^{-\frac{s_\Theta+1}{s_\Theta}}$$

$$+ C_6^2 T^{-2\frac{s_\Theta}{d_\Theta}}(\log T)^{\frac{2s_\Theta+2}{d_\Theta}}$$

$$= 576\tau^2\left(C_7^2 N^{-2\frac{s_\mathcal{X}}{d_\mathcal{X}}}(\log N)^{\frac{2s_\mathcal{X}+2}{d_\mathcal{X}}}D_\Theta(\log T)^{-\frac{s_\Theta+1}{s_\Theta}} + M_\Theta^2 T^{-\frac{2s_\Theta}{d_\Theta}}(\log T)^{\frac{s_\Theta+1}{s_\Theta}\frac{2s_\Theta-d_\Theta}{d_\Theta}}C_6^2\right)$$

$$+ 576\tau^2\cdot 2M_\Theta^2\max\left\{C_7^2 N^{-2\frac{s_\mathcal{X}}{d_\mathcal{X}}}(\log N)^{\frac{2s_\mathcal{X}+2}{d_\mathcal{X}}}T^{-1}, T^{-\frac{2s_\Theta}{d_\Theta}}(\log T)^{\frac{s_\Theta+1}{s_\Theta}\frac{2s_\Theta-d_\Theta}{d_\Theta}}(R_\Theta+M_\Theta)^2 C_6^2\right\}\cdot(\log T)^{-\frac{s_\Theta+1}{s_\Theta}}$$

$$+ C_6^2 T^{-2\frac{s_\Theta}{d_\Theta}}(\log T)^{\frac{2s_\Theta+2}{d_\Theta}}$$

$$\overset{(i)}{\le} 576\tau^2 C_7^2 N^{-2\frac{s_\mathcal{X}}{d_\mathcal{X}}}(\log N)^{\frac{2s_\mathcal{X}+2}{d_\mathcal{X}}}D_\Theta + 576\tau^2 M_\Theta^2 T^{-\frac{2s_\Theta}{d_\Theta}}(\log T)^{\frac{2s_\Theta+2}{d_\Theta}}C_6^2$$

$$+ 576\tau^2\cdot 2M_\Theta^2 C_7^2 N^{-2\frac{s_\mathcal{X}}{d_\mathcal{X}}}(\log N)^{\frac{2s_\mathcal{X}+2}{d_\mathcal{X}}} + 576\tau^2\cdot 2M_\Theta^2 T^{-\frac{2s_\Theta}{d_\Theta}}(\log T)^{\frac{2s_\Theta+2}{d_\Theta}}(R_\Theta+M_\Theta)^2 C_6^2 + C_6^2 T^{-2\frac{s_\Theta}{d_\Theta}}(\log T)^{\frac{2s_\Theta+2}{d_\Theta}}$$

$$=: \tau^2\left(C_8^2 N^{-2\frac{s_\mathcal{X}}{d_\mathcal{X}}}(\log N)^{\frac{2s_\mathcal{X}+2}{d_\mathcal{X}}} + C_9^2 T^{-\frac{2s_\Theta}{d_\Theta}}(\log T)^{\frac{2s_\Theta+2}{d_\Theta}}\right).$$

$C_8, C_9$ are two constants independent of $N, T$. In $(i)$, we use $\max\{a_1, a_2\} \le a_1 + a_2$, we also use the following

$$(\log T)^{\frac{s_\Theta+1}{s_\Theta}\frac{2s_\Theta-d_\Theta}{d_\Theta}} \le (\log T)^{\frac{2s_\Theta+2}{d_\Theta}}, \quad (\log T)^{-\frac{s_\Theta+1}{s_\Theta}} \le 1.$$

Therefore, we have that,

$$\text{Stage II error} := \left\|\bar{F}_{\text{KQ}} - k(\cdot, \theta_{1:T})(k_\Theta(\theta_{1:T}, \theta_{1:T}) + T\lambda_\Theta \boldsymbol{I}_T)^{-1}\hat{F}_{\text{KQ}}(\theta_{1:T})\right\|_{L_2(\mathbb{Q})}$$
$$\le \tau\left(C_8 N^{-\frac{s_\mathcal{X}}{d_\mathcal{X}}}(\log N)^{\frac{s_\mathcal{X}+1}{d_\mathcal{X}}} + C_9 T^{-\frac{s_\Theta}{d_\Theta}}(\log T)^{\frac{s_\Theta+1}{d_\Theta}}\right), \tag{C.34}$$

holds with probability at least $1 - 8e^{-\tau}$.

**Combine stage I and stage II error**  Combining the stage I error of (C.18) and the stage II error of (C.34), we obtain

$$\left|I - \hat{I}_{\text{NKQ}}\right| \le \text{Stage I error} + \text{Stage II error}$$

$$\le C_3 N^{-\frac{s_\mathcal{X}}{d_\mathcal{X}}}(\log N)^{\frac{s_\mathcal{X}+1}{d_\mathcal{X}}} + \tau\left(C_8 N^{-\frac{s_\mathcal{X}}{d_\mathcal{X}}}(\log N)^{\frac{s_\mathcal{X}+1}{d_\mathcal{X}}} + C_9 T^{-\frac{s_\Theta}{d_\Theta}}(\log T)^{\frac{s_\Theta+1}{d_\Theta}}\right)$$

$$\le \tau\left((C_8 + C_3)N^{-\frac{s_\mathcal{X}}{d_\mathcal{X}}}(\log N)^{\frac{s_\mathcal{X}+1}{d_\mathcal{X}}} + C_9 T^{-\frac{s_\Theta}{d_\Theta}}(\log T)^{\frac{s_\Theta+1}{d_\Theta}}\right)$$

$$=: \tau\left(C_1 N^{-\frac{s_\mathcal{X}}{d_\mathcal{X}}}(\log N)^{\frac{s_\mathcal{X}+1}{d_\mathcal{X}}} + C_2 T^{-\frac{s_\Theta}{d_\Theta}}(\log T)^{\frac{s_\Theta+1}{d_\Theta}}\right),$$

holds with probability at least $1 - 8e^{-\tau}$. Here $C_1, C_2$ are two constants independent of $N, T$ so the proof concludes here.

# D. Multi-Level Nested Kernel Quadrature

In this section, we are going to introduce a novel method that combines nested kernel quadrature (NKQ) with multi-level construction as mentioned in Section 4.

### D.1. Multi-Level Monte Carlo for Nested Expectation

First, we briefly review multi-level Monte Carlo (MLMC) applied to nested expectations $I = \mathbb{E}_{\theta \sim \mathbb{Q}}[f(\mathbb{E}_{X \sim \mathbb{P}_\theta}[g(X, \theta)])]$ introduced in Section 9 of Giles (2015) and Giles and Goda (2019). At each level $\ell$, we are given $T_\ell$ samples $\theta_{1:T_\ell}$ sampled i.i.d from $\mathbb{Q}$ and we have $N_\ell$ samples $x^{(\theta_t)}_{1:N_\ell}$ sampled i.i.d from $\mathbb{P}_{\theta_t}$ for each $t = 1, \ldots, T_\ell$. The MLMC implementation is to construct an estimator $P_\ell$ at each level $\ell$ such that $I$ can be decomposed into the sum of $P_\ell$.

$$I \approx \mathbb{E}_{\theta \sim \mathbb{Q}}[P_L] = \mathbb{E}_{\theta \sim \mathbb{Q}}[P_0] + \sum_{\ell=1}^{L} \mathbb{E}_{\theta \sim \mathbb{Q}}[P_\ell - P_{\ell-1}], \qquad P_\ell := f\left(\frac{1}{N_\ell} \sum_{n=1}^{N_\ell} g\left(x_n^{(\theta)}, \theta\right)\right).$$

The estimator $Y_\ell$ for $\mathbb{E}_{\theta \sim \mathbb{Q}}[P_\ell - P_{\ell-1}]$ is

$$Y_\ell = \frac{1}{T_\ell} \sum_{t=1}^{T_\ell} \left\{ f\left(\frac{1}{N_\ell} \sum_{n=1}^{N_\ell} g\left(x_n^{(t)}, \theta_t\right)\right) - \frac{1}{2} f\left(\frac{1}{N_{\ell-1}} \sum_{n=1}^{N_{\ell-1}} g\left(x_n^{(t)}, \theta_t\right)\right) - \frac{1}{2} f\left(\frac{1}{N_{\ell-1}} \sum_{n=N_{\ell-1}+1}^{N_\ell} g\left(x_n^{(t)}, \theta_t\right)\right) \right\},$$

$$Y_0 := \frac{1}{T_0} \sum_{t=1}^{T_0} f\left(\frac{1}{N_0} \sum_{n=1}^{N_0} g\left(x_n^{(t)}, \theta_t\right)\right).$$

Compared with (3) in the main text, notice that here we use the 'antithetic' approach which further improves the performance of MLMC (Giles, 2015, Section 9). The MLMC estimator for nested expectation can be written as

$$\hat{I}_{\text{MLMC}} := \sum_{\ell=0}^{L} Y_\ell. \tag{D.35}$$

At each level $\ell$, the cost of $Y_\ell$ is $\mathcal{O}(N_\ell \times T_\ell)$ and the expected squared error $\mathbb{E}[(Y_\ell - \mathbb{E}_{\theta \sim \mathbb{Q}}[P_\ell - P_{\ell-1}])^2] = \mathcal{O}(N_\ell^{-2} \times T_\ell^{-1})$ provided that $f$ has bounded second order derivative (Giles, 2015, Section 9)[1]. Here the expectation is taken over the randomness of samples. So the total cost and expected absolute error of MLMC for nested expectation can be written as

$$\text{Cost} = \mathcal{O}\left(\sum_{\ell=0}^{L} N_\ell \times T_\ell\right), \qquad \mathbb{E}|I - \hat{I}_{\text{MLMC}}| = \mathcal{O}\left(\sum_{\ell=0}^{L} N_\ell^{-1} \times T_\ell^{-\frac{1}{2}}\right). \tag{D.36}$$

Theorem 1 of (Giles, 2015) shows that, in order to reach error threshold $\Delta$, one can take $N_\ell \propto 2^\ell$ and $T_\ell \propto 2^{-2\ell}\Delta^{-2}$. Therefore, one has $\mathbb{E}|I - \hat{I}_{\text{MLMC}}| = \mathcal{O}(\Delta)$ along with $\text{Cost} = \mathcal{O}(\Delta^{-2})$.

### D.2. Multi-Level Kernel Quadrature for Nested Expectation (MLKQ)

In this section, we present *multi-level kernel quadrature* applied to nested expectation (MLKQ). Note that MLKQ is different from the multi-level Bayesian quadrature proposed in Li et al. (2023) because our MLKQ is designed specifically for nested expectations. At each level $\ell$, we have $T_\ell$ samples $\theta_{1:T_\ell}$ sampled i.i.d from $\mathbb{Q}$ and we have $N_\ell$ samples $x^{(\theta_t)}_{1:N_\ell}$ sampled i.i.d from $\mathbb{P}_{\theta_t}$ for each $t = 1, \ldots, T_\ell$. Different from MLMC above, we define

$$I \approx \mathbb{E}_{\theta \sim \mathbb{Q}}[P_{\text{NKQ},L}] = \mathbb{E}_{\theta \sim \mathbb{Q}}[P_{\text{NKQ},0}] + \sum_{\ell=1}^{L} \mathbb{E}_{\theta \sim \mathbb{Q}}[P_{\text{NKQ},\ell} - P_{\text{NKQ},\ell-1}], \quad P_{\text{NKQ},\ell} := \mathbb{E}_{x^{(\theta)}_{1:N_\ell} \sim \mathbb{P}_\theta} f\left(\hat{J}_{\text{KQ}}\left(\theta; x^{(\theta)}_{1:N_\ell}\right)\right).$$

The estimator $Y_{\text{NKQ},\ell}$ for $\mathbb{E}_{\theta \sim \mathbb{Q}}[P_{\text{NKQ},\ell} - P_{\text{NKQ},\ell-1}]$ when $\ell \geq 1$ is the difference of two nested kernel quadrature estimator defined in (8).

$$Y_{\text{NKQ},\ell} := \mathbb{E}_{\theta \sim \mathbb{Q}}[k_\Theta(\theta, \theta_{1:T_\ell})](\boldsymbol{K}_{\Theta,T_\ell} + T_\ell \lambda_{\Theta,\ell} \boldsymbol{I}_{T_\ell})^{-1}\left(\hat{F}_{\text{KQ}}\left(\theta_{1:T_\ell}; x^{(\theta_{1:T_\ell})}_{1:N_\ell}\right) - \hat{F}_{\text{KQ}}\left(\theta_{1:T_\ell}; x^{(\theta_{1:T_\ell})}_{1:N_{\ell-1}}\right)\right)$$

where $\hat{F}_{\text{KQ}}(\theta_{1:T_\ell}; x^{(\theta_{1:T_\ell})}_{1:N_\ell})$ is a vectorized notation for $[\hat{F}_{\text{KQ}}(\theta_1; x^{(\theta_1)}_{1:N_\ell}), \ldots, \hat{F}_{\text{KQ}}(\theta_{T_\ell}; x^{(\theta_{T_\ell})}_{1:N_\ell})] \in \mathbb{R}^{T_\ell}$ and similarly for $\hat{F}_{\text{KQ}}(\theta_{1:T_\ell}; x^{(\theta_{1:T_\ell})}_{1:N_{\ell-1}})$. At level 0, $Y_{\text{NKQ},0} := \mathbb{E}_{\theta \sim \mathbb{Q}}[k_\Theta(\theta, \theta_{1:T_0})](\boldsymbol{K}_{\Theta,T_0} + T_0 \lambda_{\Theta,0} \boldsymbol{I}_{T_0})^{-1} \hat{F}_{\text{KQ}}(\theta_{1:T_0})$. The multi-level

---

[1]Section 9 of (Giles, 2015) uses variance $\mathbb{E}[Y_\ell^2]$, which is equivalent to the expected square error since $Y_\ell$ is an unbiased estimate of $\mathbb{E}_{\theta \sim \mathbb{Q}}[P_\ell - P_{\ell-1}]$.

nested kernel quadrature estimator is constructed as

$$\hat{I}_{\text{MLKQ}} := \sum_{\ell=0}^{L} Y_{\text{NKQ},\ell}.$$

Same as MLMC above, the cost of $Y_{\text{NKQ},\ell}$ is $\mathcal{O}(N_\ell \times T_\ell)$. The following theorem studies the error $|Y_{\text{NKQ},\ell} - \mathbb{E}_{\theta \sim \mathbb{Q}}[P_{\text{NKQ},\ell} - P_{\text{NKQ},\ell-1}]|$.

**Theorem 2.** *Let $\mathcal{X} = [0,1]^{d_\mathcal{X}}$ and $\Theta = [0,1]^{d_\Theta}$. At level $\ell \geq 1$, $\theta_1, \ldots, \theta_{T_\ell}$ are $T_\ell$ i.i.d. samples from $\mathbb{Q}$ and $x_1^{(t)}, \ldots, x_{N_\ell}^{(t)}$ are $N_\ell$ i.i.d. samples from $\mathbb{P}_{\theta_t}$ for all $t \in \{1, \cdots, T_\ell\}$. Both kernels $k_\mathcal{X}$ and $k_\Theta$ are Sobolev reproducing kernels of smoothness $s_\mathcal{X} > d_\mathcal{X}/2$ and $s_\Theta > d_\Theta/2$. Suppose the Assumptions (1), (2), (3), (4) in Theorem 1 hold. Suppose $2^{\frac{d_\mathcal{X}}{s_\mathcal{X}}} N_{\ell-1} > N_\ell > N_{\ell-1}$. Then, for sufficiently large $N_\ell \geq 1$ and $T_\ell \geq 1$, with $\lambda_{\mathcal{X},\ell} \asymp N_\ell^{-2\frac{s_\mathcal{X}}{d_\mathcal{X}}} \cdot (\log N_\ell)^{\frac{2s_\mathcal{X}+2}{d_\mathcal{X}}}$ and $\lambda_{\Theta,\ell} \asymp T_\ell^{-\frac{2s_\Theta}{2s_\Theta+d_\Theta}}$,*

$$\left| Y_{\text{NKQ},\ell} - \mathbb{E}_{\theta \sim \mathbb{Q}}[P_{\text{NKQ},\ell} - P_{\text{NKQ},\ell-1}] \right| \lesssim \tau \left( N_\ell^{-\frac{s_\mathcal{X}}{d_\mathcal{X}}} (\log N)^{\frac{s_\mathcal{X}+1}{d_\mathcal{X}}} \times T_\ell^{-\frac{s_\Theta}{2s_\Theta+d_\Theta}} \right)$$

*holds with probability at least $1 - 12e^{-\tau}$.*

The proof of the theorem is relegated to Appendix D.3.

Under Theorem 2, the expected error $\mathbb{E}[Y_{\text{NKQ},\ell} - \mathbb{E}_{\theta \sim \mathbb{Q}}[P_{\text{NKQ},\ell} - P_{\text{NKQ},\ell-1}]] = \tilde{\mathcal{O}}(N_\ell^{-\frac{s_\mathcal{X}}{d_\mathcal{X}}} \times T_\ell^{-\frac{s_\Theta}{2s_\Theta+d_\Theta}})$ based on Lemma 4, up to logarithm terms. Here, the expectation is taken over the randomness of samples. Therefore, similarly to (D.36), the total cost and expected absolute error of multi-level nested kernel quadrature can be written as

$$\text{Cost} = \mathcal{O}\left( \sum_{\ell=0}^{L} N_\ell \times T_\ell \right), \qquad \mathbb{E}\left|I - \hat{I}_{\text{MLKQ}}\right| = \tilde{\mathcal{O}}\left( \sum_{\ell=0}^{L} N_\ell^{-\frac{s_\mathcal{X}}{d_\mathcal{X}}} \times T_\ell^{-\frac{s_\Theta}{2s_\Theta+d_\Theta}} \right). \qquad (D.37)$$

If we take $N_\ell \propto 2^{\frac{d_\mathcal{X}}{s_\mathcal{X}}\ell} \Delta^{-\frac{d_\mathcal{X}}{2s_\mathcal{X}}}, T_\ell \propto 2^{-\frac{2s_\Theta+d_\Theta}{s_\Theta}\ell} \Delta^{-\frac{2s_\Theta+d_\Theta}{2s_\Theta}}$, then the error $\mathbb{E}\left|I - \hat{I}_{\text{MLKQ}}\right| = \tilde{\mathcal{O}}(\Delta)$ and the cost is

$$\sum_{\ell=0}^{L} N_\ell \times T_\ell = \left( \sum_{\ell=0}^{L} 2^{\frac{d_\mathcal{X}}{s_\mathcal{X}}\ell - \frac{2s_\Theta+d_\Theta}{s_\Theta}\ell} \right) \cdot \Delta^{-1-\frac{d_\mathcal{X}}{2s_\mathcal{X}}-\frac{d_\Theta}{2s_\Theta}} \leq \left( \sum_{\ell=0}^{L} 2^{\left(\frac{d_\mathcal{X}}{s_\mathcal{X}}-2\right)\ell} \right) \cdot \Delta^{-1-\frac{d_\mathcal{X}}{2s_\mathcal{X}}-\frac{d_\Theta}{2s_\Theta}} = \mathcal{O}(\Delta^{-1-\frac{d_\mathcal{X}}{2s_\mathcal{X}}-\frac{d_\Theta}{2s_\Theta}}).$$

Equivalently, to reach error $\mathcal{O}(\Delta)$, the cost is $\tilde{\mathcal{O}}(\Delta^{-1-\frac{d_\mathcal{X}}{2s_\mathcal{X}}-\frac{d_\Theta}{2s_\Theta}})$.

**Remark D.1** (Comparison of MLKQ and MLMC). *To reach a given threshold $\Delta$, the cost of MLKQ is $\tilde{\mathcal{O}}(\Delta^{-1-\frac{d_\mathcal{X}}{2s_\mathcal{X}}-\frac{d_\Theta}{2s_\Theta}})$, which is smaller than the cost of MLMC $\mathcal{O}(\Delta^{-2})$ when the problem has sufficient smoothness, i.e. when $\frac{d_\mathcal{X}}{s_\mathcal{X}} + \frac{d_\Theta}{s_\Theta} < 2$. If we compare (D.36) and (D.37), the superior performance of MLKQ can be explained by the faster rate of convergence in terms of $N_\ell$ at each level when $\frac{d_\mathcal{X}}{s_\mathcal{X}} \leq 1$. Nevertheless, we can see in (D.37) that the MLKQ rate at each level in terms of $T_\ell$ is $\mathcal{O}(T_\ell^{-\frac{s_\Theta}{2s_\Theta+d_\Theta}})$ which is slower than the MLMC rate $\mathcal{O}(T_\ell^{-\frac{1}{2}})$ in (D.36). An empirical study of MLKQ is included in Figure 6 which shows that MLKQ is better than MLMC in some settings but both are outperformed by NKQ by a huge margin. A more refined analysis of MLKQ is reserved for future work.*

### D.3. Proof of Theorem 2

The proof uses essentially the same analysis as in *Step Five* of Appendix C which translates $|Y_\ell - \mathbb{E}_{\theta \sim \mathbb{Q}}[P_\ell - P_{\ell-1}]|$ into the generalization error of kernel ridge regression. First, we know that by following the same derivations as in (C.29) that

$$\bar{F}_{\text{KQ},\ell}(\theta) := \mathbb{E}_{x_{1:N_\ell}^{(\theta)} \sim \mathbb{P}_\theta}\left[ \hat{F}_{\text{KQ}}\left(\theta; x_{1:N_\ell}^{(\theta)}\right) \right], \quad \bar{F}_{\text{KQ},\ell} \in W_2^{s_\Theta}(\Theta) \text{ and } \left\|\bar{F}_{\text{KQ},\ell}\right\|_{s_\Theta} \leq C_6,$$

$$\bar{F}_{\text{KQ},\ell-1}(\theta) := \mathbb{E}_{x_{1:N_{\ell-1}}^{(\theta)} \sim \mathbb{P}_\theta}\left[ \hat{F}_{\text{KQ}}\left(\theta; x_{1:N_{\ell-1}}^{(\theta)}\right) \right], \quad \bar{F}_{\text{KQ},\ell-1} \in W_2^{s_\Theta}(\Theta) \text{ and } \left\|\bar{F}_{\text{KQ},\ell-1}\right\|_{s_\Theta} \leq C_6.$$

Given i.i.d. observations $(\theta_1, \hat{F}_{\mathrm{KQ}}(\theta_1, x^{(\theta_1)}_{1:N_\ell}) - \hat{F}_{\mathrm{KQ}}(\theta_1, x^{(\theta_1)}_{1:N_{\ell-1}})), \ldots, (\theta_{T_\ell}, \hat{F}_{\mathrm{KQ}}(\theta_{T_\ell}, x^{(\theta_{T_\ell})}_{1:N_\ell}) - \hat{F}_{\mathrm{KQ}}(\theta_{T_\ell}, x^{(\theta_{T_\ell})}_{1:N_{\ell-1}}))$, the target of interest in the context of regression is the conditional mean, which in our case is precisely

$$\theta \mapsto \bar{F}_{\mathrm{KQ},\ell}(\theta) - \bar{F}_{\mathrm{KQ},\ell-1}(\theta) = \mathbb{E}_{x^{(\theta)}_{1:N_\ell} \sim \mathbb{P}_\theta}\left[\hat{F}_{\mathrm{KQ}}\left(\theta; x^{(\theta)}_{1:N_\ell}\right)\right] - \mathbb{E}_{x^{(\theta)}_{1:N_{\ell-1}} \sim \mathbb{P}_\theta}\left[\hat{F}_{\mathrm{KQ}}\left(\theta; x^{(\theta)}_{1:N_{\ell-1}}\right)\right].$$

Alternatively, $\hat{F}_{\mathrm{KQ}}(\theta, x^{(\theta)}_{1:N_\ell}) - \hat{F}_{\mathrm{KQ}}(\theta, x^{(\theta)}_{1:N_{\ell-1}}))$ can be viewed as noisy observation of the true function $\bar{F}_{\mathrm{KQ},\ell} - \bar{F}_{\mathrm{KQ},\ell-1}$ where the noise satisfied the following condition. For each $\theta \in \Theta$ and positive integer $m \geq 2$, similar to (C.30) we have,

$$\mathbb{E}\left|\left[\hat{F}_{\mathrm{KQ}}\left(\theta; x^{(\theta)}_{1:N_\ell}\right) - \hat{F}_{\mathrm{KQ}}\left(\theta; x^{(\theta)}_{1:N_{\ell-1}}\right)\right] - \left[\mathbb{E}_{x^{(\theta)}_{1:N_\ell} \sim \mathbb{P}_\theta} \hat{F}_{\mathrm{KQ}}\left(\theta; x^{(\theta)}_{1:N_\ell}\right) - \mathbb{E}_{x^{(\theta)}_{1:N_{\ell-1}} \sim \mathbb{P}_\theta} \hat{F}_{\mathrm{KQ}}\left(\theta; x^{(\theta)}_{1:N_{\ell-1}}\right)\right]\right|^m$$

$$\leq 2^m \mathbb{E}\left|\hat{F}_{\mathrm{KQ}}\left(\theta; x^{(\theta)}_{1:N_\ell}\right) - \mathbb{E}_{x^{(\theta)}_{1:N_\ell} \sim \mathbb{P}_\theta} \hat{F}_{\mathrm{KQ}}\left(\theta; x^{(\theta)}_{1:N_\ell}\right)\right|^m$$

$$+ 2^m \mathbb{E}\left|\hat{F}_{\mathrm{KQ}}\left(\theta; x^{(\theta)}_{1:N_{\ell-1}}\right) - \mathbb{E}_{x^{(\theta)}_{1:N_{\ell-1}} \sim \mathbb{P}_\theta} \hat{F}_{\mathrm{KQ}}\left(\theta; x^{(\theta)}_{1:N_{\ell-1}}\right)\right|^m$$

$$\lesssim N_\ell^{-m\frac{s_{\mathcal{X}}}{d_{\mathcal{X}}}} (\log N_\ell)^{m\frac{s_{\mathcal{X}}+1}{d_{\mathcal{X}}}} + N_{\ell-1}^{-m\frac{s_{\mathcal{X}}}{d_{\mathcal{X}}}} (\log N_{\ell-1})^{m\frac{s_{\mathcal{X}}+1}{d_{\mathcal{X}}}}$$

$$\lesssim N_\ell^{-m\frac{s_{\mathcal{X}}}{d_{\mathcal{X}}}} (\log N_\ell)^{m\frac{s_{\mathcal{X}}+1}{d_{\mathcal{X}}}},$$

where the second last inequality follows by replicating the same steps in (C.30), and the last inequality is true because $2^{d_{\mathcal{X}}/s_{\mathcal{X}}} N_{\ell-1} > N_\ell > N_{\ell-1}$. As a result, by replicating the steps for (C.31), we have

$$\left|Y_{\mathrm{NKQ},\ell} - \mathbb{E}_{\theta \sim \mathbb{Q}}[P_{\mathrm{NKQ},\ell} - P_{\mathrm{NKQ},\ell-1}]\right|^2$$

$$\leq \left\|\left(\bar{F}_{\mathrm{KQ},\ell} - \bar{F}_{\mathrm{KQ},\ell-1}\right) - k_\Theta\left(\cdot, \theta_{1:T_\ell}\right)\left(\boldsymbol{K}_{\Theta,T_\ell} + T_\ell \lambda_{\Theta,\ell} \boldsymbol{I}_{T_\ell}\right)^{-1}\left(\hat{F}_{\mathrm{KQ}}\left(\theta_{1:T_\ell}; x^{(\theta_{1:T_\ell})}_{1:N_\ell}\right) - \hat{F}_{\mathrm{KQ}}\left(\theta_{1:T_\ell}; x^{(\theta_{1:T_\ell})}_{1:N_{\ell-1}}\right)\right)\right\|^2_{L_2(\mathbb{Q})}$$

$$\lesssim \tau^2 \left(T_\ell^{-1} \lambda_{\Theta,\ell}^{-\frac{d_\Theta}{2s_\Theta}} N_\ell^{-\frac{2s_{\mathcal{X}}}{d_{\mathcal{X}}}} (\log N_\ell)^{\frac{2s_{\mathcal{X}}+2}{d_{\mathcal{X}}}} + \lambda_{\Theta,\ell}^{1-\frac{d_\Theta}{2s_\Theta}} T_\ell^{-1} \left\|\bar{F}_{\mathrm{KQ},\ell} - \bar{F}_{\mathrm{KQ},\ell-1}\right\|^2_{s_\Theta} + \lambda_{\Theta,\ell}^{-\frac{d_\Theta}{2s_\Theta}} T_\ell^{-1-\frac{2s_\Theta}{d_\Theta}} \left\|\bar{F}_{\mathrm{KQ},\ell} - \bar{F}_{\mathrm{KQ},\ell-1}\right\|^2_{s_\Theta}\right)$$

$$+ \left\|\bar{F}_{\mathrm{KQ},\ell} - \bar{F}_{\mathrm{KQ},\ell-1}\right\|^2_{s_\Theta} \lambda_{\Theta,\ell}, \tag{D.38}$$

holds with probability at least $1 - 4e^{-\tau}$. Next, we are going to upper bound $\left\|\bar{F}_{\mathrm{KQ},\ell} - \bar{F}_{\mathrm{KQ},\ell-1}\right\|_{s_\Theta}$. To this end, notice that

$$\left\|\bar{F}_{\mathrm{KQ},\ell} - \bar{F}_{\mathrm{KQ},\ell-1}\right\|^2_{s_\Theta} \leq 2 \left\|\bar{F}_{\mathrm{KQ},\ell} - F\right\|^2_{s_\Theta} + 2 \left\|\bar{F}_{\mathrm{KQ},\ell-1} - F\right\|^2_{s_\Theta}.$$

Using the same steps in (C.29) and (C.27) subsequently, we have

$$\left\|\bar{F}_{\mathrm{KQ},\ell} - F\right\|_{s_\Theta} \leq \left\|\hat{F}_{\mathrm{KQ}}\left(\cdot; x^{(\theta)}_{1:N_\ell}\right) - F\right\|_{s_\Theta} \cdot S_3^{N_\ell} \cdot \mathrm{Vol}(\mathcal{X})^{N_\ell} \lesssim \left\|\hat{J}_{\mathrm{KQ}}\left(\cdot; x^{(\theta)}_{1:N_\ell}\right) - J\right\|_{s_\Theta} \lesssim N_\ell^{-\frac{s_{\mathcal{X}}}{d_{\mathcal{X}}}} (\log N_\ell)^{\frac{s_{\mathcal{X}}+1}{d_{\mathcal{X}}}},$$

holds with probability at least $1 - 4e^{-\tau}$. Similarly, we have $\left\|\bar{F}_{\mathrm{KQ},\ell-1} - F\right\|_{s_\Theta} \lesssim N_{\ell-1}^{-\frac{s_{\mathcal{X}}}{d_{\mathcal{X}}}} (\log N_{\ell-1})^{\frac{s_{\mathcal{X}}+1}{d_{\mathcal{X}}}}$ holds with probability at least $1 - 4e^{-\tau}$. Consequently, we have $\|\bar{F}_{\mathrm{KQ},\ell} - \bar{F}_{\mathrm{KQ},\ell-1}\|_{s_\Theta} \lesssim N_\ell^{-\frac{s_{\mathcal{X}}}{d_{\mathcal{X}}}} (\log N_\ell)^{\frac{s_{\mathcal{X}}+1}{d_{\mathcal{X}}}}$ holds with probability at least $1 - 8e^{-\tau}$. Therefore, plugging it back to (D.38), we obtain

$$\left|Y_{\mathrm{NKQ},\ell} - \mathbb{E}_{\theta \sim \mathbb{Q}}[P_{\mathrm{NKQ},\ell} - P_{\mathrm{NKQ},\ell-1}]\right|^2$$

$$\lesssim \tau^2 \left(T_\ell^{-1} \lambda_{\Theta,\ell}^{-\frac{d_\Theta}{2s_\Theta}} N_\ell^{-\frac{2s_{\mathcal{X}}}{d_{\mathcal{X}}}} (\log N_\ell)^{\frac{2s_{\mathcal{X}}+2}{d_{\mathcal{X}}}} + \lambda_{\Theta,\ell}^{1-\frac{d_\Theta}{2s_\Theta}} T_\ell^{-1} N_\ell^{-\frac{2s_{\mathcal{X}}}{d_{\mathcal{X}}}} (\log N_\ell)^{\frac{2s_{\mathcal{X}}+2}{d_{\mathcal{X}}}}\right.$$

$$\left.+ \lambda_{\Theta,\ell}^{-\frac{d_\Theta}{2s_\Theta}} T_\ell^{-1-\frac{2s_\Theta}{d_\Theta}} N_\ell^{-\frac{2s_{\mathcal{X}}}{d_{\mathcal{X}}}} (\log N_\ell)^{\frac{2s_{\mathcal{X}}+2}{d_{\mathcal{X}}}} + N_\ell^{-\frac{2s_{\mathcal{X}}}{d_{\mathcal{X}}}} (\log N_\ell)^{\frac{2s_{\mathcal{X}}+2}{d_{\mathcal{X}}}} \lambda_{\Theta,\ell}\right),$$

holds with probability at least $1 - 12e^{-\tau}$. Therefore, by taking $\lambda_{\Theta,\ell} \asymp T_\ell^{-\frac{2s_\Theta}{2s_\Theta+d_\Theta}}$, we obtain with probability at least $1 - 8e^{-\tau}$,

$$\left|Y_{\mathrm{NKQ},\ell} - \mathbb{E}_{\theta \sim \mathbb{Q}}[P_{\mathrm{NKQ},\ell} - P_{\mathrm{NKQ},\ell-1}]\right| \lesssim \tau T_\ell^{-\frac{s_\Theta}{2s_\Theta+d_\Theta}} \times N_\ell^{-\frac{s_{\mathcal{X}}}{d_{\mathcal{X}}}} (\log N_\ell)^{\frac{s_{\mathcal{X}}+1}{d_{\mathcal{X}}}}.$$

The proof is concluded.

# E. Further Background and Auxiliary Lemmas

All the results in this section are existing results in the literature. We provide them here and prove some of them in the specific context of Sobolev spaces explicitly for the convenience of the reader.

**More technical notions of Sobolev spaces and the Sobolev embedding theorem** In the main text, we provide in (4) the standard definition of Sobolev spaces $W_2^s(\mathcal{X})$ when $s \in \mathbb{N}$. Actually, Sobolev spaces $W_2^s(\mathcal{X})$ can be extended to $s$ that are positive real numbers. Such extension could be realized through real interpolation spaces (see (Bennett and Sharpley, 1988, Definition 1.7)), $W_2^s(\mathcal{X}) := [W_2^k(\mathcal{X}), W_2^{k+1}(\mathcal{X})]_{r,2}$ where $k \in \mathbb{N}, s \in (k, k+1), r = s - \lfloor s \rfloor$.[2] Actually, such interpolation relations hold for any $0 \leq s, t$ and $0 < r < 1$ (Adams and Fournier, 2003, Section 7.32),

$$W_2^k(\mathcal{X}) = \left[ W_2^s(\mathcal{X}), W_2^t(\mathcal{X}) \right]_{r,2}, \quad k = (1-r)s + rt. \tag{E.39}$$

A special case of the above relation is $W_2^s(\mathcal{X}) = [L_2(\mathcal{X}), W_2^t(\mathcal{X})]_{s/t,2}$.

The Sobolev embedding theorem (Adams and Fournier, 2003), when applied to $W_2^s(\mathcal{X})$, states that if $s > \frac{d}{2}$ (where $d$ is the dimension of $\mathcal{X}$), then $W_2^s(\mathcal{X})$ can be continuously embedded into $C^0(\mathcal{X})$, the space of continuous and bounded functions. In other words, for every equivalence class $[f] \in W_2^s(\mathcal{X})$, there exists a unique continuous and bounded representative $f \in C^0(\mathcal{X})$, and the embedding map $I : W_2^s(\mathcal{X}) \to C^0(\mathcal{X})$, defined by $I([f]) = f$, is continuous. This continuous embedding $I$ can be written as $W_2^s(\mathcal{X}) \hookrightarrow C^0(\mathcal{X})$. Since every continuous linear operator is bounded, we have $\|W_2^s(\mathcal{X}) \hookrightarrow C^0(\mathcal{X})\|$ bounded by a constant that only depends on $s, \mathcal{X}$.

**More technical notions of reproducing kernel Hilbert spaces (RKHSs)** For bounded kernels, $\sup_{x \in \mathcal{X}} k(x, x) \leq \kappa$, its associated RKHS $\mathcal{H}$ can be canonically injected into $L_2(\pi)$ using the operator $\iota_\pi : \mathcal{H} \to L_2(\pi), f \mapsto f$ with its adjoint $\iota_\pi^* : L_2(\pi) \to \mathcal{H}$ given by $\iota_\pi^* f(\cdot) = \int k(x, \cdot) f(x) d\pi(x)$. $\iota_\pi$ and its adjoint can be composed to form a $L_2(\pi)$ endomorphism $\mathcal{T}_\pi := \iota_\pi \iota_\pi^*$ called the *integral operator*, and a $\mathcal{H}$ endomorphism

$$\Sigma_\pi := \iota_\pi^* \iota_\pi = \int k(\cdot, x) \otimes k(\cdot, x) d\pi(x), \tag{E.40}$$

(where $\otimes$ denotes the tensor product such that $(a \otimes b)c := \langle b, c \rangle_{\mathcal{H}} a$ for $a, b, c \in \mathcal{H}$) called the *covariance operator*. Both $\Sigma_\pi$ and $\mathcal{T}_\pi$ are compact, positive, self-adjoint, and they have the same eigenvalues $\varrho_1 \geq \cdots \varrho_i \geq \cdots \geq 0$. Please refer to Section 2 of Chen et al. (2024a) for more details.

**Lemma 1** (Effective dimension $\mathcal{N}(\lambda)$). *Let $\mathcal{X} \subset \mathbb{R}^d$ be a compact domain, $\pi$ be a probability measure on $\mathcal{X}$ with density $p : \mathcal{X} \to \mathbb{R}$. $k : \mathcal{X} \times \mathcal{X} \to \mathbb{R}$ is a Sobolev reproducing kernel of order $s > \frac{d}{2}$. $\{\varrho_m\}_{m \geq 0}$ are the eigenvalues of the integral operator $\mathcal{T}_\pi$. Define the effective dimension $\mathcal{N} : (0, \infty) \to [0, \infty)$ as $\mathcal{N}(\lambda) := \sum_{m \geq 1} \frac{\varrho_m}{\varrho_m + \lambda}$. If $p(x) \geq G > 0$ for any $x \in \mathcal{X}$, then $\mathcal{N}(\lambda) \leq D\lambda^{-\frac{d}{2s}}$ with constant $D$ that only depends on $G$ and $\mathcal{X}$.*

*Proof.* First, we study the asymptotic behavior of the eigenvalues $(\varrho_m)_{m \geq 1}$ of the integral operator $\mathcal{T}_\pi$. Theorem 15 of (Steinwart et al., 2009) shows that the eigenvalues $\varrho_m$ share the same asymptotic decay rate as the squares of the entropy number $e_m^2(I_\pi)$ of the embedding $I_\pi : W_2^s(\mathcal{X}) \to L_2(\pi)$. Denote $\mathcal{L}_{\mathcal{X}}$ as the Lebesgue measure on $\mathcal{X}$. Since $p(x) \geq G$ for any $x \in \mathcal{X}$, we know $\frac{d\mathcal{L}_{\mathcal{X}}}{d\pi} \leq G^{-1}\mathrm{Vol}(\mathcal{X})^{-1}$ so $\|L_2(\pi) \hookrightarrow L_2(\mathcal{X})\| \leq G^{-1}\mathrm{Vol}(\mathcal{X})^{-1}$, and consequently we have from Equation (A.38) of Steinwart (2008) that

$$e_m(I_\pi) \leq e_m(I_{\mathcal{L}_{\mathcal{X}}}) \|L_2(\pi) \hookrightarrow L_2(\mathcal{X})\| \leq G^{-1}\mathrm{Vol}(\mathcal{X})^{-1} e_m(I_{\mathcal{L}_{\mathcal{X}}}).$$

Moreover, (Edmunds and Triebel, 1996, Equation 4 on p. 119) shows that the entropy number $e_m(I_{\mathcal{L}_{\mathcal{X}}}) \leq \tilde{c}m^{-s/d}$ for some constant $\tilde{c}$, so we have $e_m(I_\pi) \leq G^{-1}\mathrm{Vol}(\mathcal{X})^{-1}\tilde{c}m^{-s/d}$ and consequently we have $\varrho_m \asymp e_m^2(I_\pi) \leq G^{-2}\mathrm{Vol}(\mathcal{X})^{-2}\tilde{c}^2 m^{-2s/d} =: c_2 m^{-2s/d}$.

Next, we have

$$\sum_{m \geq 1} \frac{\varrho_m}{\varrho_m + \lambda} \leq \sum_{m \geq 1} \frac{1}{1 + \lambda c_2^{-1} m^{2s/d}} \leq \int_0^\infty \frac{c_2}{c_2 + \lambda t^{2s/d}} dt = \lambda^{-\frac{d}{2s}} \int_0^\infty \frac{c_2}{c_2 + \tau^{2s/d}} d\tau$$

---

[2]Strictly speaking, the definition of (4) extended to real numbers $s$ actually corresponds to the complex interpolation space of Sobolev spaces. Fortunately, complex interpolation spaces and real interpolation spaces coincide under Hilbert spaces (Hytonen et al., 2016, Corollary C.4.2), which is precisely our setting since $p = 2$.

$$= \lambda^{-\frac{d}{2s}} \int_0^\infty \frac{1}{1 + \left(\tau c_2^{-\frac{d}{2s}}\right)^{\frac{2s}{d}}} d\tau = \lambda^{-\frac{d}{2s}} \int_0^\infty \frac{1}{1 + u^{\frac{2s}{d}}} c_2^{\frac{d}{2s}} du = \lambda^{-\frac{d}{2s}} c_2^{\frac{d}{2s}} \frac{\frac{\pi d}{2s}}{\sin\left(\frac{\pi d}{2s}\right)} =: D\lambda^{-\frac{d}{2s}},$$

where $D$ is a constant that depends on the domain $\mathcal{X}$ and $G$. $\qquad\square$

**Lemma 2.** *Let $\mathcal{X} \subset \mathbb{R}^d$ be a compact domain, $\pi$ be a probability measure on $\mathcal{X}$ with density $p : \mathcal{X} \to \mathbb{R}$. $k : \mathcal{X} \times \mathcal{X} \to \mathbb{R}$ is a Sobolev reproducing kernel of order $s > \frac{d}{2}$. $\{\varrho_m, e_m\}_{m \geq 0}$ are the eigenvalues and eigenfunctions of the integral operator $\mathcal{T}_\pi$. If there exists $G_0, G_1 > 0$ such that $G_0 \leq p(x) \leq G_1$ for any $x \in \mathcal{X}$, then*

$$k_\alpha := \sup_{x \in \mathcal{X}} \sum_{m \geq 1} \varrho_m^\alpha e_m^2(x) \leq M, \tag{E.41}$$

*holds for any $\frac{d}{2s} < \alpha$. Here, $M$ is a constant that depends on $\mathcal{X}$ and $G_1, G_0$.*

*Proof.* If $t > \frac{d}{2}$, $W_2^t(\mathcal{X})$ can be continuously embedded into $L_\infty(\mathcal{X})$ the space of bounded functions (Adams and Fournier, 2003, Case A, Theorem 4.12). Hence, the operator $W_2^s(\mathcal{X}) \hookrightarrow L_\infty(\mathcal{X})$ is a continuous linear operator between two normed vector spaces, hence a bounded operator. And $L_2(\pi)$ is norm equivalent to $L_2(\mathcal{X})$ because $G_0 \leq p(x) \leq G_1$ for any $x \in \mathcal{X}$. Notice that $k_\alpha$ defined here is exactly $\|k_\nu^\alpha\|_\infty$ defined in Equation 16 of (Fischer and Steinwart, 2020), so we know from Theorem 9 of Fischer and Steinwart (2020) that

$$\sup_{x \in \mathcal{X}} \sum_{m \geq 1} \varrho_m^\alpha e_i^2(x) = \left\| [L_2(\pi), W_2^s(\mathcal{X})]_{\alpha, 2} \hookrightarrow L_\infty(\mathcal{X}) \right\|.$$

Notice that $[L_2(\pi), W_2^s(\mathcal{X})]_{\alpha, 2} \cong [L_2(\mathcal{X}), W_2^s(\mathcal{X})]_{\alpha, 2} \cong W_2^{s\alpha}(\mathcal{X})$, and notice the fact that $W_2^{s\alpha}(\mathcal{X}) \hookrightarrow L_\infty(\mathcal{X})$ for any $s\alpha > \frac{d}{2}$, the right hand side of the above equation is bounded. Therefore, we have (E.41) holds for any $\frac{d}{2s} < \alpha$. $\qquad\square$

**Lemma 3.** *Let $\mathcal{X} \subset \mathbb{R}^d$ be a bounded domain with Lipschitz continuous boundary and $W_2^s(\mathcal{X})$ be a Sobolev space with $s > \frac{d}{2}$. If functions $f : \mathcal{X} \to \mathbb{R}$ and $g : \mathcal{X} \to \mathbb{R}$ lie in $W_2^s(\mathcal{X})$, then their product $f \cdot g$ also lies in $W_2^s(\mathcal{X})$ and satisfies $\|f \cdot g\|_s \leq \|f\|_s \|g\|_s$.*

*Proof.* This is Theorem 7.4 of Behzadan and Holst (2021) with $s_1 = s_2 = s$ and $p_1 = p_2 = 2$. $\qquad\square$

**Lemma 4.** *For a positive valued random variable $R$, and $c > 0$ such that $\mathbb{P}(R \leq c\tau) \geq 1 - \exp(-\tau)$ for any positive $\tau$, it holds that $\mathbb{E}[R^m] \leq c_o m!$ for all integers $m \geq 1$. $c_o$ is some constant that only depends on $c, m$.*

*Proof.* Notice that $R$ is essentially a sub-exponential random variable. Since a sub-exponential random variable is equivalent to the square root of a sub-Gaussian random variable, from Proposition 2.5.2 of Vershynin (2018), we have $\mathbb{E}[R^m] = \mathbb{E}[\sqrt{R}^{2m}] \leq 2c_o \Gamma(m + 1) = 2c_o m!$. Here $\Gamma$ denotes the gamma function and $c_o$ is some constant that only depends on $c, m$. $\qquad\square$

**Lemma 5.** *For a mapping $F$ from a compact domain $\mathcal{X} \subset \mathbb{R}^d$ to a Hilbert space $H$, given a measure $\mu$ on $\mathcal{X}$, if $F$ is $\mu$-Bochner integrable, then $\int F(x) d\mu(x) \in H$ and additionally $\|\int F(x) d\mu(x)\|_H \leq \int \|F(x)\|_H d\mu(x)$.*

*Proof.* This is Definition A.5.20 of Steinwart (2008). $\qquad\square$

## F. Additional Experimental Details

### F.1. "Change of Variable" Trick for Kernel Quadrature

In the main text, we have shown that the two major bottlenecks of KQ/NKQ are:

- The closed-form KME $\mathbb{E}_{X \sim \mathbb{P}}[k(X, x)]$.
- The $\mathcal{O}(N^3)$ computational cost of inverting the Gram matrix $k(x_{1:N}, x_{1:N})$.

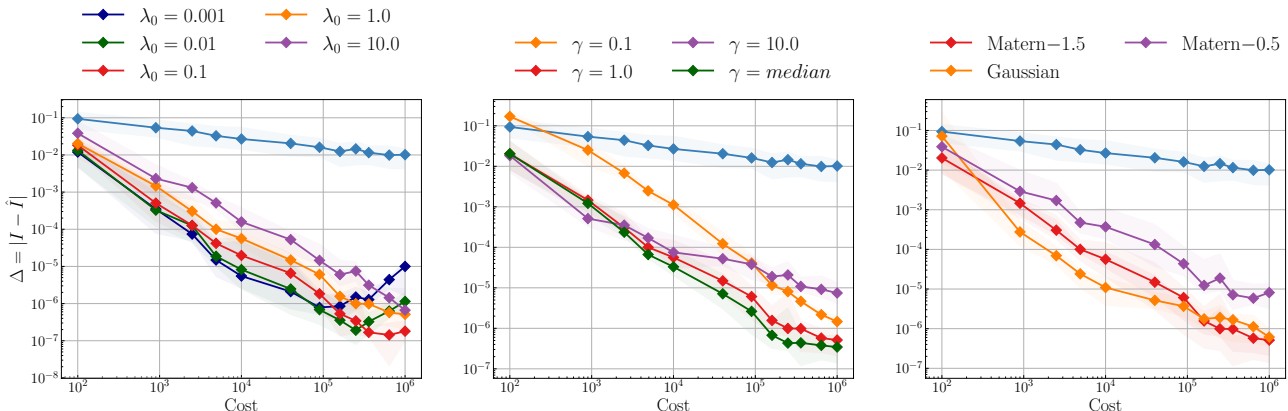

*Figure 5.* Further ablation studies in the synthetic experiment. **Left:** NKQ with different proportionality coefficients $\lambda_0$ for regularization parameter $\lambda_\mathcal{X}, \lambda_\Theta$. **Middle:** NKQ with different kernel lengthscales $\gamma$ in both stages. **Right:** NKQ with different kernels in both stages. The nested Monte Carlo (NMC) in blue is presented as a benchmark in all figures.

Fortunately, both two challenges can be solved with the "change of variable" trick. Here, we only present the idea for KQ but the same holds for NKQ in both stages.

The integral of interest is $I = \int_\mathcal{X} h(x)\mathbb{P}(dx)$. Suppose we can find a continuous transformation $\Phi$ such that $X = \Phi(U)$, where $U \sim \mathbb{U}$ is another random variable which is easy to sample from. Then the integral $I$ can be equivalently expressed as $I = \int_\mathcal{U} h(\Phi(u))d\mathbb{U}(u)$, by a direct application of change of variables theorem (Section 8.2 of (Stirzaker, 2003)). Now the integrand changes from $h : \mathcal{X} \to \mathbb{R}$ to $h \circ \Phi : \mathcal{U} \to \mathbb{R}$ and the kernel quadrature estimator becomes

$$\hat{I}_{\mathsf{KQ}} = \mathbb{E}_{U \sim \mathbb{U}}[k_\mathcal{U}(U, u_{1:N})] \left(k_\mathcal{U}(u_{1:N}, u_{1:N}) + N\lambda \boldsymbol{I}_N\right)^{-1} (h \circ \Phi)(u_{1:N}).$$

Here $k_\mathcal{U}$ is a reproducing kernel on $\mathcal{U}$. Since $\mathbb{U}$ is a simple probability distribution, we can find its closed-form KME in Table 1 in Briol et al. (2019) or the `ProbNum` package (Wenger et al., 2021), which addresses the first challenge. Additionally, notice that both the Gram matrix $k(u_{1:N}, u_{1:N})$ and the KME $\mathbb{E}_{U \sim \mathbb{U}}[k(U, u_{1:N})]$ are independent of $h$ and $\Phi$, so the KQ weights $w_{1:N}^{\mathsf{KQ}} = \mathbb{E}_{U \sim \mathbb{U}}[k(U, u_{1:N})] \left(k(u_{1:N}, u_{1:N}) + N\lambda \boldsymbol{I}_N\right)^{-1}$ can be pre-computed and stored. As a result, KQ becomes a simple weighted average of function evaluations $\sum_{i=1}^N w_i^{\mathsf{KQ}} h(x_i)$. Therefore, the computational cost reduces to linear cost $\mathcal{O}(N)$ and hence the second challenge is addressed. The downside of the "change of variable" trick is that the Sobolev smoothness of $h \circ \Phi : \mathcal{U} \to \mathbb{R}$ is unclear when $\Phi$ is not smooth, so we lose the theoretical convergence rate from Theorem 1.

### F.2. Synthetic Experiment

**Assumptions from Theorem 1** We would like to check whether the assumptions made in Theorem 1 hold in this synthetic experiment. Recall that we use both $k_\mathcal{X}$ and $k_\Theta$ to be Matérn-3/2 kernels so we need to verify Assumptions (1) — (4) with $s_\Theta = s_\mathcal{X} = 2$.

1. Both distributions $\mathbb{P}_\theta$ and $\mathbb{Q}$ are uniform distributions over $[0, 1]$, so Assumption (1) is satisfied.
2. $D_\theta^\beta g(\cdot, \theta) \in W_2^2(\mathcal{X})$ and $D_\theta^\beta p(\cdot, \theta) \in L_2(\mathcal{X})$ for $\beta = 0, 1, 2$ so Assumption (2) is satisfied.
3. Both $g(x, \cdot), p(x, \cdot) \in W_2^2(\Theta)$ so Assumption (3) is satisfied.
4. $f \in C^3(\mathbb{R})$ so Assumption (4) is satisfied.

The synthetic problem can be modified to have higher dimensions $d$. In this synthetic experiment, we set both $d_\mathcal{X} = d_\Theta = d$. For $a = [a_1, \ldots, a_d]^\top \in \mathbb{R}^d$, define $\|a\|_b = (\sum_{i=1}^d a_i^b)^{1/b}$.

$$x \sim \mathrm{U}[0, 1]^d, \quad \theta \sim \mathrm{U}[0, 1]^d, \quad g(x, \theta) = \|x\|_{2s}^{2.5} + \|\theta\|_2^{2.5}, \quad f(z) = z^2, \tag{F.42}$$

The true value of the nested expectation can be computed in closed-form: $I = \frac{16}{49}d^2 + \frac{25}{294}d$. In Figure 2, we study the mean absolute error of NMC and NKQ as dimension $d$ grows. We see that NKQ outperforms NMC by a huge margin in low dimensions, but the performance gap closes down in higher dimensions, which is expected because the rate proved in

Corollary 1 is $\mathcal{O}(\Delta^{-\frac{d_\mathcal{X}}{s_\mathcal{X}} - \frac{d_\Theta}{s_\Theta}})$ which becomes larger when dimension increases yet the smoothness of the problem remains the same.

In Figure 5, we conduct a series of ablation studies on the hyperparameter of NKQ in the synthetic experiment. Although Theorem 1 suggests choosing the regularization parameters $\lambda_\mathcal{X}, \lambda_\Theta$ that are proportionate to $N^{-2\frac{s_\mathcal{X}}{d_\mathcal{X}}}$ and $T^{-2\frac{s_\Theta}{d_\Theta}}$ respectively, it is unclear in practice how to pick the exact proportionality coefficients $\lambda_0$. Figure 5 **Left** shows that $\lambda_0 = 1.0$ and $\lambda_0 = 0.1$ give the best performances, while using $\lambda_0$ too big ($\lambda_0 = 10.0$) suffers from slower convergence rate and using $\lambda_0$ too small ($\lambda_0 = 0.01, 0.001$) causes numerical issues when $N, T$ become large. Figure 5 **Middle** shows that kernel lengthscale, if too big ($\gamma = 10.0$) or too small ($\gamma = 1.0$), would result in worse performance for NKQ and that the widely-used median heuristic is good enough to select a satisfying lengthscale. Figure 5 **Right** shows that NKQ with Matérn-3/2 kernels has better performance than with Matérn-1/2 kernels, which agrees with Theorem 1 indicating that it is preferable to use Sobolev kernels with the highest permissible orders of smoothness. Interestingly, we see that NKQ with Gaussian kernels has similar performance as with Matérn-3/2 kernels. Similar phenomenon have been shown both theoretically and empirically that kernel ridge regression with Gaussian kernels are optimal in learning Sobolev space functions when the lengthscales are chosen appropriately (Hang and Steinwart, 2021; Eberts and Steinwart, 2013).

### F.3. Risk Management in Finance

In this experiment, we consider specifically an asset whose price $S(\tau)$ at time $\tau$ follows the Black-Scholes formula $S(\tau) = S_0 \exp\left(\sigma W(\tau) - \sigma^2 \tau / 2\right)$ for $\tau \geq 0$, where $\sigma$ is the underlying volatility, $S_0$ is the initial price and $W$ is the standard Brownian motion. The financial derivative we are interested in is a butterfly call option whose payoff at time $\tau$ can be expressed as $\psi(S(\tau)) = \max(S(\tau) - K_1, 0) + \max(S(\tau) - K_2, 0) - 2\max(S(\tau) - (K_1 + K_2)/2, 0)$. We follow the setting in (Alfonsi et al., 2021; 2022; Chen et al., 2024b) assuming that a shock occur at time $\eta$, at which time the option price is $S(\eta) = \theta$, and this shock multiplies the option price by $1 + s$. The option price at maturity time $\zeta$ is denoted as $S(\zeta) = x$. To summarize, the expected loss caused by the shock can be expressed as the following nested expectation:

$$I = \mathbb{E}[f(J(\theta))], \quad f(J(\theta)) = \max(J(\theta), 0), \quad J(\theta) = \int_0^\infty g(x)\mathbb{P}_\theta(dx), \quad g(x) = \psi(x) - \psi((1+s)x).$$

Following the setting in (Alfonsi et al., 2021; 2022; Chen et al., 2024b), we consider the initial price $S_0 = 100$, the volatility $\sigma = 0.3$, the strikes $K_1 = 50, K_2 = 150$, the option maturity $\zeta = 2$ and the shock happens at $\eta = 1$ with strength $s = 0.2$. The option price at which the shock occurs are $\theta_{1:T}$ sampled from the log normal distribution deduced from the Black-Scholes formula $\theta_{1:T} \sim \mathbb{Q} = \text{Lognormal}(\log S_0 - \frac{\sigma^2}{2}\eta, \sigma^2\eta)$. Then $x_{1:N}^{(t)}$ are sampled from another log normal distribution also deduced from the Black-Scholes formula $x_{1:N}^{(t)} \sim \mathbb{P}_{\theta_t} = \text{Lognormal}(\log \theta_t - \frac{\sigma^2}{2}(\zeta - \eta), \sigma^2(\zeta - \eta))$ for $t = 1, \ldots, T$.

In this experimental setting, although both $g$ only depends on $x$ and it is a combination of piece-wise linear functions so $g \in W_2^1(\mathcal{X})$. The probability density function of $\mathbb{P}_\theta$ is infinitely times differentiable

Notice that log normal distribution $\text{LogNormal}(\bar{m}, \bar{\sigma}^2)$ can be expressed as the following transformation from uniform distribution over $[0, 1]$.

$$u \sim U[0, 1], \quad \exp(\Psi^{-1}(u)\bar{\sigma} + \bar{m}) \sim \text{LogNormal}(\bar{m}, \bar{\sigma}^2).$$

Here, $\Psi^{-1}$ is the inverse cumulative distribution function of a standard normal distribution. Therefore, we can use the "change of variables" trick from Appendix F.1 such that we have closed-form KME against uniform distribution from *Probnum* (Wenger et al., 2021), and also the computational complexity of NKQ becomes $\mathcal{O}(N \times T)$. Although $\Psi^{-1}$ is infinitely times differentiable, we still use Matérn-1/2 kernels in both stages to be conservative of the smoothness of the integrand after applying the "change of variables" trick.

### F.4. Health Economics

In the medical world, it is important to compare the cost and the relative advantages of conducting extra medical experiments. The expected value of partial perfect information (EVPPI) quantifies the expected gain from conducting extra experiments to obtain precise knowledge of some unknown variables (Brennan et al., 2007):

$$\text{EVPPI} = \mathbb{E}\left[\max_c J_c(\theta)\right] - \max_c \mathbb{E}\left[J_c(\theta)\right], \quad J_c(\theta) = \int_\mathcal{X} g_c(x, \theta)\mathbb{P}_\theta(dx)$$

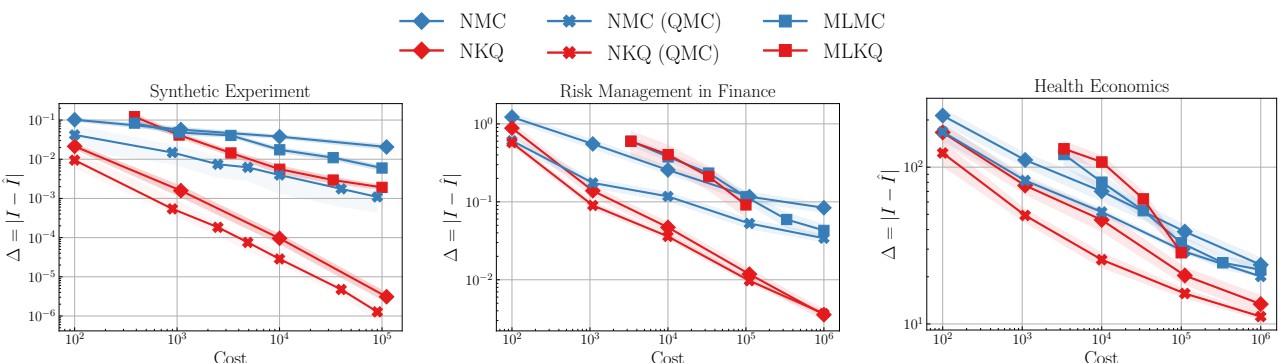

*Figure 6.* Comparison of all the methods including MLKQ on the synthetic experiment (**Left**), risk management in finance (**Middle**) and health economics (**Right**).

where $c \in \mathcal{C}$ is a set of potential treatments and $g_c$ measures the potential outcome of treatment $c$. EVPPI consists of $|\mathcal{C}|+1$ nested expectations.

We adopt the same experimental setup as delineated in (Giles and Haji-Ali, 2019), wherein $x$ and $\theta$ have a joint 19-dimensional Gaussian distribution, meaning that the conditional distribution $\mathbb{P}_\theta$ is also Gaussian. The specific meanings of all $x$ and $\theta$ are outlined in Table 2. All these variables are independent except that $\theta_1, \theta_2, x_6, x_{14}$ are pairwise correlated with a correlation coefficient 0.6. We are interested in estimating the EVPPI of a binary decision-making problem ($\mathcal{C} = \{1, 2\}$) with $g_1(x, \theta) = 10^4(\theta_1 x_5 x_6 + x_7 x_8 x_9) - (x_1 + x_2 x_3 x_4)$ and $g_2(x, \theta) = 10^4(\theta_2 x_{13} x_{14} + x_{15} x_{16} x_{17}) - (x_{10} + x_{11} x_{12} x_4)$. The ground truth EVPPI under this setting is $I = 538$ provided in (Giles and Goda, 2019).

For estimating $I_1$ with NKQ, we select $k_\mathcal{X}$ to be Gaussian kernel and $k_\Theta$ to be Matérn-1/2 kernel, because $I_1$ contains a *max* function which breaks the smoothness so we use Matérn-1/2 kernel to be conservative. For estimating $I_{2,c}$ with NKQ, we select both to be Gaussian kernels because both $g_1, g_2$ and the probability densities are all infinitely times continuously differentiable. We have access to the closed-form KME for both Matérn-1/2 and Gaussian kernels under a Gaussian distribution from *Probnum* (Wenger et al., 2021).

### F.5. Bayesian Optimization

For NKQ, we use the change of variable trick such that the Gaussian distribution of $f_{|\mathcal{D}_\mathcal{S}}(z_1, z_2)$ after $\mathcal{S}$ iterations can be expressed as the pushforward of a fixed uniform distribution $\mathbb{U}$ over $[0, 1]^2$ through a continuous mapping $\Phi_\mathcal{S}$. As a result, the KQ weights $\mathbb{E}_{U \sim \mathbb{U}}[k_\mathcal{U}(U, u_{1:N})](k_\mathcal{U}(u_{1:N}, u_{1:N}) + N\lambda \boldsymbol{I}_N)^{-1}$ become independent of $\mathcal{S}$, and can therefore be precomputed and stored in advance. We apply this change-of-variable trick to both Stage I and Stage II KQ steps in our NKQ algorithm, resulting in an overall $\mathcal{O}(N \times T)$ computational cost at each iteration, matching that of NMC.

The formulas of the synthetic `Dropwave`, `Ackley`, and `Cosine8` functions are provided below:

$$f_{\text{Dropwave}}(x, y) = -\frac{1 + \cos\left(12\sqrt{x^2 + y^2}\right)}{0.5(x^2 + y^2) + 2}, \quad (x, y) \in [-5.12, 5.12]^2,$$

$$f_{\text{Ackley}}(x) = -20\exp(-0.2\|x\|) - \exp\left(\frac{1}{2}\sum_{i=1}^{2}\cos(2\pi x_i)\right) + 20 + \exp(1), \quad x \in [-32.768, 32.768]^2$$

$$f_{\text{Cosine 8}}(x) = \sum_{i=1}^{8}\cos(5\pi x_i), \quad x \in [-1, 1]^8.$$

| Variables | Mean | Std | Meaning |
|:---:|:---:|:---:|:---:|
| $X_1$ | 1000 | 1.0 | Cost of treatment |
| $X_2$ | 0.1 | 0.02 | Probability of admissions |
| $X_3$ | 5.2 | 1.0 | Days of hospital |
| $X_4$ | 400 | 200 | Cost per day |
| $X_5$ | 0.3 | 0.1 | Utility change if response |
| $X_6$ | 3.0 | 0.5 | Duration of response |
| $X_7$ | 0.25 | 0.1 | Probability of side effects |
| $X_8$ | -0.1 | 0.02 | Change in utility if side effect |
| $X_9$ | 0.5 | 0.2 | Duration of side effects |
| $X_{10}$ | 1500 | 1.0 | Cost of treatment |
| $X_{11}$ | 0.08 | 0.02 | Probability of admissions |
| $X_{12}$ | 6.1 | 1.0 | Days of hospital |
| $X_{13}$ | 0.3 | 0.05 | Utility change if response |
| $X_{14}$ | 3.0 | 1.0 | Duration of response |
| $X_{15}$ | 0.2 | 0.05 | Probability of side effects |
| $X_{16}$ | -0.1 | 0.02 | Change in utility if side effect |
| $X_{17}$ | 0.5 | 0.2 | Duration of side effects |
| $\theta_1$ | 0.7 | 0.1 | Probability of responding |
| $\theta_2$ | 0.8 | 0.1 | Probability of responding |

*Table 2.* Variables in the health economics experiment.

