# OpenReview forum: "Nested Expectations with Kernel Quadrature"
_ICML.cc/2025/Conference — ICML 2025 poster_

### Official Review · Reviewer_fCUr · 2025-02-23

**Overall Recommendation:** 2

**Summary:**

The authors study the problem of estimating nested expectations, i.e.,
expectations over two entities where we can condition on one of the entities
$E\[g(X, \Theta)\] = E\[E\[g(X, \theta) | \Theta = \theta\]\]$. These
expectations are common in Bayesian statistics. While there are existing
techniques to estimate such expectations via Monte Carlo (e.g. nested QMC or
MLMC), the authors propose their own method nested kernel quadrature (NKQ)
as an alternative approach. The authors demonstrate that when the integrands
are sufficiently smooth, their proposed method can have a faster convergence
rate than aforementioned methods. The authors then employ their method in a
few different applied settings, comparing the accuracy of their method to
alternatives. They demonstrate that their method does indeed perform well,
matching the rates found in their theoretical analysis.

## update after rebuttal
Based on the authors feedback and other reviews, I'm inclined to keep my score of a weak reject. I think the answers have not accurately explained the applicability and computational resources required to apply this method, and without a more moderate explanation of these assumptions, I am not inclined to accept this paper.

**Claims And Evidence:**

The authors provide a thorough theoretical analysis demonstrating the
superior convergence rate of their method. This is also accompanied by three
different empirical studies that corroborate the efficacy of their approach.

**Essential References Not Discussed:**

To the best of the reviewer's knowledge, there aren't any missing references.

**Experimental Designs Or Analyses:**

Yes, the empirical studies appear reasonable. The demonstrate the soundness
of their method on some examples which do not perfectly satisfy their
assumptions.

**Methods And Evaluation Criteria:**

Yes, the datasets and benchmarks used in the paper are reasonable
examples to demonstrate the utility of their method.

**Other Comments Or Suggestions:**

There are some minor typos (search for the word "considere") and also
some missing periods in Appendix F. Moreover, the authors commonly
use big-$O$ notation when they actually should be using $\Omega$ or
$\Theta$ to represent "at least" asymptotically.

**Other Strengths And Weaknesses:**

The idea of using a nested quadrature to estimate these expectations is
interesting and worth the analysis demonstrated in this paper.

The paper is very well written and easy to follow.

However, the biggest issue with this paper is the significance of this
method. The authors do point out that using quadrature methods can have two
bottlenecks for large $N$: they must compute the kernel mean embeddings and
also invert the $N\times N$ Gram matrix, which can be $\Theta(N^3)$. While
the authors do propose the "change of kernels" trick as a workaround, they
do not discuss the limitations this imposes on the distributions $p$ and
$q$.

While it is quite easy to ascertain the pushforward map $\Phi$ when $p$ and
$q$ are univariate or multivariate Gaussians, it does not appear very simple
to find this mapping for any general multivariate $p$ or $q$. When $p$ is a
multivariate distribution with an unknown normalization constant, one
typically employs MCMC to estimate expectations. If it were so easy to
acquire $\Phi$, MCMC would not be necessary.

The authors could do a better job explaining in what circumstances it is
possible to obtain $\Phi$, or at least warn that in most cases, one must
actually use the approximate methods (e.g. Nystrom methods) already
mentioned in the paper.

**Questions For Authors:**

[Q1] Under what conditions on $p$ and $q$ can one obtain the pushforward
function $\Phi$? If one cannot acquire this function, what are the best-case
computational resources needed to employ these methods?

**Relation To Broader Scientific Literature:**

The method described here could provide a more efficient approach for
practitioners of applied statistics or machine learning when one needs to
estimate an expectation over two entities, assuming all their assumptions
hold (more on this below).

**Theoretical Claims:**

There is a section that would merit more discussion. While the authors
describe their "change of variables trick" at the end of page 3 and in
appendix F.1, they do not discuss how difficult it is to determine the
pushforward map $\Phi$ for a given problem. This could be flushed out more,
and is the main weakness of the paper.

---

> ### Author Rebuttal · Authors · 2025-03-28
>
> Thank you for your consideration of the paper. We are very glad to see that you think the idea of using NKQ to estimate these expectations is interesting and the paper is well-written and easy to follow.
>
> ## 1. The pushforward map for change of variable trick
> Thank you for your question about this. We agree that we could have fleshed out a bit more the details of this trick (see also our response to reviewer duxH) and now plan on providing an extensive discussion using the additional space available for the camera-ready version.
>
> This approach is actually general and widely available. For example, for one dimensional distributions, a simple pushforward map is the inverse CDF, in which case the base measure is the uniform distribution. This is typically called inverse transform sampling. Such pushforward maps can be obtained through a multitude of simulation tricks which are widely popular in the Monte Carlo literature; see [Chapter 2 of https://www.cs.fsu.edu/~mascagni/Devroye.pdf] for a very detailed discussion. Additionally, the idea of using such pushforward transformations is the foundational idea behind the field of simulation-based inference, in which case the pushforward map is simply called a ‘simulator’ or ‘generator’ [https://arxiv.org/pdf/1911.01429]. In this particular case, we note that the literature considers exactly the setting where the distribution $p$ is intractable, although it might not cover all cases that you mention.
>
> If the distribution $p$ is known up to a normalization constant, instead of using the change of variable trick, we have a nicer alternative than the change of variable trick by using the Stein reproducing kernel which constructs a new kernel with a standard base kernel and the score function of $p$; see the third bullet point of Section B.1 in https://arxiv.org/pdf/2406.16530 of how it can be applied for kernel / Bayesian quadrature.
>
> When the pushforward map does not exist, our method would incur the computational burden of inverting the Gram matrices. Fortunately, a wide range of techniques already exist in the literature that aim to reduce this cost. See the detailed discussion between lines 138–152 in the right column of page 3 of the paper. We will add a clear sentence in the paper emphasizing the necessity of using these approximate methods in this case.
>
> ## 2. Minors: typos and big O notations
> Thank you for catching these, we will fix these in the revised version of the paper.

---

### Official Review · Reviewer_duxH · 2025-03-13

**Overall Recommendation:** 3

**Summary:**

This paper proposes a novel kernel quadrature estimator for nested integrals involving two integration levels: the inner integral $g(X, \theta)$, which computes the conditional expectation over $X$, and the outer integral, which evaluates the expectation of this conditional expectation over $\theta$. We establish an improved convergence rate of the mean squared error for this nested integral estimator. The effectiveness of the proposed method is demonstrated through synthetic experiments and three real-world applications.

## update after rebuttal
I share Reviewer fCUr’s concerns. The computational aspects remain unclear to me. I maintain original score 3.

**Claims And Evidence:**

Their primary theoretical contribution regarding the improved convergence rate is established in Theorem 1. The empirical advantages of the proposed estimator are demonstrated in Figures 2–4.

**Essential References Not Discussed:**

One of the strengths of this paper is its comprehensive reference list. Although the references are somewhat biased toward statistical approaches, I find the current coverage satisfactory.

**Experimental Designs Or Analyses:**

The experiments are conducted systematically, and the analysis appears reasonable.

**Methods And Evaluation Criteria:**

The absolute error of the integral estimate, defined as $\Delta = \lvert I - \hat{I} \rvert$, is a standard metric in the kernel quadrature literature, and evaluating its convergence rate is a common criterion in theoretical analyses.

**Other Comments Or Suggestions:**

- From a Bayesian perspective, I am slightly hesitant about equating kernel quadrature directly with Bayesian quadrature. Although these terms are often used interchangeably in the literature, their foundational assumptions and underlying interpretations differ significantly. I believe the authors are likely aware of this nuance, but highlighting this distinction explicitly would be beneficial to readers who, like myself, appreciate the philosophical differences between these approaches.
- The references should be refined for consistency and accuracy; for instance, "Conditional bayesian quadrature" should be corrected to "Conditional Bayesian quadrature," capitalizing "Bayesian" appropriately.

**Other Strengths And Weaknesses:**

Strengths
- The paper is very clearly written. Due to the inherent complexity of nested integration tasks, however, a summary table of notations in the appendix would be beneficial. Occasionally, it was challenging to recall the precise meaning of each notation.
- The effort to include real-world applications is commendable, especially given that many papers in this community typically limit themselves to theoretical results without real-world experimental validation. Our community would greatly benefit from increased attention to practical applications beyond purely statistical interests.

Weakness
- The core idea appears somewhat incremental. Although the improved convergence rate is valuable, the idea essentially is applying kernel quadrature twice. Given that kernel quadrature is already known to outperform Monte Carlo and Quasi-Monte Carlo methods in terms of convergence rate, it is unsurprising that its nested version would exhibit superior performance without needing empirical validation. Thus, this work feels more like an “exploitation” of known results rather than an exploration of novel concepts. Readers with a statistical background might enjoy the proof technique, but the fundamental idea itself does not seem particularly innovative.
- Additionally, the application to Bayesian optimization was somewhat disappointing. When reading the introduction, I expected applications to prominent information-theoretic acquisition functions, such as Predictive Entropy Search (PES), Max-value Entropy Search (MES), or Sample Average Approximation (SAA)-based acquisition functions proposed in the original BoTorch paper [1] and Wilson et al. [2]. These functions involve nested integrals, are widely used in the Bayesian optimization community, and improvements here would have significant practical impact. Instead, the authors chose a two-step look-ahead acquisition function, which, although intriguing, lacks popularity in both practical settings and theoretical studies within the Bayesian optimization community.
- I found the change-of-variable trick interesting, but it requires further elaboration. Specifically, the claimed computational efficiency does not seem particularly relevant for the first two examples in finance and healthcare with expensive querying, making it important to clarify its effectiveness in the iterative Bayesian optimization context. The authors state that the computational complexity is linear, $\mathcal{O}(N \times T)$, assuming pre-computation of quadrature weights with the Gram matrix inversion. However, Bayesian optimization involves iterative updates of Gaussian process surrogate models, typically incurring cubic complexity per iteration. It remains unclear how pre-computation mitigates this inherent complexity. Given that popular libraries like GPyTorch and BoTorch already cache quadrature weights during hyperparameter optimization via marginal likelihood, I presume the authors intend to leverage such caching to avoid repeated computations in the acquisition function estimation—am I correct in this interpretation? If the proposed method indeed offers computational advantages comparable to simpler approaches like quasi-Monte Carlo, this should be explicitly demonstrated through experiments for wider audience.

- [1] https://arxiv.org/abs/1910.06403
- [2] https://arxiv.org/abs/1805.10196

**Questions For Authors:**

See weakness.

**Relation To Broader Scientific Literature:**

Numerical integration forms the foundation of probabilistic machine learning and plays a critical role in numerous scientific and engineering applications.

**Theoretical Claims:**

Although I did not verify the correctness of the proof in detail, the results appear reasonable given the known optimality properties of kernel quadrature estimators.

---

> ### Author Rebuttal · Authors · 2025-03-28
>
> Thank you for your consideration of the paper. We are very glad to see that you think the paper is very clearly written and appreciate our effort to include real-world applications. We have carefully addressed all your comments below, but please do let us know if there is anything we can do to help.
>
> ## 1. The core idea appears incremental
> While the core idea may appear simple from an algorithmic standpoint, we view this simplicity as a key strength rather than weakness. In practice, we believe that methods that are conceptually clear and easy to implement are  much more likely to be adopted. In this specific instance, we show that our proposed method can significantly outperform existing approaches despite this algorithmic simplicity, and believe there is therefore clear evidence of the value of our proposal.
>
> The other very strong element of novelty is in terms of the theoretical guarantees we are able to obtain (and the associated proof techniques). Although we appreciate this is something that not all users of our method might be interested in, it is still highly relevant to practitioners. Specifically, we prove a significantly faster convergence rate in $T$ than what is suggested by the existing literature (e.g., [Chen et al., 2024b]). Our rate of $T^{-s_\Theta/d_\Theta}$ notably improves upon the previously established rate of $T^{-1/4}$. This means that one does not need to take $T$ as large as existing results might suggest, which is important for practitioners who might otherwise have wasted significant computation on trying to approximate the outer expectation.
>
> ## 2. Application to other acquisition functions in Bayesian optimization
> Thank you very much for raising this as we were actually not aware that these other acquisition functions were also nested expectations. The reason we decided to focus on the setting of look-ahead optimisation is that there is an entire paper on this topic (in fact written by some of the authors of the BoTorch package) [Yang et al., 2024]. We therefore assumed that this example would be particularly relevant to the BO community. We will add a broader discussion of the applicability of our method to these additional acquisition functions to the camera-ready version.
>
> ## 3. Change of variable trick
> Thank you very much for your thoughts on this. We agree that extending our presentation and discussion of this trick could be very helpful and will do this in the camera-ready version. In the meantime, see below for an answer to each of your sub-comments relating to this.
>
> 1.   ### Applicability of the change-of-variable trick
> We want to clarify that the change-of-variable trick is in fact applicable to all experiments in our paper. We do however agree with you that it is less helpful when the number of function evaluations is already inherently limited since the cost is relatively low in that case anyway. In this paper, the finance and health economics experiments were selected specifically because they are expensive, and so having a better estimator such as NKQ is particularly important. There are however numerous related settings where this is not the case and the change-of-variable trick could still be useful.
>
> 2. ### Cubic complexity
> BO with Gaussian processes indeed requires cubic cost at each iteration, however this cost is cubic in the number of queries of the BO algorithm, not the number of samples $N$ or $T$ for estimating the nested expectation (i.e. the acquisition function). Therefore, precomputation reduces the cost in $N$ and $T$ for optimising the acquisition function at each iteration of the BO algorithm, while the cost of GP posterior update is still cubic and is the same across all methods regardless of how the acquisition function is being approximated. We will make this distinction clear in the revised manuscript.
>
> 3. ### Popular libraries already cache quadrature weights, and I presume the authors intend to leverage such caching to avoid repeated computations
> We do indeed use the idea of caching for the weights of NKQ in estimating the acquisition function so that we do not need to re-compute the weights at each iteration. We are not, however, using NKQ when computing the marginal likelihood in GP hyperparameter selection because it is a standard expectation not a nested expectation.
>
> 4. ### Comparison to QMC
> We conduct an additional experiment with Quasi Monte Carlo ($N=T=\Delta^{-2}$), as shown here https://github.com/Anonymous65536/nest_kq/blob/master/bo.pdf,  demonstrating that while QMC outperforms standard Monte Carlo in estimating the nested expectation within the utility function, it is less efficient than our proposed NKQ method across all three Bayesian optimization datasets.
>
> ## 4. Equating kernel quadrature directly with Bayesian quadrature
> Thank you for pointing this out. We of course agree and will further highlight the distinction between frequentist and Bayesian approaches in the camera-ready version of the paper.

---

### Official Review · Reviewer_uEeW · 2025-03-13

**Overall Recommendation:** 4

**Summary:**

This paper presents a method to estimate nested expectations using (essentially) bilevel application of kernel quadrature. The central result is that this gives a convergence rate that depends on the smoothness constants for each of the two functions in the expectations. This recovers the best known rates in worst-case scenarios, but can be more data-efficient. Separate to that, ideas are proposed to make *computationally* efficient, essentially by changing variables in the integral so it is possible to pre-compute the weights needed for the estimator.

**Claims And Evidence:**

Yes, both clear and convincing.

**Essential References Not Discussed:**

I am not aware of any works that are not cited.

**Experimental Designs Or Analyses:**

Not sure what I would check here. The experiments/analyses seem fine.

**Methods And Evaluation Criteria:**

Yes, the evaluations methods are appropriate. The empirical evidence isn't huge, but it's very well done and I think more than adequate given the theoretical contribution.

**Other Comments Or Suggestions:**

I'd like to remark that I think this paper is very well-written. It's quite technical, but the authors have done a truly excellent job and I think it's nearly as clear and easy to read as is possible given the content. I really appreciate this!

**Other Strengths And Weaknesses:**

This is an extremely minor point that should have no impact on anything, but the paper is pretty heavy with acronyms. In a few cases (e.g. MLMC) I couldn't actually find the original definition. Consider reducing these to the few that are standard and/really used repeatedly (QMC, RKHS, MC, KQ, KME) and/or providing a key in the appendix.

Overall, I think this is a good paper and recommend acceptance. The only reason I shy away from strong accept is the fact that I'm not an expert in this area.

**Questions For Authors:**

One question I had was to what degree is "knowledge of constants" needed? Which of the constants mentioned in Thm. 1 are actually used in the *construction* of the estimator, and not in just the analysis? I'd love to see some analysis of how this estimator compares to previous work in different "constant knowledge regimes". If no smoothness constants are known, does the guarantee reduce to previous bounds?

## Update

After looking at the other reviews, I still recommend accepting this paper. That said, I do agree with the concerns of other reviewers that the scope of the method isn't as large as the authors claim, due to the need to reparameterize. (I have read the authors' response to this, but I do not find it very convincing.) To be clear, I think this is *fine*. I think that the method applies to many problems of real interest, so I do not see this as a problem that needs to be solved. But I agree that the paper should be modified to make this limitation more clear, to avoid over-claiming. This is my only concern.

(To be clear, I still recommend acceptance. My rating of 4 should be considered to be already taking this into account—I would probably recommend 5 if only the paper had made the limitation in scope more clear.)

**Relation To Broader Scientific Literature:**

Nested expectations are an important variant of monte carlo estimation, a classical problem of immense interest. Nested expectations are important enough that even small improvements/insights are worthwhile. I believe this paper easily clears that bar.

**Theoretical Claims:**

I only briefly perused the proofs in the appendix. They look fine, but I didn't really check.

---

> ### Author Rebuttal · Authors · 2025-03-28
>
> Thank you very much for your consideration and strong support of our paper (including the very kind comment about the effort we put in the writing). You mention leaning towards a ‘strong accept (5)’ and we remain at your disposal in case there is anything we can do to help in this respect.
>
> ## 1. Acronyms
> Thank you for pointing this out. We will do our best to limit the use of acronyms to concepts which are repeatedly used, and will check that all acronyms have been properly defined.
>
> ## 2. Knowledge of constants
> Thank you for this interesting and important question – we will certainly add a discussion of this point in the camera-ready version of the paper. Firstly, most constants in Theorem 1 actually have no impact on algorithm design. The constants named $S$ or $G$ only impact $C_1$ and $C_2$, and $C_1$ and $C_2$ do not impact the use of the algorithm in any way.
>
> The only important constants for the algorithm are the smoothness parameters $s_\mathcal{X}$ and $s_{\Theta}$, which help us identify the appropriate level of smoothness to use for kernels $k_{\mathcal{X}}$ and $k_{\Theta}$. This is very common in this literature [Sommariva and Vianello, 2006a, Briol et al., 2019], and even poor choices of these smoothness parameters does not necessarily mean the rate will be impacted [Kanagawa et al., 2020, Wynne et al., 2021]. Although we initially mentioned this very briefly below Theorem 1, we will now use the extra space available to us in the camera-ready version to discuss this more extensively.

---

> > ### Comment · Reviewer_uEeW · 2025-04-03
> >
> > Thanks for the response. After reading it and the other comments, I continue to recommend acceptance. I hear the other reviewers' points regarding significance, but I personally feel the restrictions aren't *that* severe here, and nested expectations are so fundamental that significance is not a major concern for me.

---

> > > ### Author Response · Authors · 2025-04-03
> > >
> > > Thank you very much for your continued support. This is really appreciated!

---

### Official Review · Reviewer_4w5G · 2025-03-17

**Overall Recommendation:** 4

**Summary:**

The paper introduces Nested Kernel Quadrature (NKQ), which a novel method for estimating nested expectations (i.e., integrals where the outer expectation involves a function of an inner expectation). Compared to existing methods such as Nested Monte Carlo (NMC) and Multilevel Monte Carlo (MLMC), which are suitable for functions with relatively more generic assumptions on the smoothness, NKQ considers the case of stronger regularity assumptions and exploits smoothness in the integrand to achieve faster convergence rates by applying existing results from the kernel quadrature (KQ) literature to both the inner and outer expectations. Both theoretical proofs and numerical experiments on synthetic problems, Bayesian optimization, option pricing and health economics are provided to justify the findings.

## Update After Rebuttal
The reviewer agrees that the feasibility of the proposed method needs to be discussed from a practical perspective in the revised version of the manuscript. However, the reviewer is inclined to keep the original score to accept this paper, as most of the theoretical contribution are solid and sound from the reviewer's perspective.

**Claims And Evidence:**

Here is the main claim made in this paper: NKQ achieves faster convergence rates compared to existing methods like NMC and MLMC when the input functions are sufficiently smooth. Both theoretical and empirical results are included to support the claim. Theoretically, the authors proved Theorem 1 and Corollary 1 to establish the faster convergence rates under general regularity assumptions. Empirically, the authors have conducted extensive experiments on synthetic problems, Bayesian optimization, option pricing and health economics to verify that NKQ performs better on sufficiently smooth functions compared to standard methods like NMC and MLMC.

**Essential References Not Discussed:**

To the best of the reviewer's knowledge, all essential references are properly discussed in the manuscript. For some other references that are loosely related, the authors are encouraged to take a look at the series of work listed in the "Other Comments or Suggestions" section below.

**Experimental Designs Or Analyses:**

The experiments are well-designed and cover a range of scenarios, such as synthetic experiments that validate theoretical rates, applications in finance, health economics, and Bayesian optimization, ablation studies on choices of kernels and regularization parameters, as well as careful comparison with baseline methods like NMC, MLMC and NQMC.

**Methods And Evaluation Criteria:**

Yes. Please refer to the "Experimental Designs or Analyses" section below for a detailed discussion.

**Other Comments Or Suggestions:**

N/A

**Other Strengths And Weaknesses:**

The main strength of this paper is that it provides not only solid theory but also extensive experiments to justify the findings. One potential weakness is that in terms of theoretical contribution, it might be beneficial to discuss whether the proposed estimator is minmax optimal or not by deriving an information theoretic lower bound. For instance, one may refer to the bounds derived in [1,2]

References:

[1] Zhang, Zhen, Xin Liu, Shaoli Wang, and Jiaye Teng. "Minimax Optimal Two-Stage Algorithm For Moment Estimation Under Covariate Shift." In The Thirteenth International Conference on Learning Representations.

[2] Blanchet, Jose, Haoxuan Chen, Yiping Lu, and Lexing Ying. "When can regression-adjusted control variate help? rare events, sobolev embedding and minimax optimality." Advances in Neural Information Processing Systems 36 (2023): 36566-36578.

**Questions For Authors:**

N/A

**Relation To Broader Scientific Literature:**

This paper, which proposes the NKQ method and briefly shows that NKQ can also be combined with other advanced Monte Carlo methods like MLMC, should be mainly situated as a combination of Monte Carlo methods and kernel quadratures.

**Theoretical Claims:**

Yes. The convergence rate result (Theorem 1) and its derivation are technically sound under the stated assumptions. The main idea is based on decomposing the error into inner and outer quadrature errors and applying existing bounds from the kernel quadrature literature. The assumptions on smoothness, boundedness, and properties of the kernel are well-justified.

---

> ### Author Rebuttal · Authors · 2025-03-28
>
> Thank you very much for your careful consideration of our paper and for checking the proof of the main theorem. We are very happy that you agree that our convergence rate results (Theorem 1) are technically sound and that our experiments are well-designed.
>
> Thank you for also bringing up the issue of minimax rates and for highlighting papers we were unaware of —this is indeed a fundamental and important question. To the best of our knowledge, most existing approaches on minimax rates for two-stage algorithms first reduce the problem to a one-stage regression by assuming that the target of the second regression is already known. For instance, in the suggested Zhang et al (2025 ) paper, the minimax lower bound provided requires the assumption that the oracle density ratio between the source and target distributions is known (Theorem 1). Similarly, in the literature on nonparametric instrumental variable regression— see for example Theorem 4 and Appendix F.1 in https://arxiv.org/pdf/2411.19653, and Chapter 3 of https://arxiv.org/pdf/0709.2003 — minimax lower rates are provided under the assumption that the conditional expectation operator is known, even though in practice their proposed algorithms would learn this operator from data.
>
> Translating this approach to our setting of nested expectations, we could assume access to the true value of the inner expectation. This effectively reduces the problem to a standard (single-stage) expectation, for which the minimax optimal rate is well-known: $\mathcal{O}(T^{-s_\Theta/ d_\Theta})$ [Sommariva and Vianello, 2006a, Section 2 of https://arxiv.org/pdf/2307.06787]. We agree with you that this is an interesting point to discuss and will do so in the camera-ready version of our paper.
>
> That said, it clearly remains an open challenge to establish minimax lower bounds for genuinely two-stage problems without reducing them to one-stage settings. Developing such a theory would be highly nontrivial and beyond the scope of our paper, but also a very exciting direction for future research.

---

> > ### Comment · Reviewer_4w5G · 2025-04-06
> >
> > The reviewer would like to thank the authors for pointing out a detailed list of related reference on the minimax optimal rates of estimating single-stage expectation. The authors are encouraged to include all references listed above and have a discussion on minimax optimality in the revised version of the manuscript. Overall, the reviewer remains positive about the contribution of this work and will keep the score.

---

> > > ### Author Response · Authors · 2025-04-06
> > >
> > > Thank you very much for your support, we will of course include a detailed discussion of this point in the camera-ready version of the paper.

---

### Decision · Program_Chairs · 2025-05-01

**Decision:**

Accept (poster)

**Comment:**

This is overall a solid paper that should be accepted to the paper. That said, all reviewers highlight that there are computational issues that are not sufficiently addressed by the 'change of variables trick' and these issues should be clearly stated and discussed in the final version.